# STRUCTURAL-ENTROPY-BASED SAMPLE SELECTION FOR EFFICIENT AND EFFECTIVE LEARNING

**Tianchi Xie**[1,*]**, Jiangning Zhu**[1,*]**, Guozu Ma**[2]**, Minzhi Lin**[1]
**Wei Chen**[3]**, Weikai Yang**[4,†]**, Shixia Liu**[1]
[1]BNRist, Tsinghua University [2]China Telecom Wanwei Information Technology Co., Ltd
[3]Microsoft Research [4]Hong Kong University of Science and Technology (Guangzhou)

## ABSTRACT

Sample selection improves the efficiency and effectiveness of machine learning models by providing informative and representative samples. Typically, samples can be modeled as a sample graph, where nodes are samples and edges represent their similarities. Most existing methods are based on local information, such as the training difficulty of samples, thereby overlooking global information, such as connectivity patterns. This oversight can result in suboptimal selection because global information is crucial for ensuring that the selected samples well represent the structural properties of the graph. To address this issue, we employ structural entropy to quantify global information and losslessly decompose it from the whole graph to individual nodes using the Shapley value. Based on the decomposition, we present **S**tructural-**E**ntropy-based sample **S**election (**SES**), a method that integrates both global and local information to select informative and representative samples. SES begins by constructing a $k$NN-graph among samples based on their similarities. It then measures sample importance by combining structural entropy (global metric) with training difficulty (local metric). Finally, SES applies importance-biased blue noise sampling to select a set of diverse and representative samples. Comprehensive experiments in three learning scenarios — supervised learning, active learning, and continual learning — clearly demonstrate the effectiveness of our method.

## 1 INTRODUCTION

Data budgets that limit sample sizes are pervasive in machine learning applications (Liu et al., 2025). For example, researchers and practitioners often face limited annotation and computational resources, necessitating the use of fewer samples to enhance efficiency. Similarly, in continual learning scenarios (Hou et al., 2019), the memory constraint requires fewer samples from previous tasks to retain knowledge effectively. Consequently, effective sample selection becomes crucial to improving efficiency and effectiveness in machine learning. It aims to select informative and representative samples from large datasets to accelerate training and enhance the training performance. Informative samples are those that significantly reduce model uncertainty and are crucial for improving the accuracy and robustness of the training process, while representative samples are those that preserve the diversity and overall distribution of the dataset (Huang et al., 2014). During selection, samples can be modeled as a sample graph, where nodes are samples and edges represent their similarities. Existing sample selection methods primarily focus on local information, such as the training difficulty and the node degree (Maharana et al., 2024). Although these methods demonstrate promising performance on many datasets, they overlook the global information inherent in the graph structure. This global information, such as connectivity patterns, captures the structural properties of the whole graph (Leskovec & Faloutsos, 2006) and has been shown to be effective in improving the representativeness of selected samples (Zhang et al., 2023; Yuan et al., 2021; Zhao et al., 2021). Therefore, we aim to incorporate global information into the sample selection process to improve the quality of the selected samples.

---

*Equal contribution.
†Corresponding author.

The key to incorporating global information is to identify which specific metric(s) can accurately capture the global structure of the sample graph. Entropy is a class of metrics well-suited for this purpose, as it quantifies both informativeness and representativeness (Pan et al., 2005; Li & Guo, 2013). In particular, Li & Pan (2016) propose structural entropy to evaluate the amount of information required to describe a given graph structure. The main feature of this metric is that it is robust and sensitive. First, it remains stable against minor changes like the addition or removal of a few edges. This ensures that the structural entropy reliably reflects the global structure of the graph despite potential noise. Second, it is sensitive to topological changes, especially those affecting connectivity patterns. This is essential for capturing the global structure in response to even small topological changes. These two properties make structural entropy an effective metric for quantifying the global structure and therefore valuable in sample selection. However, existing methods only provide a single value for the whole graph. This presents a challenge in decomposing this metric to the level of individual nodes, limiting its utility for fine-grained, node-level selection.

To address this challenge, we use the Shapley value (Shapley, 1951), a method that fairly decomposes a metric among contributors based on their individual contributions. Specifically, it is calculated by evaluating a node's marginal contribution to structural entropy when adding this node to each subgraph of the sample graph. This decomposition process is highly time-consuming as it requires exponential-time computation to enumerate possible subgraphs. To accelerate this, we reformulate the Shapley value for structural entropy, enabling linear-time calculation with respect to the edge number. Based on this reformulation, we propose a node-level structural entropy metric that effectively measures the importance of nodes in preserving the global structure. Building on the decomposition, we present a **S**tructural-**E**ntropy-based sample **S**election (SES) method that integrates both global and local metrics to select informative and representative samples. This method begins by constructing a $k$NN-graph among samples to describe their similarity relationships. Then, it measures sample importance by combining node-level structural entropy (global metric) with training difficulty (local metric). Finally, the importance-biased blue noise sampling method is employed to iteratively select a set of diverse and representative samples.

We validate the effectiveness of our method through comprehensive experiments on three important learning scenarios: supervised learning, active learning, and continual learning. The evaluation covers many tasks, including image classification, text classification, object detection, and visual question answering. The results clearly show that our method consistently improves state-of-the-art methods across all scenarios and tasks. This indicates that our method of integrating global and local information outperforms existing methods in selecting more informative and representative samples.

The main contributions of this work are threefold:

- We propose a node-level structural entropy metric that quantifies the importance of nodes in preserving the global structure, and it can be calculated in linear time.
- We develop a structural-entropy-based sample selection method that integrates both global and local metrics to select informative and representative samples[1].
- We conduct experiments in supervised learning, active learning, and continual learning that demonstrate the effectiveness of our method.

## 2  RELATED WORK

Existing sample selection methods primarily utilize local information. They can be classified into two categories based on the information utilized: attribute-based methods and connection-based methods.

Attribute-based methods rely on the attributes of individual samples. A commonly used attribute is the training difficulty, which is typically assessed from two perspectives: confidence and error. Metrics that measure model confidence include the entropy of the prediction vector (Coleman et al., 2020) and the variance of the predicted probabilities across training epochs (Swayamdipta et al., 2020). Metrics that measure model error include EL2N (Paul et al., 2021), which calculates the $L_2$ norm of the error vector, and the Forgetting score (Toneva et al., 2019), which tracks the frequency of misclassifications after initial correct classifications. AUM (Pleiss et al., 2020) combines both perspectives by measuring the confidence for correct classifications and the error for misclassifications. Based on these metrics,

---

[1]The implementation is available at https://github.com/thu-vis/SE-based_sample_selection.

several sample selection methods have been developed. One simple yet effective method is selecting the most difficult samples, as they have a larger impact on the model performance (Paul et al., 2021). However, this method overlooks easy samples, which are crucial for model training when data budgets are limited (Sorscher et al., 2022). To address this issue, CCS (Zheng et al., 2022) divides the dataset into strata based on training difficulty and performs random sampling within each stratum. InfoBatch (Qin et al., 2023) retains some easy samples and enhances their influence by upscaling their gradient. Another line of work uses the gradient as the attribute and aims to match the average gradient of the selected samples with that of all samples (Mirzasoleiman et al., 2019; Killamsetty et al., 2021). However, these gradients depend on the model's current state during training, limiting the applicability of the selected samples to other models.

Connection-based methods utilize local connections within the sample graph to optimize sample diversity and coverage. GraphCut (Iyer et al., 2021) selects samples with weak connections among them to promote diversity while maintaining strong connections to unselected samples for better coverage. Moderate coreset (Xia et al., 2023) selects samples based on their distances from the class centers. To enhance the generalizability across different scenarios, samples near the median distance are selected. $\mathbb{D}^2$ Pruning (Maharana et al., 2024) aims to select difficult and diverse samples. It employs forward message passing to integrate training difficulty and node degree, followed by backward message passing to ensure diversity.

While these methods demonstrate promising performance on many datasets, they often overlook the global information, which is crucial for increasing the representativeness of selected samples (Yuan et al., 2021). Overlooking this global information can lead to suboptimal learning performance. To address this gap, we propose a structural-entropy-based sample selection method that integrates both global and local metrics to select informative and representative samples.

## 3  BACKGROUND: STRUCTURAL ENTROPY OF GRAPH

Structural entropy evaluates how the nodes and edges in a graph are hierarchically organized to form communities at different levels (Li & Pan, 2016). Thus, it is effective in globally quantifying the community structure of the graph regarding its overall connectivity patterns. The calculation of structural entropy is based on an encoding tree that represents the graph's hierarchical community structure. In this tree, each node represents a community, and each leaf node corresponds to a graph node that forms a single-node community. With this encoding tree, the entropy is aggregated across communities of different levels, which provides insights into the hierarchical community structure of the graph. Fig. 1 shows an undirected, weighted graph $G$ and its encoding tree $\mathcal{T}$. The entropy is calculated for each non-root node $\alpha$, which considers both intra-community and inter-community connections to reflect the connectivity patterns of its community in the graph. Lower entropy indicates denser intra-community connections and sparser inter-community connections. The intra-community connection, $vol(\alpha)$, is quantified by the total weighted degrees of the nodes, while the inter-community connection, $g(\alpha)$, is quantified by the total weight of the edges with exactly one endpoint in the community (outer edges). Given an encoding tree $\mathcal{T}$, the structural entropy of $G$ is the aggregation of the entropy values from all its non-root nodes:

$$\mathcal{H}(G, \mathcal{T}) = -\sum_{\alpha \in \mathcal{T}} \frac{g(\alpha)}{vol(V)} \log \frac{vol(\alpha)}{vol(\alpha^-)}, \quad (1)$$

where $\alpha$ is a non-root node, $\alpha^-$ is its parent node, $g(\alpha)$ is the total weight of its outer edges, and $vol(V), vol(\alpha), vol(\alpha^-)$ represent the total weighted degrees of the nodes in $V$, $\alpha$, and $\alpha^-$.

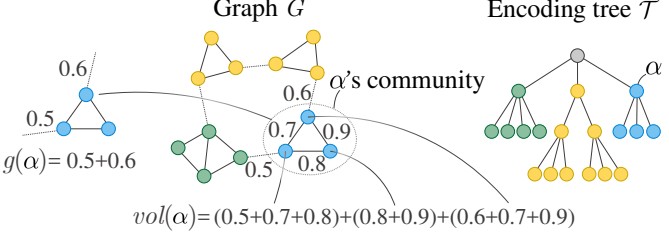

Figure 1: The structural entropy calculation for an undirected, weighted graph.

In real-world applications, the encoding tree $\mathcal{T}$ may be unknown. To best capture the hierarchical community structure in such cases, the encoding tree $\mathcal{T}$ is constructed by minimizing the structural entropy. Obtaining an exact solution for the minimization is challenging, so greedy methods similar to the Huffman tree construction have been developed (Li & Pan, 2016; Zhu et al., 2023). We demonstrate in Appendix A that the choice of the construction method has a negligible effect on the sample selection results. Therefore, we use the most recent method proposed by Zhu et al. (2023).

## 4 STRUCTURAL-ENTROPY-BASED SAMPLE SELECTION

Our sample selection method integrates global and local metrics to select informative and representative samples. Given a large set of samples, an undirected, weighted sample graph $G$ is initially constructed to model their similarity relationships. Each sample, represented by its embedding extracted by a deep neural network, corresponds uniquely to a node in the graph. To avoid excessive edge connections, each sample is connected to its $k$ nearest neighboring samples. The edge weight between any two samples, $u$ and $v$, is their cosine similarity normalized to $[0, 1]$. Based on this graph, we first propose a **node-level structural entropy** metric to globally measure the importance of each sample. Then, it is combined with a local metric, training difficulty, to assign an importance score to each sample. Using this score, we develop an **importance-biased blue noise sampling** method to select a set of informative and representative samples.

### 4.1 NODE-LEVEL STRUCTURAL ENTROPY

The core of our scoring method is to define the metric at the node level. While local metrics are well studied, global metrics have received little attention. An ideal global metric for fine-grained, node-level selection should measure the connectivity patterns of a graph at the individual node level. Previous research shows that the graph-level structural entropy effectively quantifies the global connectivity patterns (Li & Pan, 2016), making it a valuable metric for sample selection. However, it only provides a single value for the whole graph, thus failing to offer detailed insights at the node level. Consequently, the key is to decompose the graph-level structural entropy to the node level.

Chen & Teng (2017) have shown that the Shapley value (Shapley, 1951) is an effective method to decompose a value from the graph level to the node level. The key feature of this method is its lossless and fair decomposition of the value, ensuring that the aggregate node-level value equals the graph-level value. Inspired by this, we employ the Shapley value to derive the node-level structural entropy. Specifically, the Shapley value of a node $u$ reflects the average increase in structural entropy when it is added to all possible subgraphs of $G$. As a result, this value captures the node's contribution to the global connectivity patterns.

To derive the Shapley value for each node, we first calculate the structural entropy for each possible subgraph of $G$. Then, we calculate the node's contribution to these subgraphs. Formally, let $V_S$ denote a subset of the node set $V$, the Shapley value of node $u$ is:

$$\phi(u) = \frac{1}{|V|} \sum_{V_s \subseteq V \setminus \{u\}} \binom{|V|-1}{|V_S|}^{-1} \Big( \mathcal{H}(G[V_S \cup \{u\}], \mathcal{T}) - \mathcal{H}(G[V_S], \mathcal{T}) \Big), \tag{2}$$

where $G[V_S]$ is the subgraph of $G$ that consists of nodes in $V_S$ and the edges between them, and $\binom{|V|-1}{|V_S|}$ is the binomial coefficient.

Directly calculating Eq. (2) requires an enumeration of all possible subgraphs of $G$, which becomes intractable for a graph with a large number of nodes. To address this, we reformulate the Shapley value by considering the contribution of edges.

**Proposition 1.** *Let $G = (V, E, W)$ be an undirected, weighted graph. The Shapley value of node $u$ is*

$$\phi(u) = \frac{1}{vol(V)} \Bigg( \sum_{\langle u,v \rangle \in E} w_{u,v} \log vol(\alpha_{u \vee v}) - d(u) \log d(u) \Bigg), \tag{3}$$

where $w_{u,v}$ is the weight of edge $\langle u, v \rangle$, $\alpha_{u \vee v}$ is the least common ancestor of node $u$ and $v$ in the encoding tree $\mathcal{T}$, and $d(u)$ is the weighted degree of node $u$.

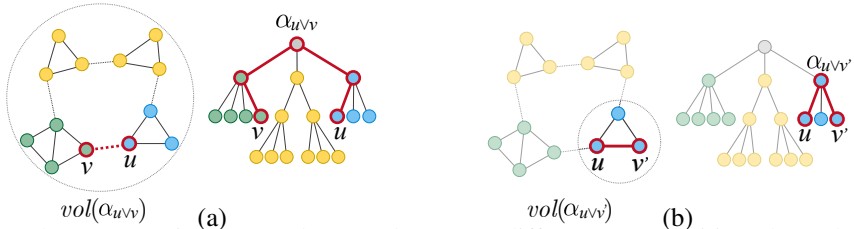

Figure 2: Edges connecting: (a) nodes $u$ and $v$ across different communities; (b) nodes $u$ and $v'$ within the same community.

The proof of Proposition 1 is provided in Appendix B. It indicates that the Shapley value can be calculated in linear time with respect to the edge number. Thus, this reformulation enables an efficient and exact calculation. Eq. (3) consists of two terms. The first decomposes the structural entropy from the encoding tree to the node, and the second reflects local connectivity through node degree. Due to the below theoretical and empirical advantages, we only use the first term to define the node-level structural entropy ($S_e$):

$$S_e(u) = \frac{1}{vol(V)} \sum_{\langle u,v \rangle \in E} w_{u,v} \log vol(\alpha_{u \vee v}). \tag{4}$$

As this definition is related to sample coverage, the first theoretical advantage lies in its ability to enhance model performance. Following previous work (Zheng et al., 2022), we assume that samples are drawn from a distribution $P_\mu$ with probability measure $\mu$, and quantify the sample coverage as $P(u, r) = \int_{B(u,r)} d\mu(x)$, where $B(u, r)$ is a $r$-radius ball centered at $u$. We demonstrate theoretically and empirically that $S_e(u)$ provides a lower bound for sample coverage:

$$\frac{1}{nk^2R} \exp\left(\frac{1}{kR}\mathbb{E}[S_e(u)]\right) \leq \mathbb{E}[P(u, r)], \tag{5}$$

where $n$ is the number of samples, $k$ is the parameter in $k$NN-graph construction, and $R$ is the upper bound of edge weights. This result indicates that maximizing $S_e(u)$ during selection inherently improves sample coverage. Given the strong correlation between coverage ability and the empirical loss (Zheng et al., 2022), selecting samples with high node-level structural entropy enhances model performance. The mathematical proof and the empirical validation are detailed in Appendix C.

The second theoretical advantage is its ability in maintaining the overall graph structure. A higher $vol(\alpha_{u \vee v})$ indicates that the edge $\langle u, v \rangle$ connects the nodes that are more distantly located in $\mathcal{T}$. For example, in Fig. 2, the edge $\langle u, v' \rangle$ stays within the same community while the edge $\langle u, v \rangle$ spans different communities, resulting in a higher $vol(\alpha_{u \vee v})$. Thus, $vol(\alpha_{u \vee v})$ effectively quantifies the extent to which an edge bridges different communities. Since $S_e(u)$ is the weighted sum of $vol(\alpha_{u \vee v})$, nodes with high structural entropy serve as boundaries between communities. Selecting these nodes is crucial for maintaining the overall structure of the graph.

The empirical advantage of this definition is demonstrated through an ablation study, as detailed in Appendix G. The results show that only using the first term slightly improves performance.

## 4.2 IMPORTANCE-BIASED BLUE NOISE SAMPLING

To select a set of high-quality samples, the developed global metric, node-level structural entropy, needs to be combined with an appropriate local metric. Previous research has shown that training difficulty ($S_t$) is an effective local metric in quantifying the sample's impact on model performance, as difficult samples are typically more informative for improving the decision boundary (Paul et al., 2021; Sorscher et al., 2022). Therefore, we employ it as the local metric. Accordingly, the overall importance score ($S$) is a combination of node-level structural entropy and training difficulty:

$$S(u) = S_e(u) \cdot S_t(u). \tag{6}$$

Given the importance scores, a straightforward solution is to select the samples with the highest scores. However, this significantly reduces the diversity of the selected samples, as the important samples tend to cluster in several narrow regions (Zheng et al., 2022). An alternative is the message passing mechanism employed by $\mathbb{D}^2$ Pruning: once a sample is selected, this method sends weighted messages to decrease the importance scores of its neighbors in the graph. However, the message

weights are sensitive to a hyperparameter and can lead to suboptimal results if not carefully tuned. Previous research has shown that blue noise sampling achieves a good balance between randomness and uniformity by excluding overly similar samples (Xiang et al., 2019; Liu et al., 2018). This method increases sampling in low-density regions, which enhances the diversity of the selected samples. Consequently, we develop an importance-biased blue noise sampling method to select a set of informative and representative samples.

Our sampling process contains two steps: 1) identifying the candidate sample with the highest importance score, 2) rejecting the sample if its similarity with any selected neighboring samples exceeds a threshold $\theta$; otherwise, accepting it as a selected sample. These two steps are performed iteratively until no more samples can be selected. To determine the threshold $\theta$ for a given sampling rate, we perform a binary search on $\theta$.

# 5 Experiments

In this section, we first demonstrate the effectiveness of our method in three learning scenarios: supervised learning, active learning, and continual learning. We then conduct ablation studies to provide insights into our method. Finally, we conduct a qualitative analysis of the selection results.

## 5.1 Supervised Learning

In supervised learning tasks, we aim to reduce computational costs by selecting a subset of informative and representative samples for training.

### 5.1.1 Experimental Setup

**Datasets and models**. For image classification, we use the widely used datasets, CIFAR10, CIFAR100 (Krizhevsky, 2009), and ImageNet-1K (Deng et al., 2009). Following Maharana et al. (2024), ResNet-18 (He et al., 2015) is used for CIFAR10 and CIFAR100, while ResNet-34 (He et al., 2015) is used for ImageNet-1K. The models are trained from scratch on the selected subsets of the training set, and we report the model accuracy.

For text classification, we use the ANLI dataset (Nie et al., 2020), which focuses on natural language inference, and the IMDB Review dataset (Maas et al., 2011), which focuses on sentiment analysis. Following Maharana et al. (2024), we fine-tune the RoBERTa model (Liu et al., 2019) and report the accuracy on the test set for both datasets.

For object detection, we use the PASCAL VOC dataset (Everingham et al., 2010), which contains bounding box annotations of objects and animals. Following Choi et al. (2021), we train SSD (Liu et al., 2016) with VGG-16 (Simonyan & Zisserman, 2015) backbone from scratch and report the mAP on the test set.

For visual question answering, we use the CC SBU Align dataset (Zhu et al., 2024), which contains high-quality, aligned image-text pairs. Following Wei et al. (2023), we fine-tune MiniGPT-4 (Zhu et al., 2024) on this dataset, and report the average accuracy of the model on five datasets: OKVQA (Schwenk et al., 2022), IconVQA (Lu et al., 2021), DocVQA (Mathew et al., 2021), GQA (Hudson & Manning, 2019), and ScienceQA (Saikh et al., 2022).

Please refer to Appendix D for more details on the dataset statistics and training hyperparameters.

**Baselines**. We compare our method with the state-of-the-art sample selection methods, which are either applicable to all tasks or designed for a specific task. Baselines that are applicable to all tasks include: 1) **Random** selection of samples, 2) **Moderate** coreset (Xia et al., 2023), 3) **CCS** (Zheng et al., 2022), 4) $\mathbb{D}^2$ **Pruning** (Maharana et al., 2024), and 5) **GraphCut** (Iyer et al., 2021). For image classification and text classification, the task-specific baselines include selecting the most difficult samples based on: 1) **Entropy** (Coleman et al., 2020), 2) **Forgetting** score (Toneva et al., 2019), 3) **EL2N** (Paul et al., 2021), 4) **AUM** (Pleiss et al., 2020), and 5) **Variance** (Swayamdipta et al., 2020). We also include two widely used baselines that prioritize diversity in sample selection, including $k$-**means** (Xu et al., 2003), which selects the samples closest to $k$-means clustering centers, and $k$-**DPP** (Kulesza & Taskar, 2011), which employs a determinantal point process to encourage diversity. For object detection, we include selection based on the **AL-MDN** uncertainty (Choi et al.,

Table 1: Results for supervised learning on: (a) ImageNet-1K; (b) ANLI; (c) PASCAL VOC; (d) CC SBU Align. The best one is **bold**, and the runner-up is underlined.

(a) ImageNet-1K

| Dataset Sampling rate | ImageNet-1K (100%:73.63) | | | | | | |
|---|---|---|---|---|---|---|---|
| | 70% | 50% | 20% | 10% | 5% | 2% | 1% |
| **Random** | 71.63 | 69.26 | 58.90 | 47.10 | 34.04 | 16.56 | 5.50 |
| **Moderate** | 71.33 | 68.72 | 55.23 | 40.97 | 25.75 | 11.33 | 4.52 |
| **CCS** | 70.74 | 69.23 | 60.04 | 50.41 | 36.92 | 19.92 | 9.43 |
| $\mathbb{D}^2$ **Pruning** | 71.29 | 70.32 | 58.91 | 50.81 | 37.12 | 18.97 | 11.23 |
| **GraphCut** | 68.91 | 68.72 | 55.28 | 44.79 | 33.54 | 20.07 | 11.49 |
| **Entropy** | 70.93 | 69.21 | 54.76 | 38.46 | 22.78 | 7.01 | 1.95 |
| **Forgetting** | 70.57 | 70.46 | 60.77 | 48.73 | 33.86 | 15.13 | 5.66 |
| **EL2N** | 71.68 | 65.98 | 31.90 | 12.57 | 6.50 | 3.25 | 1.90 |
| **AUM** | 69.94 | 65.36 | 21.91 | 10.50 | 6.42 | 3.58 | 2.24 |
| **Variance** | 70.12 | 66.09 | 35.15 | 13.85 | 7.13 | 4.72 | 1.81 |
| **$k$-means** | 70.33 | 69.47 | 59.23 | 48.12 | 35.51 | 18.67 | 9.65 |
| **$k$-DPP** | 70.84 | 69.85 | 59.92 | 46.10 | 34.41 | 16.33 | 7.49 |
| **SES (Ours)** | **72.80** | **71.05** | **63.24** | **53.59** | **41.88** | **25.59** | **13.43** |

(b) ANLI

| Dataset Sampling rate | ANLI (100%:49.25) | | | | | | |
|---|---|---|---|---|---|---|---|
| | 70% | 50% | 20% | 10% | 5% | 2% | 1% |
| **Random** | 47.08 | 45.20 | 42.13 | 39.52 | 38.82 | 37.50 | 35.96 |
| **Moderate** | 46.84 | 45.11 | 41.95 | 40.16 | 38.99 | 35.83 | 33.91 |
| **CCS** | 46.56 | 45.92 | 41.67 | 41.63 | 40.33 | 37.41 | 36.82 |
| $\mathbb{D}^2$ **Pruning** | 48.56 | 47.49 | 42.77 | 41.43 | 40.34 | 37.92 | 36.29 |
| **GraphCut** | 46.14 | 44.53 | 42.12 | 39.86 | 38.15 | 35.44 | 34.02 |
| **Entropy** | 46.32 | 45.53 | 41.45 | 39.67 | 38.54 | 36.69 | 36.40 |
| **Forgetting** | 48.73 | 42.29 | 39.82 | 38.37 | 35.95 | 35.78 | 35.03 |
| **EL2N** | 48.70 | 47.85 | 43.14 | 39.63 | 37.52 | 34.33 | 34.27 |
| **AUM** | 47.86 | 47.58 | 43.57 | 40.02 | 34.66 | 34.16 | 33.62 |
| **Variance** | 47.97 | 47.87 | 40.70 | 38.75 | 33.52 | 33.50 | 33.17 |
| **$k$-means** | 46.48 | 46.52 | 42.42 | 40.57 | 39.89 | 36.74 | 36.11 |
| **$k$-DPP** | 47.74 | 47.02 | 43.44 | 40.98 | 40.12 | 37.44 | 36.66 |
| **SES (Ours)** | **49.00** | **48.22** | **45.94** | **43.63** | **41.82** | **39.88** | **38.16** |

(c) PASCAL VOC

| Dataset Sampling rate | PASCAL VOC (100%:76.29) | | | | | | |
|---|---|---|---|---|---|---|---|
| | 70% | 50% | 20% | 10% | 5% | 2% | 1% |
| **Random** | 74.02 | 72.10 | 65.45 | 57.56 | 43.47 | 18.78 | 9.24 |
| **Moderate** | 73.42 | 72.03 | 65.12 | 54.71 | 40.20 | 15.97 | 5.13 |
| **CCS** | 74.64 | 72.27 | 65.72 | 57.35 | 39.01 | 17.26 | 8.49 |
| $\mathbb{D}^2$ **Pruning** | 74.46 | 72.55 | 65.59 | 55.73 | 44.04 | 19.16 | 10.75 |
| **GraphCut** | 67.45 | 64.15 | 53.12 | 38.29 | 26.81 | 8.56 | 8.16 |
| **AL-MDN** | 74.51 | 70.36 | 65.26 | 54.51 | 30.85 | 12.33 | 8.97 |
| **$k$-means** | 74.22 | 72.35 | 65.52 | 57.01 | 43.35 | 16.96 | 5.16 |
| **$k$-DPP** | 74.19 | 71.99 | 65.39 | 56.80 | 43.92 | 18.20 | 10.50 |
| **SES (Ours)** | **75.20** | **73.33** | **66.52** | **59.52** | **45.92** | **23.39** | **16.15** |

(d) CC SBU Align

| Dataset Sampling rate | CC SBU Align (100%:30.40) | | | | | | |
|---|---|---|---|---|---|---|---|
| | 70% | 50% | 20% | 10% | 5% | 2% | 1% |
| **Random** | 29.66 | 29.62 | 29.21 | 29.01 | 28.20 | 25.51 | 25.11 |
| **Moderate** | 29.96 | 29.67 | 29.53 | 29.11 | 27.30 | 24.85 | 26.54 |
| **CCS** | 29.93 | 29.90 | 29.94 | 29.91 | 27.71 | 25.31 | 25.59 |
| $\mathbb{D}^2$ **Pruning** | 30.09 | 29.97 | 29.44 | 29.30 | 26.44 | 25.03 | 26.29 |
| **GraphCut** | 29.86 | 29.73 | 29.53 | 29.11 | 27.30 | 24.85 | 26.54 |
| **Instruction** | 30.12 | 29.93 | 29.82 | 29.01 | 26.76 | 23.72 | 24.60 |
| **$k$-means** | 29.78 | 29.61 | 29.54 | 29.20 | 27.72 | 25.47 | 25.60 |
| **$k$-DPP** | 29.33 | 29.55 | 29.48 | 29.27 | 28.16 | 25.93 | 26.13 |
| **SES (Ours)** | **30.25** | **30.20** | **30.21** | **30.10** | **28.23** | **27.19** | **27.61** |

2021), which captures the detector's overall uncertainty for an image. For visual question answering, we include selection based on the **Instruction** score (Wei et al., 2023), which evaluates an image-text pair based on image-text matching degree and text length.

**Implementation**. For all tasks, we extract image embeddings using CLIP (Radford et al., 2021) and text embeddings using Sentence-BERT (Reimers & Gurevych, 2019) due to their demonstrated performance in capturing the semantic similarities. For visual question answering, we concatenate the image and text embeddings for each sample. To measure training difficulty, we use AUM for image classification, Variance for text classification, AL-MDN uncertainty for object detection, and Instruction score for visual question answering. We ablate the different training difficulty metrics in Appendix G and observe no significant performance difference among them. We also perform a grid search on the hyperparameters, such as $k$ in the $k$NN-graph construction (see Appendix E for details).

### 5.1.2 RESULTS

To cover a wide range of sampling rates, we select subsets that contain 1%, 2%, 5%, 10%, 20%, 50%, and 70% of the entire training set. All the results are averaged over 5 runs. Table 1 shows the results on four datasets that cover all the four tasks. The full results are provided in Appendix F.

Baselines that select the most difficult samples, such as AUM and Forgetting, perform well in high-sampling-rate settings. However, these methods fall behind in low-sampling-rate settings. This is due to their limited coverage of easy samples, which are crucial for model training when fewer samples are selected (Sorscher et al., 2022). Methods that prioritize sample coverage, such as CCS and $\mathbb{D}^2$ Pruning, address this issue and perform well in low-sampling-rate settings. However, in high-sampling-rate settings, they cannot accurately determine the most important samples that preserve global structure. This results in a lack of representativeness in the selected samples and suboptimal performance. Methods that prioritize diversity are competitive in low-sampling-rate settings because they ensure the coverage of the dataset. However, in high-sampling-rate settings, they face challenges in balancing diversity with sample importance, leading to suboptimal performance. In contrast, our method integrates both global and local metrics to better identify important samples and employs importance-biased blue noise sampling to ensure representativeness. Therefore, our method consistently performs better than baselines across all sampling rates and datasets.

In text classification and visual question answering, we observe that decreasing the sampling rate does not significantly affect performance. This is because we are fine-tuning pretrained models, which provide sufficient knowledge for these tasks. By providing high-quality samples, our method significantly accelerates fine-tuning with negligible performance loss. For example, as shown in

Table 1(d), we achieve a 10x speedup by using only $10\%$ of the dataset to fine-tune MiniGPT-4, with only a $0.3\%$ drop in accuracy compared to using the entire dataset. Specifically, our method reduces the fine-tuning time on a single Nvidia Tesla V100 GPU from approximately 30 minutes to 3 minutes, adding only a negligible selection overhead of 2 seconds.

## 5.2 ACTIVE LEARNING

Active learning (Settles, 2009) aims to reduce annotation effort by selecting a set of informative and representative samples from an unlabeled pool. These samples are then labeled to train models. The key difference between active learning and supervised learning lies in the absence of labels during sample selection, which makes the selection process more challenging.

### 5.2.1 EXPERIMENTAL SETUP

**Datasets and models**. To evaluate the effectiveness of the selection methods in an active learning task, we perform image classification on ImageNet-1K. To simulate an unlabeled pool, we remove the labels from all samples during selection. After selection, we use the ground-truth labels to simulate human annotations. We train ResNet-34 from scratch on these labeled images and report the accuracy on the ImageNet-1K validation set.

**Baselines**. We include the baselines from Sec. 5.1 that are applicable to unlabeled datasets. Additionally, we include **Prototypicality** (Sorscher et al., 2022) designed for unlabeled datasets. This method selects the most difficult samples based on the prototypicality score, which is defined as the distance between samples and their corresponding $k$-means cluster center. Difficult samples are those far from the center, as they tend to be more ambiguous than the samples closer to the center.

**Implementation**. In the active learning scenario, using a pretrained supervised model like CLIP for feature extraction is not suitable, because the domain of the unlabeled data may not be covered by its pretraining data. Therefore, we extract the image embeddings with a self-supervised model, SwAV (Caron et al., 2020). The prototypicality score is utilized to measure training difficulty.

### 5.2.2 RESULTS

We select unlabeled samples with rates of $1\%$, $2\%$, $5\%$, $10\%$, $20\%$, $50\%$, and $70\%$ and report the results averaged over 5 random seeds in Table 2. In low-sampling-rate settings, other baselines perform worse than random selection due to the absence of labels, indicating their reliance on labeled data to achieve optimal performance. In contrast, our method consistently performs better than random selection and other baseline methods. This is because structural entropy compensates for missing labels by capturing the community structure in the datasets.

## 5.3 CONTINUAL LEARNING

Continual learning (Kirkpatrick et al., 2017) aims to alleviate the catastrophic forgetting of previously learned tasks when learning new tasks over time. We focus on the replay-based method (Hou et al., 2019), which selects a small set of informative and representative samples from previous tasks and replays them during the training of new tasks.

### 5.3.1 EXPERIMENTAL SETUP

**Datasets and models**. We use the datasets commonly used in continual learning, including Permuted MNIST, Split MNIST, Split CIFAR10, Split CIFAR100, and Split Tiny-ImageNet. Permuted MNIST

Table 2: Results for active learning. The best one is **bold**, and the runner-up is underlined.

| Dataset | ImageNet-1K (100%:73.63) | | | | | | |
|---|---|---|---|---|---|---|---|
| Sampling rate | 70% | 50% | 20% | 10% | 5% | 2% | 1% |
| Random | 71.12 | 69.43 | 58.77 | 47.36 | 33.41 | 16.41 | 5.41 |
| Moderate | 71.48 | 68.68 | 56.35 | 42.29 | 26.77 | 11.37 | 4.22 |
| CCS | 71.46 | 69.50 | 58.85 | 45.06 | 28.02 | 9.03 | 2.33 |
| GraphCut | 71.50 | 69.16 | 56.08 | 40.99 | 24.30 | 7.90 | 2.46 |
| $\mathbb{D}^2$ Pruning | 71.62 | 69.02 | 59.65 | 45.97 | 28.08 | 14.24 | 4.79 |
| Prototypicality | 70.12 | 66.00 | 49.20 | 35.27 | 24.14 | 13.88 | 4.95 |
| SES (Ours) | **72.11** | **70.15** | **60.22** | **48.10** | **34.82** | **17.97** | **6.69** |

Table 3: Results for continual learning. The best one is **bold**, and the runner-up is underlined.

| Dataset | Permuted MNIST | | Split MNIST | | Split CIFAR10 | | Split CIFAR100 | | Split Tiny-ImageNet | |
|---|---|---|---|---|---|---|---|---|---|---|
| Memory size | 100 | 200 | 100 | 200 | 100 | 200 | 100 | 200 | 100 | 200 |
| $\mathbb{D}^2$ **Pruning** | 78.25 | 79.94 | 96.79 | 97.69 | 64.54 | 66.08 | 51.86 | 54.50 | 19.08 | 19.50 |
| **GraphCut** | 76.98 | 78.61 | 91.34 | 94.25 | 61.02 | 61.66 | 53.35 | 54.66 | 19.76 | 20.53 |
| **$k$-center** | 78.17 | 79.75 | 94.39 | 96.61 | 61.47 | 62.74 | 51.16 | 53.10 | 18.87 | 18.90 |
| **Gradient Matching** | 77.30 | 79.27 | 95.39 | 97.54 | 61.65 | 62.65 | 54.13 | 56.29 | 19.19 | 19.00 |
| **FRCL** | 77.33 | 79.21 | 94.48 | 97.10 | 61.67 | 62.93 | 51.40 | 54.28 | 18.86 | 19.01 |
| **iCaRL** | 78.94 | 80.65 | 89.50 | 97.59 | 62.33 | 64.08 | 54.62 | 56.11 | 19.58 | 19.85 |
| **Greedy Coreset** | 78.71 | 80.13 | 96.07 | 97.76 | 63.18 | 62.98 | 56.17 | 57.72 | 19.24 | 19.98 |
| **BCSR** | 77.74 | 79.51 | 94.77 | 96.98 | 63.23 | 64.59 | 50.21 | 51.49 | 18.75 | 18.74 |
| **SES (Ours)** | **79.92** | **81.18** | **96.94** | **98.28** | **68.26** | **69.32** | **57.60** | **59.69** | **20.80** | **21.20** |

splits MNIST (LeCun et al., 1998) into 10 segments, where a fixed permutation of the pixel order is applied to all images in each segment to simulate different distributions. Thus, it contains 10 classification tasks with samples from different distributions. The other four datasets split the image classes in MNIST, CIFAR10, CIFAR100, and Tiny-ImageNet (Le & Yang, 2015) into 5, 5, 20, and 20 segments, respectively, and each segment corresponds to a different classification task. In alignment with prior studies (Borsos et al., 2020; Hao et al., 2024), we use increasingly complex models as dataset complexity grows: a two-layer MLP for Permuted MNIST, a four-layer CNN for Split MNIST, ResNet-18 for Split CIFAR10, and ResNet-18 with multi-head output (Zenke et al., 2017) for Split CIFAR100 and Split Tiny-ImageNet. We conduct the experiments under three settings: class-incremental, task-incremental, and task-free (Wang et al., 2024). In the task-incremental setting, the models are trained sequentially for each task while maintaining a fixed-size replay memory that contains an equal number of samples for each previous task. During training, samples from both the replay memory and the current task are used, with the replay samples weighted by a hyperparameter that controls their influence. After each task is completed, samples are selected from the current task to replace a portion of the replay memory. We report the average accuracy on all tasks, with the model unaware of a sample's task during testing. The other two settings are introduced in Appendix F.

**Baselines**. We include baselines from Sec. 5.1 to select samples that update the replay memory. Additionally, we include nine widely used selection methods for continual learning: 1) **$k$-center** (Sener & Savarese, 2018), 2) **Gradient Matching** (Campbell & Broderick, 2019), 3) **FRCL** (Titsias et al., 2020), 4) **iCaRL** (Rebuffi et al., 2017), 5) **Greedy Coreset** (Borsos et al., 2020), 6) **BCSR** (Hao et al., 2024), 7) **DER++** (Buzzega et al., 2020), 8) **GSS** (Aljundi et al., 2019), and 9) **GCR** (Tiwari et al., 2022). Detailed introduction of these methods is provided in Appendix F.

**Implementation**. We extract the image embeddings with CLIP and measure training difficulty with AUM. For all baselines and our method, we perform a grid search on the weight of replay samples to determine its optimal value (see Appendix D for details).

### 5.3.2 RESULTS

We test replay memory sizes of 50, 100, 200, and 400 and report the results averaged over 5 random seeds. Due to space limitations, Table 3 shows the results for replay memory sizes of 100 and 200 under the class-incremental setting, excluding baselines from Sec. 5.1 that are neither the best nor the runner-up in any dataset or memory size. The full results are provided in Appendix F. Our method consistently achieves better performance than baselines across all memory sizes and datasets. This is because the combination of node-level structural entropy and training difficulty captures both the global structure of samples and the model training dynamics in retaining knowledge.

Table 4: Ablation of modules. The best one is **bold**, and the runner-up is underlined.

| Module | | Sampling rate | | | | | | | Avg. |
|---|---|---|---|---|---|---|---|---|---|
| Scoring | Sampling | 70% | 50% | 20% | 10% | 5% | 2% | 1% | |
| TD | HS | 94.85 | 93.90 | 70.88 | 60.68 | 47.36 | 38.30 | 32.41 | 62.63 |
| TD | MP | 94.86 | 94.28 | 87.88 | 77.68 | 65.30 | 49.69 | 41.41 | 73.01 |
| TD | BNS | 94.88 | 94.05 | 88.06 | 79.55 | 68.05 | 53.37 | 43.07 | 74.43 |
| SE+TD | HS | 94.92 | 94.39 | 82.03 | 70.42 | 57.63 | 40.64 | 33.68 | 67.67 |
| SE+TD | MP | **95.08** | **94.67** | **88.54** | 80.14 | 69.34 | 54.26 | 44.51 | 75.22 |
| SE+TD | BNS | 95.01 | 94.50 | 88.31 | **80.24** | **69.82** | **54.78** | **45.25** | **75.42** |

## 5.4 ABLATION STUDY

We conduct ablation studies on supervised learning using CIFAR10 to evaluate the impact and behavior of each module in our method. This section presents results on the effect of modules. Additional ablation studies are provided in Appendix G.

**Effect of modules**. We ablate the two key modules in our method, node-level structural entropy (SE) and importance-biased blue noise sampling (BNS), by replacing them with alternatives. Without SE, we score samples based solely on training difficulty (TD). Without BNS, we either select samples with the highest scores (HS) or use the message passing (MP) method from $\mathbb{D}^2$ pruning. The hyperparameter controlling message weights in MP is determined through grid search as in the original paper (Maharana et al., 2024). Table 4 shows the results. Using TD and HS yields the lowest performance. Incorporating either SE, BNS, or MP largely improves performance, demonstrating their individual effectiveness. Combining SE with either BNS or MP further improves the performance, indicating that SE complements both methods in selecting informative and representative samples. Notably, BNS achieves comparable performance to MP without the need for the time-consuming grid search on the additional hyperparameter, demonstrating its effectiveness.

## 5.5 QUALITATIVE ANALYSIS

We visualize the selection results of different methods when selecting $2\%$ of the samples from CIFAR10 by projecting them onto a two-dimensional plane using t-SNE (van der Maaten & Hinton, 2008). Fig. 3 shows the results of AUM, $\mathbb{D}^2$ Pruning, and our method. Each point is a selected sample, and the colored contours indicate the high-density regions for each class. The full results are provided in Appendix H. Methods that select the most difficult samples, such as AUM, oversample near several class boundaries and undersample in several classes that are easier to classify (Fig. 3(a)). Methods that prioritize sample coverage, such as $\mathbb{D}^2$ Pruning, achieve a better sample coverage but still miss critical samples near class boundaries (Fig. 3(b)). This gap indicates that these methods may not effectively preserve the global structure of the samples. Our method well covers the data distribution, providing a set of informative and representative samples for model training (Fig. 3(c)).

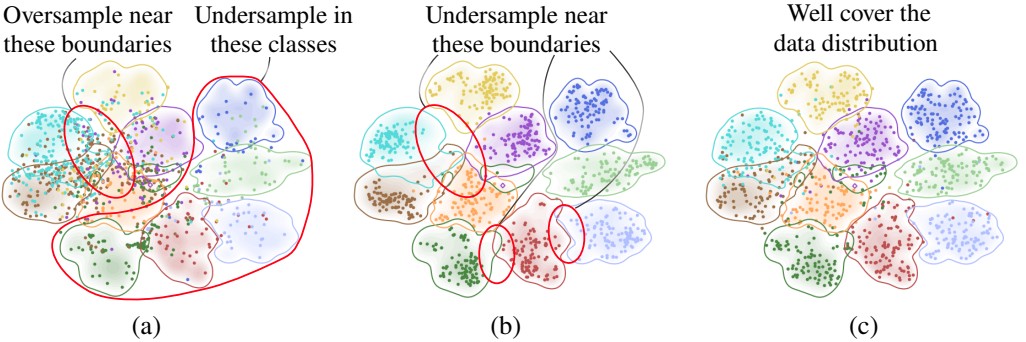

Figure 3: Visualizations of results when selecting $2\%$ of the samples from CIFAR10 using: (a) AUM; (b) $\mathbb{D}^2$ Pruning; (c) our structural-entropy-based sample selection method.

## 6 CONCLUSION

In this paper, we present a structural-entropy-based sample selection method for efficient and effective learning. The key idea behind our method is to decompose graph-level structural entropy to a node-level global metric using the Shapley value. This global metric is combined with a local metric, training difficulty, for selecting informative and representative samples. The effectiveness of our method is validated by comprehensive experiments in three learning scenarios. Although our method has proven effective, future work on the following aspects is still promising. First, automating the hyperparameter selection based on data characteristics can reduce the computational costs of the grid search. Second, improving the support for multimodal data could strengthen its performance across a wider range of tasks, such as infographics VQA (Mathew et al., 2022) and building foundation models (Yang et al., 2024). Third, applying techniques such as hyper-class representation (Zhang et al., 2022) to enhance the image and text embeddings can improve the quality of selected samples.

# 7 ACKNOWLEDGEMENTS

Tianchi Xie, Jiangning Zhu, Minzhi Lin, and Shixia Liu are supported by the National Natural Science Foundation of China under grant U21A20469 and in part by the Tsinghua University-China Telecom Wanwei Joint Research Center. The authors would like to thank Duan Li, Yukai Guo, and Zhen Li for their valuable discussions and comments.

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

## A    COMPARISON OF ENCODING TREE CONSTRUCTION METHODS

The construction of the encoding tree is an important step in our method. To the best of our knowledge, there are two methods for constructing the encoding tree: the one proposed by Li & Pan (2016) and the one proposed by Zhu et al. (2023). Both methods adopt the variations of the Huffman tree to construct the encoding tree. The key difference is that Zhu et al. (2023) further compresses the tree to a certain height to improve efficiency in subsequent processing. We assess the two methods by calculating the Pearson correlation of the resulting node-level structural entropy. As shown in Table 5, the correlations on CIFAR10, CIFAR100, and ImageNet-1K exceed 0.99. Therefore, the choice of the construction method has a negligible effect on the sample selection results. In this paper, we employ the more recent method proposed by Zhu et al. (2023).

Table 5: Pearson correlation of the node-level structural entropy obtained from the two methods on CIFAR10, CIFAR100, and ImageNet-1K.

|  | CIAFR10 | CIFAR100 | ImageNet-1K |
|---|---|---|---|
| Correlation | 0.996 | 0.999 | 0.992 |

## B    PROOF OF PROPOSITION 1

We first present two lemmas essential for the proof of Proposition 1.

**Lemma 1.** *Let $G = (V, E, W)$ be an undirected, weighted graph and $\mathcal{T}$ be its encoding tree. Then the structural entropy $\mathcal{H}(G, \mathcal{T})$ can be written as:*

$$\mathcal{H}(G, \mathcal{T}) = \frac{1}{vol(V)} \left( 2 \sum_{\langle u,v \rangle \in E} w_{u,v} \log vol(u \vee v) - \sum_{u \in V} d(u) \log d(u) \right),$$

*where $w_{u,v}$ is the weight of edge $\langle u, v \rangle$, $u \vee v$ is the least common ancestor of node $u$ and $v$ in $\mathcal{T}$, and $d(u)$ is the weighted degree of node $u$.*

*Proof of Lemma 1.* According to Eq. (1), we can derive that:

$$
\begin{aligned}
\mathcal{H}(G, \mathcal{T}) &= -\sum_{\alpha \in \mathcal{T}} \frac{g(\alpha)}{vol(V)} \log \frac{vol(\alpha)}{vol(\alpha^-)} \\
&= -\frac{1}{vol(V)} \sum_{\alpha \in \mathcal{T}} g(\alpha) \log \frac{vol(\alpha)}{vol(\alpha^-)} \\
&= \frac{1}{vol(V)} \sum_{\alpha \in \mathcal{T}} (g(\alpha) \log vol(\alpha^-) - g(\alpha) \log vol(\alpha)).
\end{aligned}
\tag{7}
$$

For a node $\alpha$ with $k$ children, the contribution of each child $\beta_i$ to the summation is $g(\beta_i) \log vol(\alpha) - g(\beta_i) \log vol(\beta_i)$. The term $g(\beta_i) \log vol(\alpha)$ can be combined into $-g(\alpha) \log vol(\alpha)$ due to the shared $\log vol(\alpha)$ factor. By combining the terms with shared $\log vol(\cdot)$ factors, Eq. (7) can be rewritten as:

$$\mathcal{H}(G, \mathcal{T}) = \frac{1}{vol(V)} \left( \sum_{\alpha \text{ is non-leaf}} \left( \left( \sum_{\beta \in \text{children}(\alpha)} g(\beta) \right) - g(\alpha) \right) \log vol(\alpha) - \sum_{\alpha \text{ is leaf}} g(\alpha) \log vol(\alpha) \right).
\tag{8}$$

Note that $\sum_{\beta \in \text{children}(\alpha)} g(\beta) - g(\alpha)$ represents the difference between the total weight of outer edges of $\alpha$'s children and the total weight of outer edges of $\alpha$. This difference is precisely twice the total weights of edges between the communities represented by $\alpha$'s children. Furthermore, each edge

$\langle u, v \rangle$ in the graph contributes only to the node $\alpha = u \vee v$. Therefore, we can transform the first term into a sum over edges in the graph, yielding:

$$\mathcal{H}(G, \mathcal{T}) = \frac{1}{vol(V)} \left( 2 \sum_{\langle u,v \rangle \in E} w_{u,v} \log vol(u \vee v) - \sum_{\alpha \text{ is leaf}} g(\alpha) \log vol(\alpha) \right). \tag{9}$$

The set of leaf nodes in the encoding tree corresponds to the set of nodes in the graph. Additionally, for a leaf node $\alpha$ and its correpsonding graph node $u$, we have $g(\alpha) = vol(\alpha) = d(u)$. We can conclude that:

$$\mathcal{H}(G, \mathcal{T}) = \frac{1}{vol(V)} \left( 2 \sum_{\langle u,v \rangle \in E} w_{u,v} \log vol(u \vee v) - \sum_{u \in V} d(u) \log d(u) \right), \tag{10}$$

which proves the lemma.

**Lemma 2** (Winter, 2002). *Eq. (2) is equivalent to:*

$$\phi(u) = \frac{1}{|V|!} \sum_{\pi \in \Pi} \left( \mathcal{H}(G[V_{\pi,u} \cup \{u\}], \mathcal{T}) - \mathcal{H}(G[V_{\pi,u}], \mathcal{T}) \right), \tag{11}$$

*where $\Pi$ is the set of all permutations of nodes in $V$, and $V_{\pi,u}$ denotes the set of nodes preceding $u$ in permutation $\pi$.*

Based on the two lemmas, we develop a proof of Proposition 1.

*Proof of Proposition 1.* For brevity, we denote the subgraph $G[V_{\pi,u} \cup \{u\}]$ by $G_{\pi,u}^+ = (V_{\pi,u}^+, E_{\pi,u}^+, W)$ and $G[V_{\pi,u}]$ by $G_{\pi,u} = (V_{\pi,u}, E_{\pi,u}, W)$. Lemma 1 gives:

$$\mathcal{H}(G_{\pi,u}^+, \mathcal{T}) = \frac{1}{vol(V)} (2 \sum_{\langle x,y \rangle \in E_{\pi,u}^+} w_{x,y} \log vol(x \vee y) - \sum_{x \in V_{\pi,u}^+} d(x) \log d(x)) \tag{12}$$

and

$$\mathcal{H}(G_{\pi,u}, \mathcal{T}) = \frac{1}{vol(V)} (2 \sum_{\langle x,y \rangle \in E_{\pi,u}} w_{x,y} \log vol(x \vee y) - \sum_{x \in V_{\pi,u}} d(x) \log d(x)). \tag{13}$$

Then

$$\mathcal{H}(G_{\pi,u}^+, \mathcal{T}) - \mathcal{H}(G_{\pi,u}, \mathcal{T})$$
$$= \frac{1}{vol(V)} \left( 2( \sum_{\langle x,y \rangle \in E_{\pi,u}^+} w_{x,y} \log vol(x \vee y) - \sum_{\langle x,y \rangle \in E_{\pi,u}} w_{x,y} \log vol(x \vee y)) \right.$$
$$\left. - ( \sum_{x \in V_{\pi,u}^+} d(x) \log(x) - \sum_{x \in V_{\pi,u}} d(x) \log d(x)) \right). \tag{14}$$

Note that $V_{\pi,u}^+ = V_{\pi,u} \cup \{u\}$ and $E_{\pi,u}^+ = E_{\pi,u} \cup \{\langle u,v \rangle : v \in \mathcal{N}(u) \cap V_{\pi,u}\}$, where $\mathcal{N}(u)$ is the set of neighbors of $u$ in $G$. Therefore,

$$\mathcal{H}(G_{\pi,u}^+, \mathcal{T}) - \mathcal{H}(G_{\pi,u}, \mathcal{T}) = \frac{1}{vol(V)} \left( 2 \sum_{v \in \mathcal{N}(u) \cap V_{\pi,u}} w_{u,v} \log vol(u \vee v) - d(u) \log d(u) \right). \tag{15}$$

We can rewrite $\sum_{v \in \mathcal{N}(u) \cap V_{\pi,u}} \log vol(u \vee v)$ as $\sum_{v \in \mathcal{N}(u)} \mathbb{I}[v \in V_{\pi,u}] \log vol(u \vee v)$, where $\mathbb{I}$ is the indicator function. Then, the Shapley value in Eq. (11) can be rewritten as:

$$
\begin{aligned}
\phi(u) &= \frac{1}{|V|!} \sum_{\pi \in \Pi} \Big( \mathcal{H}(G[V_{\pi,u} \cup \{u\}], \mathcal{T}) - \mathcal{H}(G[V_{\pi,u}], \mathcal{T}) \Big) \\
&= \frac{1}{|V|!} \sum_{\pi \in \Pi} \left( \frac{1}{vol(V)} \left( 2 \sum_{v \in \mathcal{N}(u)} \mathbb{I}[v \in V_{\pi,u}] w_{u,v} \log vol(u \vee v) - d(u) \log d(u) \right) \right) \\
&= \frac{2}{vol(V)|V|!} \sum_{\pi \in \Pi} \sum_{v \in \mathcal{N}(u)} \mathbb{I}[v \in V_{\pi,u}] w_{u,v} \log vol(u \vee v) - \frac{1}{vol(V)} d(u) \log d(u) \\
&= \frac{2}{vol(V)|V|!} \sum_{v \in \mathcal{N}(u)} w_{u,v} \log vol(u \vee v) \sum_{\pi \in \Pi} \mathbb{I}[v \in V_{\pi,u}] - \frac{1}{vol(V)} d(u) \log d(u).
\end{aligned}
\tag{16}
$$

Note that $\sum_{\pi \in \Pi} \mathbb{I}[v \in V_{\pi,u}]$ is $\frac{|V|!}{2}$ since there are $\frac{|V|!}{2}$ permutations where $v$ precedes $u$. Thus, we have:

$$
\begin{aligned}
\phi(u) &= \frac{1}{vol(V)} \sum_{v \in \mathcal{N}(u)} w_{u,v} \log vol(u \vee v) - \frac{1}{vol(V)} d(u) \log d(u) \\
&= \frac{1}{vol(V)} \left( \sum_{v \in \mathcal{N}(u)} w_{u,v} \log vol(\alpha_{u \vee v}) - d(u) \log d(u) \right) \\
&= \frac{1}{vol(V)} \left( \sum_{\langle u,v \rangle \in E} w_{u,v} \log vol(\alpha_{u \vee v}) - d(u) \log d(u) \right),
\end{aligned}
\tag{17}
$$

which proves the proposition.

## C  THEORETICAL ANALYSIS OF NODE-LEVEL STRUCTURAL ENTROPY

### C.1  NODE-LEVEL STRUCTURAL ENTROPY AND SAMPLE COVERAGE

In this section, we prove that node-level structural entropy serves as a lower bound for the sample coverage. Following previous work (Zheng et al., 2022), we assume that the dataset $S = \{\mathbf{x}_i, y_i\}$ contains $n$ i.i.d. samples drawn from an underlying distribution $P_\mu$ with probability measure $\mu$. We discuss a certain sample $u$'s ability to cover the entire distribution, i.e., how well the sample can cover the entire distribution. The sample coverage of a sample $\mathbf{u}$ can be quantified by the coverage of the entire distribution by a ball $B(\mathbf{u}, r)$ centered at $\mathbf{u}$ with radius $r$ (Zheng et al., 2022):

$$
P(u, r) = \int_{B(u,r)} \mathrm{d}\mu(x).
$$

When $r$ is fixed, the larger $P(u, r)$ is, the better sample covers the distribution. Directly calculating $P(u, r)$ is difficult because we do not know the underlying distribution $P_\mu$. We next prove that node-level structural entropy serves as a lower bound for the sample coverage.

We assume that the edge weights are in the range $[L, R]$, where $R > L > 0$. Consider a node $u$ connected to $k$ other nodes $\{v_1, v_2, \ldots, v_k\}$ with edge weights $\{w_1, w_2, \ldots, w_k\}$. In the BFS tree rooted at $u$, let the total $vol$ of the subtrees of $v_1, v_2 \ldots, v_k$ be $vol_1, vol_2, \ldots, vol_k$. From the perspective of $u$, the calculation of $S_e(u)$ is as follows: $vol(u)$ is initially set to 0; For each node $v_i$ connected to $u$, add $vol_i$ of $v_i$ to $vol(u)$, and then add $w_i vol(u)$ to $S_e(u)$. If the nodes are connected in the oreder $\{\pi_1, \pi_2, \ldots, \pi_k\}$, the results is:

$$
S_e(u) = \sum_{i=1}^{k} w_{\pi_i} \log(vol_{\pi_1} + vol_{\pi_2} + \ldots vol_{\pi_k}).
$$

In practice, the nodes are connected in a specific order to minimize $S_e(u)$. Consequently, $S_e(u)$ is less than or equal to values obtained in any other order. Consider the order in which nodes are connected from the smallest $w_i$ to the largest $w_i$. We derive that:

$$
\begin{aligned}
S_e(u) &\leq \sum_{i=1}^{k} w_{\pi_i} \log(vol_{\pi_1} + vol_{\pi_2} + ...vol_{\pi_k}) \\
&\leq \left( \sum_{i=1}^{k} w_i \right) \log \sum_{i=1}^{k} \left( \frac{w_{\pi_k}}{\sum_{i=1}^{k} w_i} (vol_{\pi_1} + ... + vol_{\pi_k}) \right) \text{ (Jensen's Inequality)} \\
&= \left( \sum_{i=1}^{k} w_i \right) \log \sum_{i=1}^{k} (vol_{\pi_k}(w_{\pi_1} + w_{\pi_2} + ...w_{\pi_k})/\sum_{i=1}^{k} w_i) \\
&\leq kR \log \sum_{i=1}^{k} (vol_{\pi_k}(w_{\pi_1} + w_{\pi_2} + ...w_{\pi_k})/\sum_{i=1}^{k} w_i) \\
&\leq kR \log \sum_{i=1}^{k} (kvol_{\pi_k} w_{\pi_k}/\sum_{i=1}^{k} w_i)
\end{aligned}
\tag{18}
$$

Because $vol_{\pi_k} \in [kLn_{\pi_k}, kRn_{\pi_k}]$, where $n_{\pi_k}$ is the number of nodes in the subtree of $v_{\pi_k}$. So we have:

$$
S_e(u) \leq kR \log(k^2 R \sum_{i=1}^{k} n_{\pi_i} w_{\pi_i}/\sum_{i=1}^{k} w_i)
$$

Assume that all nodes in the subtree of $v_i$ are in $B(v_i, r^*)$. Then we have:

$$
\mathbb{E}[n_{i_k}] = n \int_{B(v_i,r^*)} \mathrm{d}\mu = nP(v_i, r^*)
$$

which can then lead to:

$$
\begin{aligned}
\mathbb{E}[S_e(u)] &\leq \mathbb{E}[kR \log \left( k^2 R(\sum_{i=1}^{k} n_{\pi_i} w_{\pi_i}/\sum_{i=1}^{k} w_i) \right)] \\
&\leq kR \log \mathbb{E}\left[ k^2 R \left( \sum_{i=1}^{k} n_{\pi_i} w_{\pi_i}/\sum_{i=1}^{k} w_i \right) \right] \text{(Jensen's Inequality)} \\
&= kR \log \mathbb{E}\left[ nk^2 R \left( \sum P(v_{\pi_i}, r^*) w_{\pi_i}/\sum_{i=1}^{k} w_i \right) \right]
\end{aligned}
\tag{19}
$$

Since

$$
\sum_{i=1}^{k} P(v_{\pi_i}, r^*) w_{\pi_i}/\sum_{i=1}^{k} w_i = \hat{P}(u, r)
$$

is actually the Nadaraya–Watson estimator of $P(u, r^*)$, we have:

$$
\mathbb{E}[S_e(u)] \leq KR \log \mathbb{E}\left[ nK^2 R \hat{P}(u, r) \right]
$$

Thus, the following theorem is obtained:

**Theorem 1.** *For $r > r^*$, $S_e(u)$ ensures an lower-bound of the Nadaraya–Watson estimator $\hat{P}(u, r^*)$ of $P(u, r^*)$ with inequality:*

$$
\frac{1}{nK^2 R} \exp \left( \frac{1}{KR} \mathbb{E}[S_e(u)] \right) \leq \mathbb{E}[\hat{P}(u, r)] \approx \mathbb{E}[P(u, r)]
$$

*where $r^*$ is a constant related to the distribution $P_\mu$ and the location of $u$ in the metric space.*

Theorem 1 shows the correlation between $S_e(u)$ and $P(u, r)$. For a certain sample $u$, a larger $S_e(u)$ guarantees a stronger sample coverage.

We further conduct an empirical validation to demonstrate that $\frac{1}{nk^2R}\exp(\frac{1}{kR}S_e(u))$ provides a relatively tight lower bound for $P(u,r)$. To precisely calculate $P(u,r)$, we simulate the data using a Gaussian Mixture Model, where the class centers $\mu_1,\ldots,\mu_C$ are drawn from a standard Gaussian distribution $\mathcal{N}(0,I)$. For each class $c$, the samples are drawn from $\mathcal{N}(\mu_c,I)$. We align the synthetic data with real-world scales by generating 10 classes with 5,000 samples each, matching the statistics of CIFAR10. Our results show that: 1) For 90% of the samples, $P(u,r)/\left(\frac{1}{nk^2R}\exp(\frac{1}{kR}S_e(u))\right)$ lies between $1.00$ and $1.45$; 2) For 99% of the samples, $P(u,r)/\left(\frac{1}{nk^2R}\exp(\frac{1}{kR}S_e(u))\right)$ is less than $1.97$. These findings support the theorem that maximizing $S_e(u)$ during selection improves sample coverage.

### C.2 SAMPLE COVERAGE AND MODEL PERFORMANCE

Previous work (Zheng et al., 2022) has demonstrated that a better distribution coverage brings a lower empirical loss, as depicted by the following theorem:

**Theorem 2.** Zheng et al. (2022) *Given $n$ i.i.d. samples drawn from $P_\mu$ as $S = \{\boldsymbol{x_i}, y_i\}_{i\in[n]}$ where $\boldsymbol{x_i} \in X = \mathbb{R}^d$ is the feature representation of example $i$ and $y_i \in [C]$ is the class label for example $i$, a coreset $S'$ which is a $p$-partial $r$-cover for $P_\mu$ on the feature space $X$, and an $\epsilon > 1 - p$, if the loss function $l(\cdot, y, w)$ is $\lambda_l$ -Lipschitz continuous for all $y$, $w$ and bounded by $L$, the class-specific regression function $\eta_c(x) = p(y = c|\boldsymbol{x})$ is $\lambda_\eta$-Lipschitz for all $c$, and $l(\boldsymbol{x}, y; h_{S'}) = 0, \forall(\boldsymbol{x}, y) \in S'$, then with probability at least $1 - \epsilon$:*

$$\left| \frac{1}{n} \sum_{\boldsymbol{x},y\in S} l(x, y; h_{S'}) \right| \leq r(\lambda_l + \lambda_\eta LC) + L\sqrt{\frac{\log\frac{p}{p+\epsilon-1}}{2n}} \tag{20}$$

Theorem 2 suggests that better sample coverage results in a tighter bound on the empirical loss for the entire set. Given the relationship between $S_e(u)$ and sample coverage, selecting samples with high node-level structural entropy effectively enhances model performance.

## D DATASET STATISTICS AND DETAILED EXPERIMENTAL SETTING

### D.1 SUPERVISED LEARNING

**Image classification**. The CIFAR10 and CIFAR100 datasets each consist of $50,000$ images of $32\times32$ pixels for the training set, with an additional $10,000$ images for testing. CIFAR10 includes 10 distinct classes, while CIFAR100 includes 100 classes. The ImageNet-1K dataset includes $1,281,167$ images across $1,000$ real-world classes for training, along with $50,000$ images for validation. Following common practice (Maharana et al., 2024), we trained ResNet-18 for 200 epochs on CIFAR10 and CIFAR100, and ResNet-34 for 60 epochs on ImageNet-1K. The batch size is set to 64. We use an SGD optimizer with an initial learning rate of 0.1, momentum of 0.9, and weight decay of 0.0002. We use a cosine annealing learning rate scheduler with a minimum learning rate of 0.0001.

**Text classification**. The ANLI dataset is a natural language inference dataset created through multiple rounds of iterative human-and-model-in-the-loop adversarial procedures. We utilize the data from the final round, which consists of $100,459$ training samples and $1,200$ test samples. Following previous work (Maharana et al., 2024), we fine-tune the RoBERTa model for $10,000$ iterations with a batch size of 16. We use the SGD optimizer with an initial learning rate of 0.1, momentum of 0.9, and weight decay of 0.0005. We use a cosine annealing scheduler with a minimum learning rate of 0.0001.

The IMDB Review dataset contains $25,000$ movie reviews each in the training and test splits, with each review labeled by sentiment (positive/negative). Following previous work (Maharana et al., 2024), we randomly select $2,000$ samples from the original training set due to the excessive samples in it, and use the original test set for evaluation. We fine-tune the RoBERTa model for $500$ iterations with a batch size of 16. The optimizer and scheduler settings are the same as that for the ANLI dataset.

**Object detection**. We train the model using the combined trainval sets of PASCAL VOC 2007 and 2012, which include $16,551$ images and $40,058$ objects across 20 categories. The model is evaluated on the PASCAL VOC 2007 test set, which comprises $4,952$ images and $12,032$ objects. We train

SSD (Liu et al., 2016) with VGG-16 (Simonyan & Zisserman, 2015) backbone from scratch for 80 epochs with a batch size of 64. We use an SGD optimizer with a learning rate of 0.001, momentum of 0.9, and weight decay of 0.0005. The learning rate follows a linear warm-up strategy for the first 8 epochs and is then reduced by a factor of 10 at epochs 50 and 70.

**Visual question answering**. The CC SBU Align dataset contains $3,439$ high-quality, aligned image-text pairs for the fine-tuning stage (stage 2) of Mini-GPT4 (Zhu et al., 2024). The visual question answering datasets used for validation test the model's abilities in various aspects, including logical reasoning, visual reasoning, knowledge retention, and abstract understanding. We use the same setting as the second-stage fine-tuning for Mini-GPT4 (Zhu et al., 2024). We fine-tune MiniGPT-4 for 400 iterations with a batch size of 12. We use an SGD optimizer with an initial learning rate of 0.00003, momentum of 0.9, and weight decay of 0.05. We use a cosine annealing scheduler with a minimum learning rate of 0.00001.

## D.2 ACTIVE LEARNING

We use the same setting as training on ImageNet-1K in Appendix D.1.

## D.3 CONTINUAL LEARNING

For continual learning, we follow Borsos et al. (2020) for experiments on Permuted MNIST, Split MNIST, and Split CIFAR10, and follow Hao et al. (2024) for experiments on Split CIFAR100 and Split Tiny-ImageNet. We use increasingly complex models as dataset complexity grows: a two-layer MLP for Permuted MNIST, a four-layer CNN for Split MNIST, ResNet-18 for Split CIFAR10, and ResNet-18 with multi-head output (Zenke et al., 2017) for Split CIFAR100 and Split Tiny-ImageNet. For each task, we first randomly select $\mathcal{M}$ samples from all available samples. Then, we train the model on these samples for $E$ epochs with a batch size of $\mathcal{B}$. During each training iteration, all replay samples are used with a weight of $\lambda$. The initial learning rate is set to $lr_t$ for the first task and decays by a factor of $\eta$ for each task. All hyperparameters, except $\lambda$, are fixed for all baselines and our method. The value of $\lambda$ is determined through a grid search over $\{0.01, 0.1, 1, 10, 100, 1000\}$. Table 6 shows the fixed hyperparameters for different datasets. Table 7 shows the optimal $\lambda$ for all baselines and our method.

## E SELECTION HYPERPARAMETER SETTINGS

### E.1 SELECTION HYPERPARAMETERS

We conduct a grid search to optimize three hyperparameters: the number of neighbors $k$ for constructing the $k$NN graph, the cutoff ratio $\beta$ to remove the most difficult samples, and the imbalance factor $\gamma$ to maintain the balance between different classes.

Selecting an appropriate value of $k$ is crucial as it can significantly affect the quality of the sample graph and thus affect the selection result. If $k$ is too small, the graph becomes too sparse, making the selection sensitive to noise and outliers. If $k$ is too large, the graph will be too dense and contain many edges connecting irrelevant neighbors, making it hard to identify the most important samples in the dataset. Thus, the choice of $k$ must strike a balance to preserve meaningful structures without introducing noise or irrelevant information.

Table 6: Training hyperparameters for continual learning.

| Hyperparameters | Permuted MNIST | Split MNIST | Split CIFAR10 | Split CIFAR100 | Split Tiny-ImageNet |
|---|---|---|---|---|---|
| $\mathcal{M}$ | 1000 | 1000 | 1000 | 2500 | 5000 |
| Optimizer | Adam | Adam | Adam | SGD | SGD |
| $E$ | 400 | 400 | 400 | 1 | 1 |
| $\mathcal{B}$ | 256 | 256 | 256 | 10 | 20 |
| $lr_t$ | 0.0005 | 0.0005 | 0.0005 | 0.15 | 0.20 |
| $\eta$ | 1 | 1 | 1 | 0.875 | 0.875 |

Inspired by Zheng et al. (2022), we also search the hard cutoff ratio $\beta$ that removes $\beta$ of the most difficult samples because they are usually outliers and contain noisy samples in the dataset. In addition, we also allow negative $\beta$ during the grid search, which indicates that we will remove $|\beta|$ of the easiest samples to focus on difficult samples, which has been adopted by Sorscher et al. (2022).

Maintaining a balanced distribution of samples from different classes is also beneficial for model training. A common strategy is to enforce strict class balance (Guo et al., 2022) by selecting an equal number of samples from each class. However, this overlooks the difference between classes, where some may be more easily confused with others and require more samples to distinguish. To address this, we introduce an imbalance factor $\gamma > 1$, which allows for the selection of up to $n\dot{\gamma}$ samples per class, rather than a strict count of $n$. This adjustment provides flexibility in addressing class-specific complexities. In the active learning scenario, we treat the $k$-means clusters as the classes for balancing.

### E.2 PRELIMINARY EXPERIMENTS ON k

As $k$ decides the sample graph and the range of possible $k$ is large, we conduct preliminary experiments on CIFAR10, CIFAR100, and ImageNet-1k to determine a candidate range where the structure of the sample graph is effectively preserved by the $k$NN-graph.

Spectral clustering is an effective method for revealing the underlying structural features of a graph. Therefore, we use the spectral clustering results to evaluate the choice of $k$ in the preliminary experiments. First, we perform spectral clustering on the $k$NN-graph. Based on the spectral clustering results, we measure the structure preservation of the $k$NN-graph using both external and internal metrics. The external metrics measure the consistency of the clustering results with the ground truth labels of samples. We use the accuracy, the Rand Index, and the mutual information as the external metrics. The internal metrics assess the quality of clustering by evaluating the compactness and separation of clusters. We use the Silhouette Score and the Davis-Bouldin Index as the internal metrics.

Directly applying spectral clustering to large-scale datasets is unsuitable due to its high complexity of $O(n^3)$, where $n$ is the number of samples in the dataset. To address this, we use growing neural gas (Fritzke, 1995) to generate a reduced graph with fewer nodes. The key feature of this method is the preservation of the topology of the $k$NN-graph. Specifically, this method generates neurons representing the sample distribution of the $k$NN-graph. Then, it integrates the edges in the $k$NN-graph into the neuron connections, resulting in a sparse, topology-preserving graph. We apply spectral clustering to this reduced graph to accelerate the evaluation of the choice of $k$.

In the preliminary experiments, we test $k$ in $\{1, 2, \ldots, 100\}$ on CIFAR10, CIFAR100, and ImageNet-1K. Fig. 4 presents the results. As the value of $k$ increases, the performance metrics initially improve, then stabilize, and may eventually decline. This suggests that the first few nearest neighbors effectively capture the structure of the original graph, while including too many neighbors may lead to underfitting due to edges connecting vastly different nodes. We discover that the metrics begin to

Table 7: Optimal replay sample weight $\lambda$ in continual learning.

| Dataset | Permuted MNIST | | Split MNIST | | Split CIFAR10 | | Split CIFAR100 | | Split Tiny-ImageNet | |
|---|---|---|---|---|---|---|---|---|---|---|
| Memory size | 100 | 200 | 100 | 200 | 100 | 200 | 100 | 200 | 100 | 200 |
| Random | 0.01 | 0.01 | 100 | 100 | 0.01 | 0.01 | 0.01 | 0.01 | 0.01 | 0.01 |
| Moderate | 0.01 | 0.01 | 100 | 100 | 0.1 | 0.1 | 0.01 | 0.01 | 0.01 | 0.01 |
| CCS | 0.01 | 0.01 | 1000 | 100 | 0.01 | 0.01 | 0.01 | 0.01 | 0.01 | 0.01 |
| $\mathbb{D}^2$ Pruning | 0.01 | 0.01 | 10 | 100 | 0.1 | 0.01 | 0.01 | 0.01 | 0.01 | 0.01 |
| GraphCut | 0.01 | 0.01 | 100 | 1 | 0.01 | 0.1 | 0.01 | 0.1 | 1 | 0.1 |
| Entropy | 0.01 | 0.01 | 1000 | 100 | 0.01 | 0.01 | 0.01 | 0.01 | 0.01 | 0.01 |
| Forgetting | 0.01 | 0.01 | 100 | 10 | 0.1 | 0.1 | 0.01 | 0.01 | 0.01 | 0.1 |
| EL2N | 0.01 | 0.01 | 100 | 100 | 0.01 | 0.1 | 0.01 | 0.01 | 0.01 | 0.01 |
| AUM | 0.01 | 0.01 | 100 | 100 | 10 | 1 | 0.01 | 0.01 | 0.01 | 0.01 |
| Variance | 0.01 | 0.01 | 1000 | 1000 | 0.01 | 0.01 | 0.01 | 0.01 | 0.01 | 0.01 |
| iCaRL | 0.01 | 0.1 | 100 | 100 | 1 | 1 | 0.01 | 0.01 | 0.1 | 0.1 |
| Greedy Coreset | 0.01 | 0.01 | 100 | 10 | 0.01 | 0.01 | 0.01 | 0.1 | 0.01 | 0.01 |
| BCSR | 0.01 | 0.01 | 100 | 10 | 0.01 | 0.01 | 0.01 | 0.01 | 0.01 | 0.01 |
| SES (Ours) | 0.01 | 0.1 | 100 | 100 | 1 | 1 | 0.01 | 0.1 | 0.01 | 0.1 |

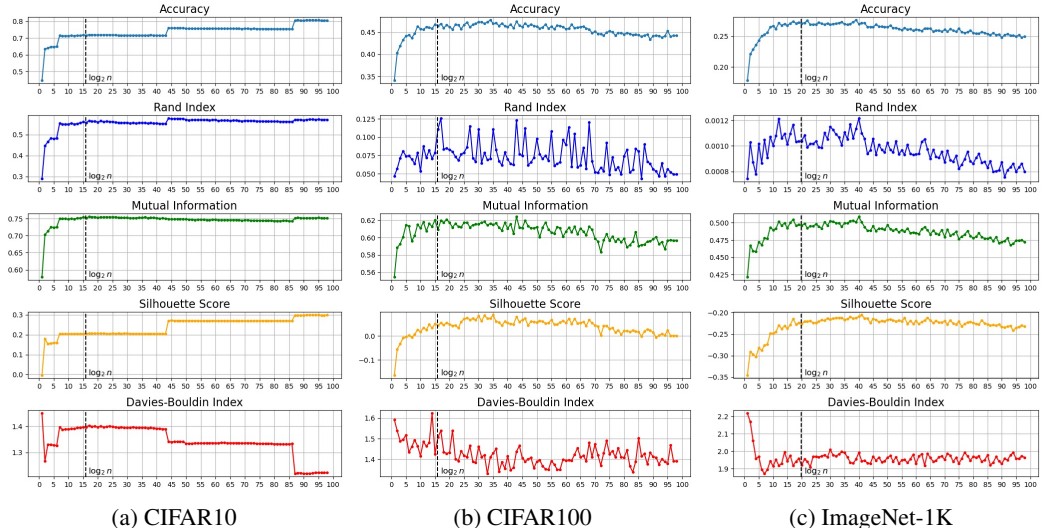

Figure 4: Results for the preliminary experiments on: (a) CIFAR10; (b) CIFAR100; (c) ImageNet-1K. An increase in the accuracy, the Rand Index, the mutual information, and the Silhouette Score, as well as a decrease in the Davis-Bouldin Index, indicates better structure preservation.

Table 8: Optimal hyperparameters of our method in supervised learning and active learning. The three numbers in each tuple refer to the number of neighbors $k$, the cutoff ratio $\beta$, and the imbalance factor $\gamma$, respectively.

| Sampling rate | CIFAR10 | CIFAR100 | ImageNet-1K | ANLI | IMDB Review | PASCAL VOC | VQA | ImageNet-1K (active learning) |
|---|---|---|---|---|---|---|---|---|
| 1% | (17,0.75,1.05) | (14,0.85,1.2) | (20,0.9,1.3) | (15,0.35,1.05) | (10,0.7,1.2) | (15,0.7,/) | (10,0.15,/) | (25,0.25,1.05) |
| 2% | (14,0.7,1.1) | (10,0.85,1.15) | (25,0.75,1.2) | (15,0.3,1.05) | (10,0.85,1.1) | (15,0.7,/) | (10,0.1,/) | (25,0.2,1.45) |
| 5% | (19,0.6,1.05) | (21,0.8,1.2) | (25,0.65,1.2) | (15,0.25,1.05) | (10,0.5,1.0) | (15,0.5,/) | (10,0.1,/) | (25,0.4,1.35) |
| 10% | (11,0.35,1.1) | (19,0.6,1.1) | (25,0.4,1.15) | (15,0.3,1.15) | (10,0.4,1.05) | (15,0.35,/) | (10,0.15,/) | (25,0.45,1.05) |
| 20% | (13,0.2,1.0) | (15,0.45,1.15) | (25,0.3,1.1) | (15,0.0,1.05) | (10,0.3,1.1) | (15,0.1,/) | (10,0.15,/) | (25,0.35,1.15) |
| 50% | (15,-0.15,1.0) | (15,0.15,1.0) | (20,0.05,1.05) | (15,0.0,1.1) | (10,0.05,1.05) | (15,0.15,/) | (10,0.1,/) | (25,0.0,1.1) |
| 70% | (24,0.0,1.0) | (18,0.1,1.0) | (20,0.0,1.0) | (15,0.0,1.2) | (10,0.05,1.0) | (15,0.05,/) | (10,0.0,/) | (25,0.0,1.15) |

stabilize around $\log_2 n$, where $n$ is the number of samples in the dataset. Thus, we focus our grid search for $k$ values near $\log_2 n$.

### E.3 GRID SEARCH RESULTS

For all experiments, we search $\beta$ in $\{-1, -0.95, \ldots, -0.05, 0, 0.05, \ldots, 0.95, 1\}$ and $\gamma$ in $\{1, 1.05, \ldots, 1.5\}$. For CIFAR10 and CIFAR100, we search $k$ in $\{10, 11, \ldots, 30\}$. For ImageNet-1K, we search it only in $\{20, 25\}$ due to computational costs. To save computational costs, we set $k$ to around $\log_2 n$ for experiments on other datasets. With this choice, the $k$NN graph has $O(n \log_2 n)$ edges, resulting in a Shapley value calculation time complexity of $O(n \log_2 n)$. This complexity is comparable to sorting and is lower than or equal to that of typical sample selection methods, which is at least $O(n \log_2 n)$ when sorting is required and $O(n^2)$ when an enumeration of all pairs of samples is required. For extremely large datasets (e.g. DataComp (Gadre et al., 2024) with billions of samples) where this complexity becomes impractical, a potential solution is to split the dataset into bins before selection. Determining an optimal splitting method remains an open question and a promising direction for future research. We evaluate every possible combination of the hyperparameters. Table 8 presents the optimal hyperparameters for supervised learning and active learning. Table 9 presents the optimal hyperparameters for continual learning.

Table 9: Optimal hyperparameters of our method in continual learning. The three numbers in each tuple refer to the number of neighbors $k$, the cutoff ratio $\beta$, and the imbalance factor $\gamma$, respectively.

| Memory size | Permuted MNIST | Split MNIST | Split CIFAR10 | Split CIFAR100 | Split Tiny-ImageNet |
|---|---|---|---|---|---|
| 100 | (15,-0.15,1.25) | (15,0.7,1.5) | (15,-0.35,1.0) | (15,-0.95,1.0) | (15,-0.15,1.0) |
| 200 | (15,-0.45,1.5) | (15,0.75,1.1) | (15,-0.7,1.1) | (15,-0.85,1.0) | (15,-0.9,1.0) |

# F  DETAILED EXPERIMENTAL RESULTS

## F.1  SUPERIVISED LEARNING

We present detailed experimental results for supervised learning from Tables 10 to 16. Our method consistently performs better than baselines across all tasks and sampling rates. Meanwhile, several factors prevent us from achieving more significant improvements over the baselines in certain settings:

- In high-sampling-rate settings for simple datasets, such as the 70% setting for CIFAR10, all methods perform closely to using the entire training set, leaving little room for improvement.

- In low-sampling-rate settings for challenging datasets, such as the 1% setting for ImageNet-1K, the performance of all methods is limited due to the insufficient number of samples per class.

- In scenarios where models are fine-tuned from pretrained weights, the knowledge within these models anchors the performance at a certain level, leading to similar performance across all methods.

Despite these factors, our method achieves significant improvement in many settings. For example, it achieves a 5.52% increase in accuracy when selecting 2% of the samples from ImageNet-1K.

Table 10: Results on image classification (CIFAR10). The best one is **bold**, and the runner-up is underlined.

| Dataset Sampling rate | CIFAR10 (100%:95.49) | | | | | | |
|---|---|---|---|---|---|---|---|
| | 70% | 50% | 20% | 10% | 5% | 2% | 1% |
| **Random** | $94.29 \pm 0.08$ | $92.33 \pm 0.15$ | $84.17 \pm 0.62$ | $71.95 \pm 1.50$ | $59.88 \pm 2.45$ | $45.51 \pm 1.68$ | $35.44 \pm 1.49$ |
| **Moderate** | $94.04 \pm 0.26$ | $92.36 \pm 0.17$ | $80.96 \pm 2.56$ | $63.91 \pm 1.56$ | $49.94 \pm 0.70$ | $32.00 \pm 2.11$ | $26.18 \pm 1.94$ |
| **CCS** | $94.23 \pm 0.35$ | $93.83 \pm 0.15$ | $83.85 \pm 1.10$ | $75.56 \pm 2.39$ | $\underline{65.89 \pm 1.04}$ | $\underline{49.02 \pm 1.35}$ | $\underline{40.24 \pm 0.96}$ |
| $\mathbb{D}^2$ **Pruning** | $93.17 \pm 0.11$ | $93.70 \pm 0.20$ | $\underline{86.83 \pm 0.48}$ | $\underline{76.56 \pm 1.14}$ | $65.12 \pm 0.88$ | $45.44 \pm 1.75$ | $39.03 \pm 1.69$ |
| **GraphCut** | $94.38 \pm 0.20$ | $92.43 \pm 0.23$ | $81.98 \pm 0.48$ | $69.29 \pm 0.57$ | $59.36 \pm 2.43$ | $46.93 \pm 2.21$ | $39.68 \pm 1.15$ |
| **Entropy** | $94.13 \pm 0.30$ | $92.39 \pm 0.40$ | $75.34 \pm 4.32$ | $60.66 \pm 6.23$ | $46.17 \pm 5.96$ | $35.85 \pm 4.33$ | $29.28 \pm 4.29$ |
| **Forgetting** | $94.76 \pm 0.38$ | $94.34 \pm 0.17$ | $57.72 \pm 1.17$ | $35.21 \pm 0.87$ | $30.87 \pm 0.57$ | $27.22 \pm 0.49$ | $23.62 \pm 0.74$ |
| **EL2N** | $94.75 \pm 0.32$ | $94.10 \pm 0.45$ | $46.41 \pm 4.84$ | $22.40 \pm 0.62$ | $15.66 \pm 0.18$ | $13.05 \pm 0.20$ | $12.75 \pm 0.56$ |
| **AUM** | $\underline{94.90 \pm 0.17}$ | $\underline{94.38 \pm 0.20}$ | $51.98 \pm 1.32$ | $30.56 \pm 0.42$ | $22.92 \pm 0.38$ | $18.10 \pm 0.47$ | $14.50 \pm 1.35$ |
| **Variance** | $90.45 \pm 0.09$ | $85.81 \pm 0.39$ | $64.90 \pm 0.53$ | $53.64 \pm 1.02$ | $44.45 \pm 0.77$ | $36.61 \pm 0.70$ | $31.46 \pm 0.92$ |
| $k$**-means** | $93.87 \pm 0.33$ | $93.19 \pm 0.36$ | $83.68 \pm 0.94$ | $72.46 \pm 0.51$ | $62.68 \pm 1.12$ | $48.35 \pm 2.29$ | $38.73 \pm 0.87$ |
| $k$**-DPP** | $93.84 \pm 0.10$ | $92.72 \pm 0.18$ | $84.09 \pm 1.21$ | $74.62 \pm 0.40$ | $61.95 \pm 4.87$ | $46.58 \pm 1.76$ | $35.16 \pm 2.92$ |
| **SES (Ours)** | $\mathbf{95.01 \pm 0.08}$ | $\mathbf{94.50 \pm 0.05}$ | $\mathbf{88.31 \pm 0.13}$ | $\mathbf{80.24 \pm 0.72}$ | $\mathbf{69.82 \pm 0.84}$ | $\mathbf{54.78 \pm 1.36}$ | $\mathbf{45.25 \pm 0.81}$ |

Table 11: Results on image classification (CIFAR100). The best one is **bold**, and the runner-up is underlined.

| Dataset Sampling rate | CIFAR100 (100%:77.90) | | | | | | |
|---|---|---|---|---|---|---|---|
| | 70% | 50% | 20% | 10% | 5% | 2% | 1% |
| **Random** | $74.72 \pm 0.16$ | $70.51 \pm 0.43$ | $52.29 \pm 2.32$ | $37.38 \pm 0.80$ | $23.03 \pm 0.48$ | $13.38 \pm 0.36$ | $8.57 \pm 0.65$ |
| **Moderate** | $74.61 \pm 0.10$ | $69.95 \pm 0.32$ | $49.37 \pm 0.58$ | $30.53 \pm 1.44$ | $17.68 \pm 0.50$ | $9.28 \pm 0.34$ | $5.97 \pm 0.35$ |
| **CCS** | $76.10 \pm 0.29$ | $73.12 \pm 0.21$ | $58.39 \pm 1.04$ | $\underline{44.92 \pm 1.81}$ | $\underline{28.68 \pm 0.99}$ | $\underline{15.93 \pm 0.62}$ | $\underline{10.28 \pm 0.54}$ |
| $\mathbb{D}^2$ **Pruning** | $76.00 \pm 0.56$ | $\underline{74.45 \pm 0.36}$ | $\underline{59.08 \pm 0.53}$ | $44.78 \pm 0.78$ | $26.68 \pm 0.85$ | $13.90 \pm 0.66$ | $9.61 \pm 0.44$ |
| **GraphCut** | $76.64 \pm 0.13$ | $68.79 \pm 0.35$ | $25.08 \pm 1.10$ | $16.46 \pm 0.35$ | $10.78 \pm 0.52$ | $7.84 \pm 0.35$ | $5.94 \pm 0.16$ |
| **Entropy** | $76.81 \pm 0.12$ | $62.52 \pm 0.26$ | $33.28 \pm 7.01$ | $21.85 \pm 7.11$ | $13.90 \pm 4.75$ | $8.31 \pm 2.69$ | $5.30 \pm 1.71$ |
| **Forgetting** | $76.64 \pm 0.13$ | $68.75 \pm 0.34$ | $25.17 \pm 1.12$ | $16.45 \pm 0.37$ | $10.70 \pm 0.49$ | $7.82 \pm 0.37$ | $5.90 \pm 0.09$ |
| **EL2N** | $76.15 \pm 0.50$ | $66.19 \pm 0.48$ | $14.85 \pm 0.43$ | $7.50 \pm 0.27$ | $5.28 \pm 0.12$ | $3.83 \pm 0.07$ | $3.17 \pm 0.14$ |
| **AUM** | $\underline{76.84 \pm 0.40}$ | $67.81 \pm 0.59$ | $16.66 \pm 0.90$ | $8.54 \pm 0.10$ | $5.60 \pm 0.18$ | $4.35 \pm 0.17$ | $3.56 \pm 0.10$ |
| **Variance** | $73.69 \pm 0.14$ | $68.42 \pm 0.16$ | $49.60 \pm 0.28$ | $34.63 \pm 0.42$ | $22.70 \pm 0.35$ | $13.71 \pm 0.38$ | $9.18 \pm 0.52$ |
| $k$-**means** | $74.93 \pm 0.12$ | $72.86 \pm 0.17$ | $56.86 \pm 0.88$ | $42.69 \pm 0.52$ | $26.39 \pm 0.73$ | $14.92 \pm 0.75$ | $8.90 \pm 0.31$ |
| $k$-**DPP** | $75.34 \pm 0.45$ | $73.16 \pm 0.45$ | $56.88 \pm 1.01$ | $42.09 \pm 0.41$ | $26.55 \pm 0.9$ | $14.31 \pm 0.33$ | $8.23 \pm 0.27$ |
| **SES (Ours)** | $\mathbf{77.23 \pm 0.10}$ | $\mathbf{74.63 \pm 0.17}$ | $\mathbf{61.52 \pm 0.76}$ | $\mathbf{48.02 \pm 0.98}$ | $\mathbf{33.39 \pm 0.27}$ | $\mathbf{19.68 \pm 0.51}$ | $\mathbf{14.16 \pm 0.48}$ |

Table 12: Results on image classification (ImageNet-1K). The best one is **bold**, and the runner-up is underlined.

| Dataset Sampling rate | ImageNet-1K (100%:73.63) | | | | | | |
|---|---|---|---|---|---|---|---|
| | 70% | 50% | 20% | 10% | 5% | 2% | 1% |
| **Random** | $71.63 \pm 0.12$ | $69.26 \pm 0.17$ | $58.90 \pm 0.12$ | $47.10 \pm 0.26$ | $34.04 \pm 0.42$ | $16.56 \pm 0.11$ | $5.50 \pm 0.28$ |
| **Moderate** | $71.33 \pm 0.04$ | $68.72 \pm 0.07$ | $55.23 \pm 0.27$ | $40.97 \pm 0.18$ | $25.75 \pm 0.16$ | $11.33 \pm 0.31$ | $4.52 \pm 0.24$ |
| **CCS** | $70.74 \pm 0.09$ | $69.23 \pm 0.09$ | $60.04 \pm 0.95$ | $50.41 \pm 0.11$ | $36.92 \pm 1.62$ | $19.92 \pm 1.36$ | $9.43 \pm 0.31$ |
| $\mathbb{D}^2$ **Pruning** | $71.29 \pm 0.07$ | $70.32 \pm 0.05$ | $58.91 \pm 0.04$ | $\underline{50.81 \pm 0.24}$ | $\underline{37.12 \pm 0.09}$ | $18.97 \pm 0.05$ | $11.23 \pm 0.26$ |
| **GraphCut** | $68.91 \pm 0.05$ | $68.72 \pm 0.07$ | $55.28 \pm 0.17$ | $44.79 \pm 0.06$ | $33.54 \pm 0.12$ | $\underline{20.07 \pm 0.21}$ | $\underline{11.49 \pm 0.12}$ |
| **Entropy** | $70.93 \pm 0.74$ | $69.21 \pm 0.12$ | $54.76 \pm 0.16$ | $38.46 \pm 0.34$ | $22.78 \pm 0.73$ | $7.01 \pm 0.24$ | $1.95 \pm 0.15$ |
| **Forgetting** | $70.57 \pm 0.04$ | $\underline{70.46 \pm 0.10}$ | $\underline{60.77 \pm 0.02}$ | $48.73 \pm 0.12$ | $33.86 \pm 0.15$ | $15.13 \pm 0.23$ | $5.66 \pm 0.30$ |
| **EL2N** | $\underline{71.68 \pm 0.04}$ | $65.98 \pm 0.12$ | $31.90 \pm 3.64$ | $12.57 \pm 0.11$ | $6.50 \pm 0.36$ | $3.25 \pm 0.13$ | $1.90 \pm 0.09$ |
| **AUM** | $69.94 \pm 0.30$ | $65.36 \pm 0.11$ | $21.91 \pm 0.13$ | $10.50 \pm 0.19$ | $6.42 \pm 0.07$ | $3.58 \pm 0.07$ | $2.24 \pm 0.05$ |
| **Variance** | $70.12 \pm 0.09$ | $66.09 \pm 0.01$ | $35.15 \pm 0.09$ | $13.85 \pm 0.02$ | $7.13 \pm 0.05$ | $4.72 \pm 0.02$ | $1.81 \pm 0.06$ |
| $k$-**means** | $70.33 \pm 0.07$ | $69.47 \pm 0.11$ | $59.23 \pm 0.22$ | $48.12 \pm 0.21$ | $35.51 \pm 0.13$ | $18.67 \pm 0.08$ | $9.65 \pm 0.33$ |
| $k$-**DPP** | $70.84 \pm 0.09$ | $69.85 \pm 0.14$ | $59.92 \pm 0.33$ | $46.10 \pm 0.33$ | $34.41 \pm 0.07$ | $16.33 \pm 0.15$ | $7.49 \pm 0.17$ |
| **SES (Ours)** | $\mathbf{72.80 \pm 0.03}$ | $\mathbf{71.05 \pm 0.09}$ | $\mathbf{63.24 \pm 0.06}$ | $\mathbf{53.59 \pm 0.09}$ | $\mathbf{41.88 \pm 0.13}$ | $\mathbf{25.59 \pm 0.17}$ | $\mathbf{13.43 \pm 0.37}$ |

Table 13: Results on text classification (ANLI). The best one is **bold**, and the runner-up is underlined.

| Dataset Sampling rate | ANLI (100%:49.25) | | | | | | |
|---|---|---|---|---|---|---|---|
| | 70% | 50% | 20% | 10% | 5% | 2% | 1% |
| **Random** | $47.08 \pm 0.26$ | $45.20 \pm 0.50$ | $42.13 \pm 0.54$ | $39.52 \pm 0.93$ | $38.82 \pm 1.06$ | $37.50 \pm 1.29$ | $35.96 \pm 1.04$ |
| **Moderate** | $46.84 \pm 0.31$ | $45.11 \pm 0.39$ | $41.95 \pm 0.33$ | $40.16 \pm 0.85$ | $38.99 \pm 0.45$ | $35.83 \pm 1.39$ | $33.91 \pm 0.62$ |
| **CCS** | $46.56 \pm 0.23$ | $45.92 \pm 0.70$ | $41.67 \pm 1.39$ | $\underline{41.63 \pm 0.97}$ | $40.33 \pm 0.65$ | $37.41 \pm 0.05$ | $\underline{36.82 \pm 0.49}$ |
| $\mathbb{D}^2$ **Pruning** | $48.56 \pm 1.22$ | $47.49 \pm 0.17$ | $42.77 \pm 0.36$ | $41.43 \pm 0.24$ | $\underline{40.34 \pm 0.07}$ | $\underline{37.92 \pm 1.44}$ | $36.29 \pm 0.59$ |
| **GraphCut** | $46.14 \pm 0.67$ | $44.53 \pm 0.55$ | $42.12 \pm 0.66$ | $39.86 \pm 0.27$ | $38.15 \pm 0.74$ | $35.44 \pm 0.80$ | $34.02 \pm 0.61$ |
| **Entropy** | $46.32 \pm 1.11$ | $45.53 \pm 0.44$ | $41.45 \pm 0.33$ | $39.67 \pm 0.28$ | $38.54 \pm 0.66$ | $36.69 \pm 0.53$ | $36.40 \pm 0.19$ |
| **Forgetting** | $\underline{48.73 \pm 0.35}$ | $42.29 \pm 0.17$ | $39.82 \pm 0.16$ | $38.37 \pm 0.60$ | $35.95 \pm 0.33$ | $35.78 \pm 0.42$ | $35.03 \pm 0.77$ |
| **EL2N** | $48.70 \pm 0.65$ | $47.85 \pm 0.98$ | $43.14 \pm 1.68$ | $39.63 \pm 2.20$ | $37.52 \pm 1.05$ | $34.33 \pm 1.15$ | $34.27 \pm 0.42$ |
| **AUM** | $47.86 \pm 0.27$ | $47.58 \pm 0.37$ | $\underline{43.57 \pm 0.71}$ | $40.02 \pm 1.51$ | $34.66 \pm 0.73$ | $34.16 \pm 0.12$ | $33.62 \pm 0.12$ |
| **Variance** | $47.97 \pm 0.30$ | $\underline{47.87 \pm 0.56}$ | $40.70 \pm 0.16$ | $38.75 \pm 0.62$ | $33.52 \pm 0.03$ | $33.50 \pm 0.03$ | $33.17 \pm 0.12$ |
| $k$-**means** | $46.48 \pm 0.24$ | $46.52 \pm 1.06$ | $42.42 \pm 0.72$ | $40.57 \pm 0.65$ | $39.89 \pm 0.10$ | $36.74 \pm 0.29$ | $36.11 \pm 0.40$ |
| $k$-**DPP** | $47.74 \pm 0.07$ | $47.02 \pm 0.02$ | $43.44 \pm 0.59$ | $40.98 \pm 0.46$ | $40.12 \pm 0.26$ | $37.44 \pm 0.11$ | $36.66 \pm 0.23$ |
| **SES (Ours)** | $\mathbf{49.00 \pm 0.08}$ | $\mathbf{48.22 \pm 0.54}$ | $\mathbf{45.94 \pm 0.27}$ | $\mathbf{43.63 \pm 1.00}$ | $\mathbf{41.82 \pm 0.43}$ | $\mathbf{39.88 \pm 0.7}$ | $\mathbf{38.16 \pm 0.38}$ |

Table 14: Results on text classification (IMDB). The best one is **bold**, and the runner-up is underlined.

| Dataset
Sampling rate | IMDB Review(100%:95.90) | | | | | | |
|---|---|---|---|---|---|---|---|
| | **70%** | **50%** | **20%** | **10%** | **5%** | **2%** | **1%** |
| **Random** | $95.25 \pm 0.21$ | $95.04 \pm 0.31$ | $93.53 \pm 0.68$ | $91.89 \pm 0.17$ | $89.60 \pm 1.89$ | $83.25 \pm 4.96$ | $73.31 \pm 1.94$ |
| **Moderate** | $95.39 \pm 0.19$ | $95.31 \pm 0.07$ | $\underline{94.03 \pm 0.73}$ | $92.55 \pm 0.47$ | $90.34 \pm 0.43$ | $58.34 \pm 4.29$ | $50.81 \pm 0.55$ |
| **CCS** | $95.22 \pm 0.18$ | $95.35 \pm 0.19$ | $93.44 \pm 0.61$ | $91.87 \pm 0.23$ | $89.89 \pm 1.47$ | $85.73 \pm 4.76$ | $82.05 \pm 4.21$ |
| $\mathbb{D}^2$ **Pruning** | $95.43 \pm 0.16$ | $\underline{95.40 \pm 0.20}$ | $93.75 \pm 0.46$ | $92.44 \pm 0.60$ | $90.77 \pm 0.33$ | $\underline{87.40 \pm 0.78}$ | $80.06 \pm 0.85$ |
| **GraphCut** | $95.39 \pm 0.15$ | $95.21 \pm 0.29$ | $93.37 \pm 0.57$ | $91.69 \pm 0.66$ | $\underline{90.93 \pm 0.31}$ | $86.85 \pm 1.81$ | $\underline{82.26 \pm 0.90}$ |
| **Entropy** | $95.39 \pm 0.31$ | $95.17 \pm 0.22$ | $93.92 \pm 0.50$ | $\underline{92.57 \pm 0.97}$ | $90.30 \pm 0.63$ | $77.16 \pm 4.52$ | $59.41 \pm 4.70$ |
| **Forgetting** | $95.41 \pm 0.11$ | $95.31 \pm 0.17$ | $93.66 \pm 0.30$ | $89.41 \pm 0.58$ | $58.78 \pm 2.28$ | $55.81 \pm 4.65$ | $52.38 \pm 1.03$ |
| **EL2N** | $95.29 \pm 0.19$ | $95.34 \pm 0.21$ | $91.31 \pm 0.19$ | $60.29 \pm 0.09$ | $49.88 \pm 1.35$ | $47.18 \pm 3.30$ | $43.74 \pm 0.81$ |
| **AUM** | $95.27 \pm 0.66$ | $95.23 \pm 0.16$ | $90.60 \pm 0.13$ | $55.68 \pm 0.21$ | $49.81 \pm 1.50$ | $43.13 \pm 4.53$ | $36.29 \pm 1.45$ |
| **Variance** | $\underline{95.44 \pm 0.16}$ | $\underline{95.40 \pm 0.20}$ | $93.75 \pm 0.46$ | $92.44 \pm 0.60$ | $90.77 \pm 0.33$ | $87.40 \pm 0.78$ | $80.06 \pm 0.85$ |
| $k$-**means** | $95.34 \pm 0.17$ | $95.22 \pm 0.23$ | $93.98 \pm 0.54$ | $92.49 \pm 0.74$ | $90.21 \pm 0.88$ | $86.96 \pm 1.84$ | $81.51 \pm 6.72$ |
| $k$-**DPP** | $95.29 \pm 0.37$ | $95.13 \pm 0.21$ | $93.76 \pm 0.59$ | $92.50 \pm 0.58$ | $89.97 \pm 1.42$ | $86.14 \pm 2.53$ | $77.07 \pm 3.31$ |
| **SES (Ours)** | $\mathbf{95.60 \pm 0.16}$ | $\mathbf{95.40 \pm 0.12}$ | $\mathbf{94.57 \pm 0.09}$ | $\mathbf{92.96 \pm 0.56}$ | $\mathbf{91.42 \pm 0.24}$ | $\mathbf{88.58 \pm 0.47}$ | $\mathbf{83.14 \pm 2.05}$ |

Table 15: Results on object detection. The best one is **bold**, and the runner-up is underlined.

| Dataset
Sampling rate | PASCAL VOC (100%:76.29) | | | | | | |
|---|---|---|---|---|---|---|---|
| | **70%** | **50%** | **20%** | **10%** | **5%** | **2%** | **1%** |
| **Random** | $74.02 \pm 0.26$ | $72.10 \pm 0.32$ | $65.45 \pm 0.45$ | $\underline{57.56 \pm 0.45}$ | $43.47 \pm 0.67$ | $18.78 \pm 0.21$ | $9.24 \pm 0.96$ |
| **Moderate** | $73.42 \pm 0.46$ | $72.03 \pm 0.52$ | $65.12 \pm 0.57$ | $54.71 \pm 0.10$ | $43.35 \pm 0.60$ | $15.97 \pm 0.36$ | $5.13 \pm 0.49$ |
| **CCS** | $\underline{74.64 \pm 0.15}$ | $72.27 \pm 0.23$ | $\underline{65.72 \pm 0.59}$ | $57.35 \pm 0.63$ | $39.01 \pm 0.81$ | $17.26 \pm 0.62$ | $8.49 \pm 0.57$ |
| $\mathbb{D}^2$ **Pruning** | $74.46 \pm 0.52$ | $\underline{72.55 \pm 0.23}$ | $65.59 \pm 0.84$ | $55.73 \pm 0.34$ | $\underline{44.04 \pm 0.41}$ | $\underline{19.16 \pm 0.14}$ | $\underline{10.75 \pm 0.75}$ |
| **GraphCut** | $67.45 \pm 0.35$ | $64.15 \pm 0.54$ | $53.12 \pm 0.30$ | $38.29 \pm 0.12$ | $26.81 \pm 0.32$ | $8.56 \pm 0.64$ | $8.16 \pm 0.54$ |
| **AL-MDN** | $74.51 \pm 0.35$ | $70.36 \pm 0.17$ | $65.26 \pm 0.82$ | $54.51 \pm 0.94$ | $30.85 \pm 1.71$ | $12.33 \pm 1.44$ | $8.97 \pm 1.33$ |
| $k$-**means** | $74.22 \pm 0.31$ | $72.35 \pm 0.19$ | $65.52 \pm 0.98$ | $57.01 \pm 0.49$ | $43.35 \pm 0.60$ | $16.96 \pm 0.78$ | $5.16 \pm 0.69$ |
| $k$-**DPP** | $74.19 \pm 0.38$ | $71.99 \pm 0.27$ | $65.39 \pm 0.53$ | $56.80 \pm 0.17$ | $43.92 \pm 0.53$ | $18.20 \pm 0.44$ | $10.50 \pm 0.62$ |
| **SES (Ours)** | $\mathbf{75.20 \pm 0.29}$ | $\mathbf{73.33 \pm 0.33}$ | $\mathbf{66.52 \pm 0.41}$ | $\mathbf{59.52 \pm 0.14}$ | $\mathbf{45.92 \pm 0.46}$ | $\mathbf{23.39 \pm 0.22}$ | $\mathbf{16.15 \pm 0.62}$ |

Table 16: Results on visual question answering. The best one is **bold**, and the runner-up is underlined.

| Dataset
Sampling rate | CC SBU Align (100%:30.40) | | | | | | |
|---|---|---|---|---|---|---|---|
| | **70%** | **50%** | **20%** | **10%** | **5%** | **2%** | **1%** |
| **Random** | $29.66 \pm 0.15$ | $29.62 \pm 0.05$ | $29.21 \pm 0.26$ | $29.01 \pm 0.14$ | $\underline{28.20 \pm 0.12}$ | $25.51 \pm 0.46$ | $25.11 \pm 0.42$ |
| **Moderate** | $29.96 \pm 0.14$ | $29.67 \pm 0.13$ | $29.53 \pm 0.27$ | $29.11 \pm 0.23$ | $27.30 \pm 0.25$ | $24.85 \pm 0.36$ | $\underline{26.54 \pm 0.41}$ |
| **CCS** | $29.93 \pm 0.06$ | $29.90 \pm 0.05$ | $\underline{29.94 \pm 0.16}$ | $\underline{29.91 \pm 0.25}$ | $27.71 \pm 0.37$ | $25.31 \pm 0.37$ | $25.59 \pm 0.41$ |
| $\mathbb{D}^2$ **Pruning** | $30.09 \pm 0.13$ | $\underline{29.97 \pm 0.19}$ | $29.44 \pm 0.22$ | $29.30 \pm 0.27$ | $26.44 \pm 0.23$ | $25.03 \pm 0.30$ | $26.29 \pm 0.37$ |
| **GraphCut** | $29.86 \pm 0.08$ | $29.73 \pm 0.06$ | $29.53 \pm 0.23$ | $29.11 \pm 0.25$ | $27.30 \pm 0.36$ | $24.85 \pm 0.41$ | $\underline{26.54 \pm 0.29}$ |
| **Instruction** | $\underline{30.12 \pm 0.15}$ | $29.93 \pm 0.25$ | $29.82 \pm 0.27$ | $29.01 \pm 0.28$ | $26.76 \pm 0.42$ | $23.72 \pm 0.49$ | $24.60 \pm 0.46$ |
| $k$-**means** | $29.78 \pm 0.14$ | $29.61 \pm 0.14$ | $29.54 \pm 0.11$ | $29.20 \pm 0.20$ | $27.72 \pm 0.35$ | $25.47 \pm 0.39$ | $25.60 \pm 0.41$ |
| $k$-**DPP** | $29.33 \pm 0.15$ | $29.55 \pm 0.12$ | $29.48 \pm 0.08$ | $29.27 \pm 0.12$ | $28.16 \pm 0.34$ | $\underline{25.93 \pm 0.46}$ | $26.13 \pm 0.34$ |
| **SES (Ours)** | $\mathbf{30.25 \pm 0.10}$ | $\mathbf{30.20 \pm 0.13}$ | $\mathbf{30.21 \pm 0.18}$ | $\mathbf{30.10 \pm 0.11}$ | $\mathbf{28.23 \pm 0.40}$ | $\mathbf{27.19 \pm 0.46}$ | $\mathbf{27.61 \pm 0.39}$ |

## F.2 ACTIVE LEARNING

We present detailed experimental results for active learning in Table 17. Our method consistently performs better than baselines across all sampling rates.

Table 17: Results for active learning. The best one is **bold**, and the runner-up is underlined.

| Dataset
Sampling rate | ImageNet-1K (100%:73.63) | | | | | | |
|---|---|---|---|---|---|---|---|
| | **70%** | **50%** | **20%** | **10%** | **5%** | **2%** | **1%** |
| **Random** | $71.12 \pm 0.25$ | $69.43 \pm 0.04$ | $58.77 \pm 0.35$ | $\underline{47.36 \pm 0.20}$ | $\underline{33.41 \pm 0.15}$ | $\underline{16.41 \pm 0.40}$ | $\underline{5.41 \pm 0.23}$ |
| **Moderate** | $71.48 \pm 0.05$ | $68.68 \pm 0.06$ | $56.35 \pm 0.01$ | $42.29 \pm 0.17$ | $26.77 \pm 0.01$ | $11.37 \pm 0.22$ | $4.22 \pm 0.17$ |
| **CCS** | $71.46 \pm 0.11$ | $\underline{69.50 \pm 0.07}$ | $58.85 \pm 0.16$ | $45.06 \pm 0.22$ | $28.02 \pm 0.17$ | $9.03 \pm 0.40$ | $2.33 \pm 0.18$ |
| **GraphCut** | $71.50 \pm 0.21$ | $69.16 \pm 0.09$ | $56.08 \pm 0.24$ | $40.99 \pm 0.21$ | $24.30 \pm 0.17$ | $7.90 \pm 0.13$ | $2.46 \pm 0.08$ |
| $\mathbb{D}^2$ **Pruning** | $\underline{71.62 \pm 0.11}$ | $69.02 \pm 0.21$ | $\underline{59.65 \pm 0.11}$ | $45.97 \pm 0.32$ | $28.08 \pm 0.37$ | $14.24 \pm 0.11$ | $4.79 \pm 0.31$ |
| **Prototypicality** | $70.12 \pm 0.04$ | $66.00 \pm 0.06$ | $49.20 \pm 0.09$ | $35.27 \pm 0.04$ | $24.14 \pm 0.15$ | $13.88 \pm 0.08$ | $4.95 \pm 0.05$ |
| **SES (Ours)** | $\mathbf{72.11 \pm 0.07}$ | $\mathbf{70.15 \pm 0.17}$ | $\mathbf{60.22 \pm 0.34}$ | $\mathbf{48.10 \pm 0.21}$ | $\mathbf{34.82 \pm 0.22}$ | $\mathbf{17.97 \pm 0.35}$ | $\mathbf{6.69 \pm 0.62}$ |

## F.3 CONTINUAL LEARNING

In addition to baselines from Sec. 5.1, we include nine widely used baseline selection methods for continual learning: 1) **$k$-center** (Sener & Savarese, 2018), which iteratively selects samples

that are least similar to those already selected, 2) **Gradient matching** (Campbell & Broderick, 2019), which selects samples whose average gradient closely approximates the average gradient of all samples, 3) **FRCL** (Titsias et al., 2020), which optimizes a subset of samples to minimize the posterior uncertainty of the Gaussian process induced from the embedding representations, 4) **iCaRL** (Rebuffi et al., 2017) that selects samples whose average embedding closely approximates the average embedding of all samples, 5) **Greedy Coreset** (Borsos et al., 2020) that formulates the selection as a bilevel optimization problem and greedily selects samples such that the model trained on them minimizes the loss across the entire dataset, 6) **BCSR** (Hao et al., 2024) that formulates the selection as a bilevel optimization problem on the probability simplex over samples and introduces a regularizer to control the number of selected samples, 7) **DER++** (Buzzega et al., 2020) that employs reservoir sampling to select samples from the input data stream, and leverages both model logits and ground-truth labels to mitigate the impact of distribution shifts, 8) **GSS** (Aljundi et al., 2019) that selects samples with diverse parameter gradients to ensure that the model's performance on previously learned tasks does not degrade, and 9) **GCR** (Tiwari et al., 2022) that selects and weights samples to minimize the discrepancy between their replay loss gradients and those of the entire dataset.

In addition to the aforementioned memory construction methods, there are methods that optimize the usage of the replay memory. For example, GEM (Lopez-Paz & Ranzato, 2017) prevents an increase in the loss on previous tasks by projecting the gradient updates onto a subspace that satisfies the non-increasing loss constraint. These methods, when combined with memory construction methods, effectively utilize replay memory to mitigate catastrophic forgetting of previously learned tasks. However, our focus is primarily on memory construction, so these methods are not included in our experimental scope.

We conduct the experiments under three settings: class-incremental, task-incremental, and task-free. The class-incremental setting is introduced in Sec. 5.3.1. The task-incremental setting is similar to the class-incremental setting, except that the model is aware of the task of each sample during testing. This specificity allows the model to select labels appropriate for the given task rather than from a pool of all tasks. The task-free setting is designed to simulate continuous shifts in data distribution, where task boundaries are not known during model training. Consequently, the replay memory is updated after each data batch rather than after each task. Specifically, we employ the merge-reduce method for memory updates, as proposed by Borsos et al. (2020). This method divides the memory into equally sized slots, and for each new data batch, representative samples are selected to fill a slot. If all slots are full, two slots that represent the same number of original samples are merged. This is done by selecting representative samples from both slots to form a single updated slot, which then represents the combined samples of the original two. In our implementation, the memory is divided into 10 slots for all memory sizes.

We present detailed experimental results for continual learning in Tables 18, 19, 20, and 21. Our method consistently performs better than baselines across all settings, datasets, and memory sizes.

Table 18: Results for continual learning on Permuted MNIST and Split MNIST. The best one is **bold**, and the runner-up is underlined.

| Dataset | Permuted MNIST | | | | Split MNIST | | | |
|---|---|---|---|---|---|---|---|---|
| Memory size | 50 | 100 | 200 | 400 | 50 | 100 | 200 | 400 |
| **Random** | $75.99 \pm 0.70$ | $77.43 \pm 0.41$ | $79.69 \pm 0.44$ | $81.32 \pm 0.33$ | $92.66 \pm 1.46$ | $95.79 \pm 0.30$ | $97.16 \pm 0.13$ | $98.18 \pm 0.50$ |
| **Moderate** | $76.34 \pm 0.59$ | $78.29 \pm 0.52$ | $79.44 \pm 0.31$ | $78.88 \pm 0.62$ | $92.74 \pm 0.59$ | $95.34 \pm 0.18$ | $97.05 \pm 0.07$ | $97.61 \pm 0.44$ |
| **CCS** | $74.28 \pm 0.75$ | $75.48 \pm 0.46$ | $76.53 \pm 0.53$ | $81.64 \pm 0.16$ | $80.62 \pm 1.01$ | $89.50 \pm 0.92$ | $94.92 \pm 0.76$ | $98.34 \pm 0.17$ |
| $\mathbb{D}^2$ **Pruning** | $\underline{77.19 \pm 0.46}$ | $78.25 \pm 0.30$ | $79.94 \pm 0.36$ | $80.93 \pm 0.12$ | $93.92 \pm 0.22$ | $\underline{96.79 \pm 0.12}$ | $97.69 \pm 6.88$ | $98.07 \pm 0.08$ |
| **GraphCut** | $75.31 \pm 0.42$ | $76.98 \pm 6.74$ | $78.61 \pm 0.36$ | $80.39 \pm 0.29$ | $88.37 \pm 0.37$ | $91.34 \pm 1.09$ | $94.25 \pm 0.71$ | $\underline{98.44 \pm 0.20}$ |
| **Entropy** | $76.74 \pm 0.35$ | $75.54 \pm 0.35$ | $78.18 \pm 0.44$ | $82.08 \pm 0.19$ | $94.33 \pm 0.34$ | $91.54 \pm 0.73$ | $96.16 \pm 0.26$ | $98.27 \pm 0.11$ |
| **Forgetting** | $74.38 \pm 0.54$ | $75.30 \pm 0.25$ | $77.47 \pm 0.40$ | $79.13 \pm 0.70$ | $89.00 \pm 2.43$ | $92.13 \pm 0.43$ | $96.37 \pm 0.20$ | $98.25 \pm 0.18$ |
| **EL2N** | $73.32 \pm 0.80$ | $75.57 \pm 0.24$ | $77.48 \pm 0.20$ | $80.34 \pm 0.47$ | $83.56 \pm 1.05$ | $88.54 \pm 1.01$ | $94.87 \pm 0.48$ | $97.45 \pm 0.13$ |
| **AUM** | $73.98 \pm 0.79$ | $75.54 \pm 0.35$ | $77.47 \pm 0.33$ | $79.90 \pm 0.61$ | $82.31 \pm 2.15$ | $90.38 \pm 1.99$ | $95.79 \pm 0.34$ | $98.07 \pm 0.24$ |
| **Variance** | $74.53 \pm 0.47$ | $75.67 \pm 0.38$ | $77.42 \pm 0.32$ | $78.70 \pm 0.34$ | $84.08 \pm 3.14$ | $91.90 \pm 0.72$ | $94.96 \pm 0.88$ | $97.49 \pm 0.38$ |
| $k$-**center** | $75.64 \pm 0.40$ | $78.17 \pm 0.23$ | $79.75 \pm 0.17$ | $80.94 \pm 0.38$ | $90.41 \pm 1.23$ | $94.39 \pm 0.41$ | $96.61 \pm 0.64$ | $97.80 \pm 0.13$ |
| **Gradient Matching** | $75.73 \pm 0.57$ | $77.30 \pm 0.08$ | $79.27 \pm 0.38$ | $80.71 \pm 0.50$ | $91.58 \pm 1.06$ | $95.39 \pm 0.93$ | $97.54 \pm 0.23$ | $98.39 \pm 0.26$ |
| **FRCL** | $75.78 \pm 0.46$ | $77.33 \pm 0.26$ | $79.21 \pm 0.53$ | $80.88 \pm 0.30$ | $88.46 \pm 0.61$ | $94.48 \pm 0.93$ | $97.10 \pm 0.31$ | $98.33 \pm 0.20$ |
| **iCaRL** | $77.01 \pm 0.19$ | $\underline{78.94 \pm 0.46}$ | $\underline{80.65 \pm 0.36}$ | $81.94 \pm 0.14$ | $92.92 \pm 1.23$ | $89.50 \pm 0.92$ | $97.59 \pm 8.14$ | $98.39 \pm 0.05$ |
| **Greedy Coreset** | $\underline{77.19 \pm 0.56}$ | $78.71 \pm 0.67$ | $80.13 \pm 0.11$ | $\underline{82.17 \pm 0.33}$ | $\underline{94.35 \pm 0.35}$ | $96.07 \pm 0.42$ | $\underline{97.76 \pm 0.85}$ | $98.18 \pm 0.16$ |
| **BCSR** | $75.92 \pm 0.35$ | $77.74 \pm 0.39$ | $79.51 \pm 0.28$ | $81.65 \pm 0.09$ | $93.83 \pm 1.18$ | $94.77 \pm 0.56$ | $96.98 \pm 0.29$ | $98.26 \pm 0.11$ |
| **DER++** | $77.02 \pm 0.40$ | $78.25 \pm 0.07$ | $88.82 \pm 0.05$ | $81.66 \pm 0.09$ | $93.94 \pm 0.18$ | $96.00 \pm 0.16$ | $97.75 \pm 0.08$ | $98.20 \pm 0.15$ |
| **GSS** | $76.11 \pm 1.12$ | $78.32 \pm 0.22$ | $79.79 \pm 0.22$ | $81.69 \pm 0.10$ | $87.51 \pm 0.98$ | $95.37 \pm 0.49$ | $97.61 \pm 0.03$ | $98.29 \pm 0.88$ |
| **GCR** | $76.29 \pm 0.34$ | $78.12 \pm 0.27$ | $79.60 \pm 0.25$ | $80.81 \pm 0.11$ | $93.76 \pm 0.68$ | $95.82 \pm 0.29$ | $97.70 \pm 0.12$ | $98.06 \pm 0.05$ |
| **SES (Ours)** | $\mathbf{78.52 \pm 0.25}$ | $\mathbf{79.92 \pm 0.36}$ | $\mathbf{81.18 \pm 0.26}$ | $\mathbf{82.69 \pm 0.10}$ | $\mathbf{94.73 \pm 0.33}$ | $\mathbf{96.94 \pm 0.34}$ | $\mathbf{98.28 \pm 0.10}$ | $\mathbf{98.54 \pm 0.07}$ |

Table 19: Results for continual learning on Split CIFAR10 and Split CIFAR100. The best one is **bold**, and the runner-up is underlined.

| Dataset | Split CIFAR10 | | | | Split CIFAR100 | | | |
|---|---|---|---|---|---|---|---|---|
| Memory size | 50 | 100 | 200 | 400 | 50 | 100 | 200 | 400 |
| Random | 62.49 ± 0.73 | 62.34 ± 0.89 | 63.69 ± 1.58 | 63.63 ± 0.86 | 46.59 ± 1.28 | 51.29 ± 0.77 | 54.22 ± 0.58 | 56.21 ± 0.46 |
| Moderate | 61.55 ± 0.61 | 61.51 ± 0.46 | 63.01 ± 0.58 | 65.42 ± 3.68 | 47.38 ± 0.39 | 51.91 ± 0.63 | 53.45 ± 8.69 | 55.06 ± 0.86 |
| CCS | 61.14 ± 0.92 | 60.56 ± 1.27 | 61.04 ± 1.73 | 59.33 ± 1.21 | 44.91 ± 0.35 | 48.70 ± 6.90 | 51.26 ± 0.39 | 52.69 ± 0.29 |
| $\mathbb{D}^2$ Pruning | 61.90 ± 1.41 | 64.54 ± 0.63 | 66.08 ± 1.45 | 65.23 ± 1.18 | 47.72 ± 0.61 | 51.86 ± 0.62 | 54.50 ± 0.46 | 57.14 ± 0.42 |
| GraphCut | 60.46 ± 1.29 | 61.02 ± 1.17 | 61.66 ± 1.67 | 63.51 ± 0.77 | 48.72 ± 0.56 | 53.35 ± 0.38 | 54.66 ± 0.28 | 56.74 ± 0.36 |
| Entropy | 61.78 ± 0.44 | 61.53 ± 0.72 | 62.72 ± 0.73 | 63.99 ± 1.01 | 45.08 ± 1.86 | 47.97 ± 1.34 | 51.51 ± 0.34 | 54.42 ± 0.69 |
| Forgetting | 60.81 ± 0.78 | 59.56 ± 0.24 | 61.38 ± 1.22 | 61.59 ± 1.07 | 50.60 ± 1.08 | 53.06 ± 0.57 | 54.68 ± 0.31 | 57.30 ± 0.30 |
| EL2N | 59.00 ± 0.54 | 57.79 ± 0.75 | 58.34 ± 0.68 | 58.58 ± 1.34 | 44.76 ± 1.10 | 46.47 ± 0.62 | 48.11 ± 1.01 | 50.59 ± 0.64 |
| AUM | 58.95 ± 0.77 | 58.32 ± 0.46 | 58.06 ± 0.86 | 59.01 ± 0.67 | 44.70 ± 0.73 | 46.14 ± 0.58 | 47.20 ± 0.40 | 49.24 ± 0.88 |
| Variance | 58.57 ± 0.57 | 58.69 ± 0.24 | 57.77 ± 0.34 | 58.50 ± 0.74 | 50.30 ± 0.84 | 52.83 ± 0.76 | 54.00 ± 0.85 | 56.91 ± 0.28 |
| $k$-center | 61.99 ± 1.36 | 61.47 ± 1.71 | 62.74 ± 1.31 | 64.45 ± 2.24 | 46.90 ± 1.86 | 51.16 ± 0.57 | 53.10 ± 0.17 | 55.77 ± 0.34 |
| Gradient Matching | 60.54 ± 0.92 | 61.65 ± 6.98 | 62.65 ± 1.11 | 63.26 ± 0.96 | 51.15 ± 0.66 | 54.13 ± 0.53 | 56.29 ± 0.26 | 57.24 ± 0.37 |
| FRCL | 61.22 ± 1.61 | 61.67 ± 1.02 | 62.93 ± 0.86 | 65.51 ± 1.57 | 46.81 ± 0.71 | 51.40 ± 0.44 | 54.28 ± 0.55 | 56.01 ± 0.62 |
| iCaRL | 61.95 ± 1.09 | 62.33 ± 0.89 | 64.08 ± 1.58 | 64.09 ± 1.15 | 51.67 ± 0.71 | 54.62 ± 0.51 | 56.11 ± 0.32 | 56.95 ± 0.26 |
| Greedy Coreset | 61.28 ± 1.74 | 63.18 ± 0.84 | 62.98 ± 0.91 | 65.02 ± 1.50 | 52.58 ± 0.39 | 56.17 ± 0.42 | 57.72 ± 0.23 | 55.27 ± 0.66 |
| BCSR | 61.88 ± 0.59 | 63.23 ± 2.60 | 64.59 ± 2.86 | 65.45 ± 0.70 | 47.37 ± 1.25 | 50.21 ± 1.14 | 51.49 ± 0.62 | 59.45 ± 0.42 |
| DER++ | 63.96 ± 0.51 | 62.65 ± 0.02 | 63.85 ± 0.38 | 66.66 ± 1.06 | 45.38 ± 0.10 | 50.00 ± 0.77 | 54.10 ± 0.04 | 56.19 ± 0.11 |
| GSS | 61.33 ± 0.12 | 62.73 ± 0.37 | 62.73 ± 0.37 | 65.17 ± 0.20 | 45.56 ± 1.23 | 50.06 ± 0.95 | 54.88 ± 0.22 | 56.07 ± 0.47 |
| GCR | 62.76 ± 0.51 | 63.12 ± 2.33 | 62.25 ± 0.02 | 66.39 ± 2.01 | 46.83 ± 0.50 | 52.78 ± 0.43 | 55.50 ± 0.35 | 56.31 ± 0.13 |
| SES (Ours) | 67.52 ± 0.56 | 68.26 ± 1.24 | 69.32 ± 0.99 | 70.76 ± 1.22 | 54.71 ± 0.53 | 57.60 ± 0.39 | 59.69 ± 0.31 | 61.06 ± 0.17 |

Table 20: Results for continual learning on Split Tiny-ImageNet. The best one is **bold**, and the runner-up is underlined.

| Dataset | Split Tiny-ImageNet | | | |
|---|---|---|---|---|
| Memory size | 50 | 100 | 200 | 400 |
| Random | 18.38 ± 0.26 | 18.52 ± 0.23 | 19.22 ± 0.29 | 19.22 ± 0.26 |
| Moderate | 18.08 ± 0.30 | 18.63 ± 0.36 | 19.47 ± 6.18 | 19.75 ± 0.24 |
| CCS | 18.47 ± 0.41 | 18.30 ± 0.17 | 18.90 ± 0.13 | 18.50 ± 0.22 |
| $\mathbb{D}^2$ Pruning | 18.98 ± 0.10 | 19.08 ± 0.39 | 19.50 ± 0.28 | 19.76 ± 0.14 |
| GraphCut | 19.50 ± 0.19 | 19.76 ± 6.15 | 20.53 ± 0.12 | 21.57 ± 0.36 |
| Entropy | 18.38 ± 0.42 | 17.99 ± 0.28 | 18.62 ± 0.22 | 19.20 ± 0.25 |
| Forgetting | 18.92 ± 0.19 | 19.55 ± 0.13 | 19.56 ± 0.33 | 19.85 ± 0.30 |
| EL2N | 18.04 ± 0.35 | 17.95 ± 0.31 | 17.98 ± 0.32 | 18.37 ± 0.24 |
| AUM | 18.36 ± 0.41 | 18.24 ± 0.10 | 17.89 ± 0.17 | 18.16 ± 0.21 |
| Variance | 19.66 ± 0.42 | 19.32 ± 0.07 | 19.50 ± 0.33 | 20.19 ± 0.13 |
| $k$-center | 18.81 ± 0.21 | 18.87 ± 0.18 | 18.90 ± 0.30 | 19.13 ± 0.14 |
| Gradient Matching | 19.20 ± 0.32 | 19.19 ± 8.72 | 19.00 ± 0.16 | 19.46 ± 0.18 |
| FRCL | 18.59 ± 0.21 | 18.86 ± 0.15 | 19.01 ± 0.27 | 19.55 ± 0.20 |
| iCaRL | 19.05 ± 0.45 | 19.58 ± 0.18 | 19.85 ± 0.16 | 19.86 ± 0.58 |
| Greedy coreset | 19.61 ± 0.28 | 19.24 ± 0.34 | 19.98 ± 0.17 | 20.62 ± 0.36 |
| BCSR | 19.04 ± 0.36 | 18.75 ± 0.17 | 18.74 ± 0.26 | 19.42 ± 0.17 |
| DER++ | 18.20 ± 0.34 | 18.86 ± 0.34 | 19.09 ± 0.10 | 19.63 ± 0.27 |
| GSS | 18.71 ± 0.11 | 19.40 ± 0.07 | 19.09 ± 0.25 | 19.64 ± 0.15 |
| GCR | 18.23 ± 0.02 | 19.60 ± 0.41 | 18.91 ± 0.24 | 19.85 ± 0.23 |
| SES (Ours) | 20.40 ± 0.34 | 20.80 ± 0.18 | 21.20 ± 0.12 | 21.82 ± 0.20 |

Table 21: Results for continual learning under the task-incremental and task-free settings. The best one is **bold**, and the runner-up is underlined.

| Setting | Task-incremental | | | | Task-free | | | |
|---|---|---|---|---|---|---|---|---|
| Memory size | 50 | 100 | 200 | 400 | 50 | 100 | 200 | 400 |
| Random | 75.41 ± 0.52 | 75.44 ± 0.14 | 75.82 ± 1.99 | 79.78 ± 0.40 | 61.35 ± 0.60 | 61.54 ± 0.09 | 64.28 ± 2.09 | 65.19 ± 1.66 |
| Moderate | 74.79 ± 0.01 | 74.08 ± 0.30 | 75.67 ± 0.03 | 79.22 ± 0.34 | 60.61 ± 0.25 | 62.52 ± 2.70 | 62.90 ± 0.93 | 63.15 ± 1.86 |
| CCS | 69.14 ± 0.74 | 56.93 ± 0.01 | 58.81 ± 0.34 | 67.89 ± 0.73 | 64.33 ± 0.04 | 62.23 ± 0.45 | 64.14 ± 1.33 | 62.02 ± 0.59 |
| $\mathbb{D}^2$ Pruning | 76.75 ± 0.26 | 74.47 ± 1.11 | 77.03 ± 0.04 | 80.69 ± 6.64 | 61.16 ± 0.46 | 64.30 ± 0.40 | 66.15 ± 0.65 | 68.43 ± 8.17 |
| GraphCut | 73.14 ± 2.52 | 74.90 ± 0.73 | 75.22 ± 1.41 | 79.94 ± 0.71 | 60.01 ± 3.41 | 61.26 ± 0.38 | 63.30 ± 4.07 | 63.80 ± 3.35 |
| Entropy | 74.33 ± 0.30 | 73.86 ± 0.36 | 75.34 ± 0.82 | 79.10 ± 0.64 | 60.24 ± 1.77 | 63.28 ± 3.41 | 64.79 ± 1.86 | 63.56 ± 0.16 |
| Forgetting | 65.69 ± 1.03 | 58.62 ± 0.90 | 58.28 ± 0.85 | 71.47 ± 0.80 | 63.02 ± 0.67 | 62.72 ± 1.22 | 61.67 ± 0.70 | 64.77 ± 0.10 |
| EL2N | 56.38 ± 0.27 | 53.86 ± 0.74 | 53.92 ± 2.04 | 65.23 ± 0.03 | 58.95 ± 0.76 | 59.39 ± 2.75 | 60.51 ± 3.64 | 62.82 ± 0.53 |
| AUM | 56.17 ± 1.69 | 53.28 ± 1.64 | 57.43 ± 0.68 | 63.08 ± 0.33 | 60.78 ± 0.37 | 60.70 ± 4.02 | 61.63 ± 0.32 | 63.49 ± 2.52 |
| Variance | 60.07 ± 0.46 | 52.45 ± 1.13 | 54.89 ± 2.54 | 71.81 ± 0.34 | 61.63 ± 1.47 | 66.84 ± 1.21 | 63.93 ± 1.35 | 64.41 ± 1.37 |
| $k$-center | 75.27 ± 0.89 | 72.52 ± 0.35 | 74.76 ± 0.11 | 79.63 ± 0.77 | 63.67 ± 0.02 | 63.91 ± 1.33 | 65.19 ± 3.38 | 68.59 ± 0.50 |
| Gradient Matching | 75.19 ± 0.79 | 73.85 ± 0.30 | 76.31 ± 0.89 | 79.64 ± 1.21 | 64.58 ± 0.14 | 60.56 ± 1.86 | 60.53 ± 2.44 | 64.33 ± 1.30 |
| FRCL | 75.27 ± 0.55 | 74.80 ± 0.13 | 75.78 ± 0.95 | 80.06 ± 0.10 | 64.95 ± 0.40 | 62.18 ± 2.42 | 61.26 ± 0.79 | 62.54 ± 0.31 |
| iCaRL | 74.92 ± 0.66 | 72.95 ± 1.79 | 76.81 ± 0.71 | 80.62 ± 0.30 | 64.68 ± 0.52 | 61.37 ± 1.73 | 63.03 ± 2.75 | 62.86 ± 0.94 |
| Greedy Coreset | 76.04 ± 0.22 | 76.30 ± 1.56 | 79.35 ± 1.09 | 81.00 ± 0.15 | 61.97 ± 4.59 | 61.75 ± 0.87 | 62.63 ± 4.24 | 62.77 ± 0.80 |
| BCSR | 74.15 ± 0.01 | 75.68 ± 0.08 | 79.26 ± 6.28 | 79.80 ± 0.03 | 58.87 ± 0.13 | 59.54 ± 0.44 | 59.99 ± 3.41 | 63.87 ± 2.40 |
| DER++ | 74.59 ± 1.12 | 72.33 ± 1.38 | 76.86 ± 0.73 | 79.63 ± 8.07 | 60.65 ± 1.83 | 60.61 ± 1.48 | 64.00 ± 1.25 | 64.22 ± 0.88 |
| GSS | 75.16 ± 0.59 | 75.32 ± 0.14 | 76.79 ± 0.80 | 79.63 ± 0.14 | 60.47 ± 0.86 | 60.51 ± 3.09 | 62.85 ± 2.30 | 66.18 ± 2.11 |
| GCR | 72.08 ± 2.95 | 72.93 ± 0.30 | 75.09 ± 0.34 | 80.21 ± 0.12 | 63.04 ± 1.37 | 63.75 ± 1.71 | 64.07 ± 3.18 | 62.90 ± 0.62 |
| SES (Ours) | 78.71 ± 0.63 | 80.20 ± 1.17 | 81.61 ± 0.57 | 82.20 ± 6.86 | 66.55 ± 1.75 | 67.33 ± 0.73 | 66.56 ± 1.09 | 69.69 ± 0.53 |

## G   ADDITIONAL ABLATION STUDY

**Robustness to different training difficulty metrics**. We test the effect of using different training difficulty metrics, including Forgetting, EL2N, and AUM. We also include the identity baseline, which assigns identical scores to all samples. Table 22 shows the results. Using identical scores leads to inferior performance, indicating the importance of incorporating training difficulty. Meanwhile, using AUM, Forgetting, and EL2N yields comparable performance, with average accuracies across rates differing by no more than $0.5\%$. This demonstrates the robustness of our method to the choice of the training difficulty metric.

Table 22: Ablation of training difficulty metrics. The best one is **bold**, and the runner-up is underlined.

| Difficulty metric | Sampling rate | | | | | | | Avg. |
|---|---|---|---|---|---|---|---|---|
| | 70% | 50% | 20% | 10% | 5% | 2% | 1% | |
| **Identity** | 94.58 | 92.66 | 86.67 | 78.50 | 67.68 | 53.21 | 44.43 | 73.96 |
| **Forgetting** | 94.86 | 93.99 | 88.11 | 79.06 | **70.45** | 53.99 | **46.43** | 75.27 |
| **EL2N** | 94.61 | 93.74 | **88.45** | 79.93 | 69.52 | 54.01 | 45.15 | 75.06 |
| **AUM** | **95.01** | **94.50** | 88.31 | **80.24** | 69.82 | **54.78** | 45.25 | **75.42** |

**Quantification of node-level structural entropy**. In Eq. (4), we quantify the node-level structural entropy as $S_e(u) = \frac{1}{vol(V)} \sum_{\langle u,v \rangle \in E} w_{u,v} \log vol(\alpha_{u \vee v})$ by removing the second term from Eq. (3). Alternatively, the node-level structural entropy can retain the second term and be quantified as $S_e(u) = \frac{1}{vol(V)}(\sum_{\langle u,v \rangle \in E} w_{u,v} \log vol(\alpha_{u \vee v}) - d(u) \log d(u))$. As shown in Table 23, removing the second term slightly increases performance across all sampling rates. This finding supports our decision to remove the second term in the definition of node-level structural entropy.

Table 23: Ablation on the quantification of node-level structural entropy. The best one is **bold**, and the runner-up is underlined.

| Quantification method | Sampling rate | | | | | | |
|---|---|---|---|---|---|---|---|
| | 70% | 50% | 20% | 10% | 5% | 2% | 1% |
| With the second term | 94.51 | 93.39 | 88.01 | 79.73 | 67.75 | 53.88 | 43.02 |
| Without the second term | **95.01** | **94.50** | **88.31** | **80.24** | **69.82** | **54.78** | **45.25** |

**Methods for combining global and local metrics**. We explore possible alternative methods to combine the global metric $S_e(u)$ and the local metric $S_t(u)$ other than the proposed multiplication. Specifically, we experimented with the sum ($S_e(u) + S_t(u)$), the harmonic mean ($\frac{S_e(u)S_t(u)}{S_e(u)+S_t(u)}$), and the maximum ($\max(S_e(u), S_t(u))$). As shown in Table 24, multiplication achieves the best average performance, validating our choice for combining the two metrics.

Table 24: Ablation on different combination of $S_e$ and $S_t$. The best one is **bold**, and the runner-up is underlined.

| Combination method | Sampling rate | | | | | | | Avg. |
|---|---|---|---|---|---|---|---|---|
| | 70% | 50% | 20% | 10% | 5% | 2% | 1% | |
| $S_e(u) + S_t(u)$ | 94.77 | 94.43 | 87.17 | 79.64 | 69.08 | 53.24 | 44.53 | 71.35 |
| $\frac{S_e(u)S_t(u)}{S_e(u)+S_t(u)}$ | 94.84 | **94.64** | 88.11 | 79.97 | 69.75 | 53.95 | 44.88 | 71.88 |
| $\max(S_e(u), S_t(u))$ | 94.22 | 93.22 | 87.24 | 78.33 | 68.49 | 53.04 | 43.30 | 70.60 |
| $S_e(u) \cdot S_t(u)$ | **95.01** | 94.50 | **88.31** | **80.24** | **69.82** | **54.78** | **45.25** | **72.15** |

**Replacing graphs with hypergraphs**. We have conducted an experiment using hypergraphs (Zhou et al., 2006) to capture the relationships between samples. Following Feng et al. (2019), we construct a hypergraph by adding a hyperedge for each node that contains the node itself and its $k$-nearest neighbors. Due to the lack of the structural entropy theory for hypergraphs, we apply clique

expansion (Agarwal et al., 2006), which creates an edge between every pair of nodes in a hyperedge, to convert the hypergraph into a graph. We then apply our selection method to the resulting graph. Table 25 shows the results. This hypergraph-based method slightly degrades performance compared to the graph-based method, but still performs better than the baseline methods in low-sampling-rate settings. We attribute this slight decline in performance to the information loss during clique expansion, for which a straightforward alternative is not currently available. We see significant potential for future work in developing structural entropy theories for hypergraphs and then applying our method to sample selection.

Table 25: Ablation study on replacing graphs with hypergraphs. The best one is **bold**, and the runner-up is underlined.

| Graph Type | Sampling rate | | | | | | |
|---|---|---|---|---|---|---|---|
| | **70%** | **50%** | **20%** | **10%** | **5%** | **2%** | **1%** |
| **Hypergraph** | 94.81 | 94.37 | 88.12 | 80.05 | 69.30 | 54.50 | 44.72 |
| **Graph** | **95.01** | **94.50** | **88.31** | **80.24** | **69.82** | **54.78** | **45.25** |

**Effect of $k$ in the $k$NN-graph construction**. We test the effect of $k$ in the $k$NN-graph construction. Fig. 5 shows the results when selecting $10\%$ of the samples from CIFAR10. As $k$ increases, the performance first increases and then remains relatively stable within a narrow range. This indicates that the first few nearest neighbors effectively capture the global structure of the samples, which aligns with prior research (Jaffe et al., 2020). Based on the grid search results across datasets, we empirically determine $\log_2 n$ to be an appropriate value for $k$, where $n$ is the number of samples.

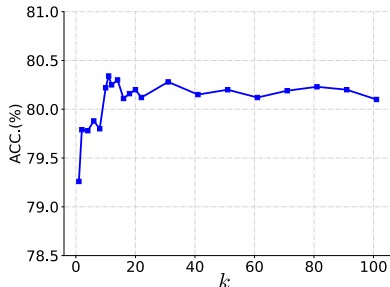

Figure 5: Ablation of $k$ in the $k$NN-graph construction.

## H    FULL QUALITATIVE RESULTS

Fig. 6 visualizes the results of different methods when selecting 2% of the samples from CIFAR10. Methods that select the most difficult samples, such as AUM, oversample near several class boundaries and undersample in several classes that are easier to classify. Methods that prioritize sample coverage, such as $\mathbb{D}^2$ Pruning and CCS, achieve a better sample coverage but may undersample near several class boundaries and fail to preserve the global structure. Our method well covers the data distribution, providing a set of informative and representative samples for model training.

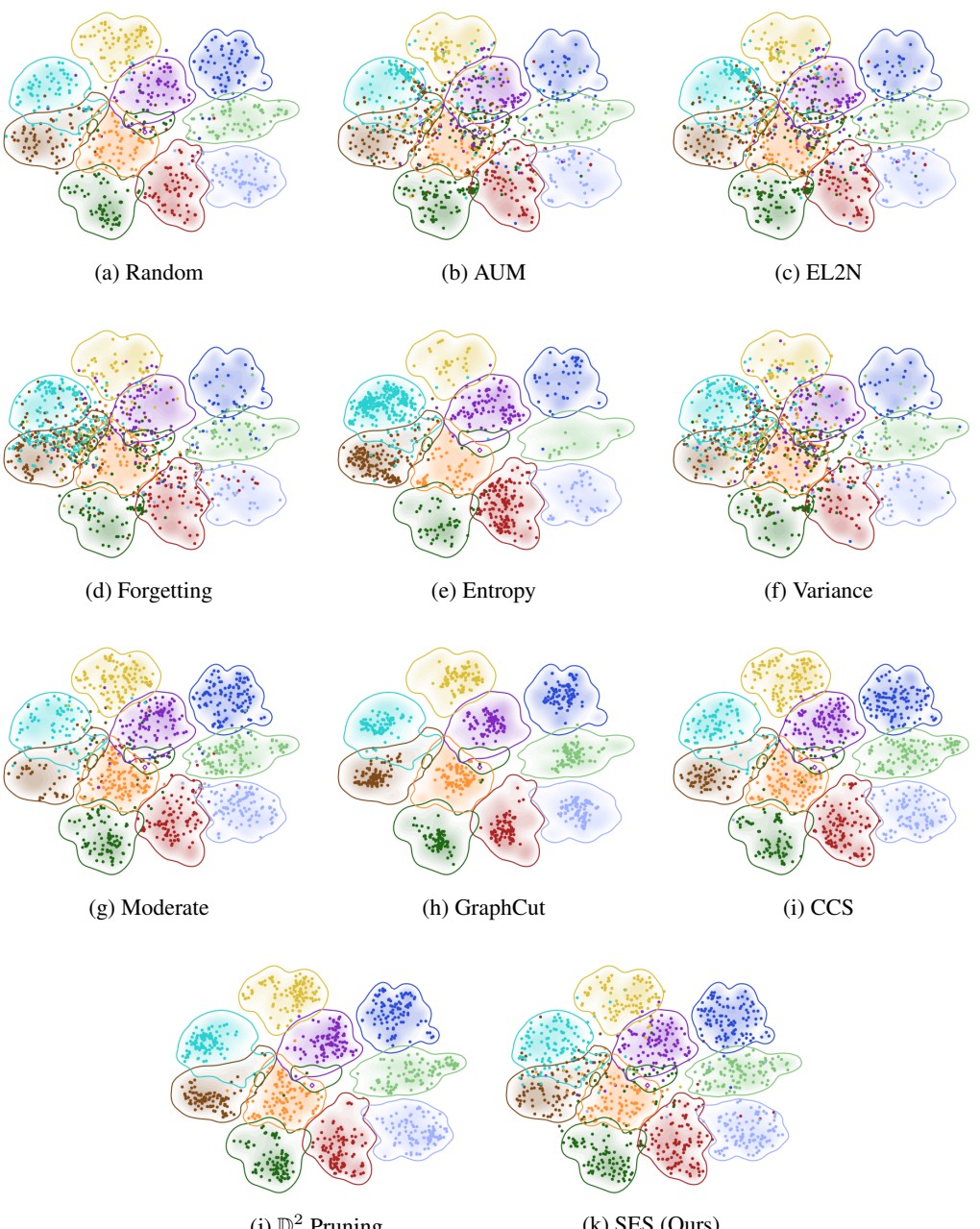

Figure 6: Visualizations of selection results of different methods when selecting 2% of the samples from CIFAR10.

