# OpenReview forum: "Structural-Entropy-Based Sample Selection for Efficient and Effective Learning"
_ICLR.cc/2025/Conference — ICLR 2025 Poster_

### Official Review · Reviewer_j2F4 · 2024-10-26

**Soundness:** 2
**Presentation:** 3
**Contribution:** 2
**Rating:** 6
**Confidence:** 4

**Summary:**

This paper proposes a sample selection method with structural entropy. It first introduces the structural entropy of a graph into the sample selection task, and then applies the Shapley value to decompose the structural entropy of a graph to the node-level structural entropy. At last, it designs a score for sample selection with the node-level structural entropy.

**Strengths:**

1. The idea is interesting and reasonable.
2. The experiments are sufficient and the experimental results are good.

**Weaknesses:**

1. The main techniques used in this paper are all existing methods, e.g. the structural entropy and Shapley value. The paper seems an application of these techniques in the sample selection task. The paper should clarify its novelty and technical contributions.

2. The paper derives Eq.(3) from the Shapley value, and then removes the second term, leading to Eq.(4), which is used in the method. However, the motivation is unclear. Why should we remove the second term? What is the advantage of removing this term? The ablation study of this term should be conducted in the experiments. The paper only claims that they want to design a global metric and the second term is a local metric. I do not think it's convincing enough. What is the advantage of only considering global information? Moreover, in Eq.(5), the method combines the global score S_e and the local score S_t. It seems a little contradictory. In Eq.(4), they do not want the local term but in Eq.(5) they need the local term. It seems strange and not well-motivated.

3. Eq.(5) needs more justification. Why should we multiply S_e and S_t? What is the advantage? Why not just sum S_e and S_t, or combine them with other forms? I think more ablation study is needed to justify the effectiveness of Eq.(5).

4. In the experiments, the experimental setup should be introduced in more detail. For example, what is the difference between the supervised setting and the active learning setting? They both need to select some samples to train the model and then is used to predict the test data.

**Questions:**

Please see Weaknesses.

---

> ### Author Response · Authors · 2024-11-19
> **Response to Review (1/2)**
>
> We sincerely thank you for your appreciation of our idea and your insightful suggestions. We have carefully considered your feedback and revised the paper accordingly. Please see our response below.
>
> ## **Q1**: Clarification of novelty and technical contribution.
>
> **A1**: We have clarified the contribution of our work in the third and fifth paragraphs of the **Introduction**. The **first contribution** of our work is a node-level structural entropy metric that quantifies the importance of individual nodes in preserving the global structure. Existing sample selection methods focus primarily on local information and overlook global information, which can result in suboptimal selection. To address this limitation, we explored possible solutions and identified graph structural entropy as a metric that accurately captures the structure of a sample graph. However, it only provides a single value for the whole graph. It is challenging to decompose this metric to the level of individual nodes, limiting its utility for fine-grained, node-level selection. To overcome this, we used the Shapley value to decompose the structural entropy. However, this introduced another challenge: the exponential-time complexity of calculating the Shapley value. To address this, we reformulated the Shapley value and proposed a node-level structural entropy metric that enables linear-time calculation with respect to the edge number. Based on this, our **second technical contribution** is developing a structural-entropy-based sample selection method that tightly integrates both global and local metrics. The experiment results in **Tables 1, 2, 3** of our paper clearly demonstrate that our method consistently improves state-of-the-art methods across all scenarios and tasks. We hope that our work can encourage a closer look at the incorporation of global information into sample selection and potentially spark interest in further exploration.
>
> ## **Q2**: Justification for Eq. (4) and Eq. (5).
>
> **A2**:  We have fully revised the justification for Eq. (4) and added a theoretical justification and an ablation study to support the quantification of node-level structural entropy by Eq. (4). The theoretical justification is provided in **Sec. 4.1** and **Appendix C**. We have proved that $S_e(u)$ provides a lower bound for the sample coverage. This result indicates that maximizing $S_e(u)$ during selection inherently improves sample coverage. Given the strong correlation between sample coverage and the empirical loss of learning algorithms [1], selecting samples with high node-level structural entropy effectively enhances model performance. The ablation study is provided in **Appendix G**, and the results on CIFAR10 are as follows. These results demonstrate that removing the second term slightly increases performance across all sampling rates.  This finding supports our decision to remove the second term in quantifying node-level structural entropy.
>
> | Quantification of node-level structural entropy              | 70%                 | 50%                 | 20%                 | 10%                 | 5%                  | 2%                  | 1%                  |
> | ------------------------------------------------------------ | ------------------- | ------------------- | ------------------- | ------------------- | ------------------- | ------------------- | ------------------- |
> | $S_e(u)=\frac{1}{vol(V)}(\sum_{\langle u, v \rangle\in E} w_{u,v}\log vol(\alpha_{u\vee v})-d(u)\log d(u))$ | $\underline{94.51}$ | $\underline{93.39}$ | $\underline{88.01}$ | $\underline{79.73}$ | $\underline{67.75}$ | $\underline{53.88}$ | $\underline{43.02}$ |
> | $S_e(u)=\frac{1}{vol(V)}\sum_{\langle u, v \rangle\in E} w_{u,v}\log vol(\alpha_{u\vee v})$ | **95.01**           | **94.50**           | **88.31**           | **80.24**           | **69.82**           | **54.78**           | **45.25**           |
>
> For prioritizing high-difficulty samples (local information), previous research [2, 3] has demonstrated that training difficulty is an effective local metric for quantifying a sample’s impact on model performance.  This is because prioritizing high-difficulty samples enables the model to focus on challenging cases, which are typically more informative for improving decision boundaries and generalization [2, 3]. We make this clearer in **Sec. 4.2**.

---

> ### Author Response · Authors · 2024-11-19
> **Response to Review (2/2)**
>
> ## **Q3**: Combination methods of global and local metrics.
>
> **A3**: We explore possible alternative methods to combine the global metric $S_e(u)$ and the local metric $S_t(u)$ other than the proposed multiplication.
> Specifically, we experiment with the sum ($S_e(u)+S_t(u)$), the harmonic mean ($\frac{S_e(u) S_t(u)}{S_e(u)+S_t(u)}$), and the maximum ($\max(S_e(u),S_t(u))$). The ablation study is provided in **Appendix G**, and the results on CIFAR10 are as follows. These results demonstrate that multiplication achieves the best average performance, validating our choice for combining the two metrics.
>
> | Combination method                   | 70%                 | 50%                 | 20%                 | 10%                 | 5%                  | 2%                  | 1%                  | Avg.                |
> | ------------------------------------ | ------------------- | ------------------- | ------------------- | ------------------- | ------------------- | ------------------- | ------------------- | ------------------- |
> | $S_e(u) + S_t(u)$                    | 94.77               | 94.43               | 87.17               | 79.64               | 69.08               | 53.24               | 44.53               | 71.35               |
> | $\frac{S_e(u)S_t(u)}{S_e(u)+S_t(u)}$ | $\underline{94.84}$ | **94.64**           | $\underline{88.11}$ | $\underline{79.97}$ | $\underline{69.75}$ | $\underline{53.95}$ | $\underline{44.88}$ | $\underline{71.88}$ |
> | $\max(S_e(u), S_t(u))$               | 94.22               | 93.22               | 87.24               | 78.33               | 68.49               | 53.04               | 43.30               | 70.60               |
> | $S_e(u) \cdot S_t(u)$                | **95.01**           | $\underline{94.50}$ | **88.31**           | **80.24**           | **69.82**           | **54.78**           | **45.25**           | **72.15**           |
>
> ## **Q4**: Detailed experimental setup.
>
> **A4**: Thank you for pointing this out. We have clarified in **Sec. 5.2** that the key difference between active learning and supervised learning lies in the absence of labels during sample selection, which makes the selection process more challenging. Additionally, we have revised the whole **Sec. 5** and included more details on the experimental setup.
>
> ## References
>
> [1] Haizhong Zheng, Rui Liu, Fan Lai, and Atul Prakash. Coverage-Centric Coreset Selection for High Pruning Rates. In *Proceedings of the International Conference on Learning Representations*, 2022.
>
> [2] Mansheej Paul, Surya Ganguli, and Gintare Karolina Dziugaite. Deep Learning on a Data Diet: Finding Important Examples Early in Training. In *Proceedings of Advances in Neural Information Processing Systems*, pp. 20596–20607, 2021.
>
> [3] Ben Sorscher, Robert Geirhos, Shashank Shekhar, Surya Ganguli, and Ari Morcos. Beyond Neural Scaling Laws: Beating Power Law Scaling via Data Pruning. In *Proceedings of Advances in Neural Information Processing Systems*, pp. 19523–19536, 2022.
>
> We hope that our responses have addressed your concerns satisfactorily. We are grateful for your insightful comments, which have helped us improve the clarity and quality of our manuscript.

---

> > ### Comment · Reviewer_j2F4 · 2024-11-26
> >
> > Thanks for your responses. Some of my concerns have been addressed, and I'd like to rise my score to 6.

---

> > > ### Author Response · Authors · 2024-11-26
> > > **Thank You Very Much for Your Feedback**
> > >
> > > We sincerely thank you for your feedback.
> > > We are glad to hear that our responses address your concerns.
> > > We deeply appreciate your time and effort in reviewing our paper and providing insightful and constructive comments, which are valuable for improving our paper.
> > > If you have any further suggestions or questions, we will make every effort to address them.

---

> ### Author Response · Authors · 2024-11-24
> **Looking forward to further feedback**
>
> We sincerely thank you again for your recognition of our work. As we mentioned in the previous response, we have revised the paper per your suggestions. We hope that these responses have addressed your concerns satisfactorily. We are grateful for your insightful comments, which have helped us improve the clarity and quality of our manuscript.
>
> If you have any additional comments, we will do our best to address them.

---

### Official Review · Reviewer_4FM2 · 2024-10-30

**Soundness:** 2
**Presentation:** 3
**Contribution:** 2
**Rating:** 6
**Confidence:** 4

**Summary:**

This paper proposes a novel sample selection method that can capture both local and global information of data simultaneously. Specially, first, they use a $k$-NN to construct the sample graph of original data. Second, they employ structural entropy to measure global information and use training difficulty to capture the local information of the graph. At lats, they utilize the importance-biased blue noise sampling to select a set of diverse and representative samples.

The main contribution of this paper is to propose a new global metric for sample selection.

**Strengths:**

1.The writing of this paper is good and easy to follow.

2.This paper has a certain degree of innovation, introducing a structural entropy as a global metric for sample selection.

**Weaknesses:**

1. Hypergraphs [1] can capture more complex relationships, and I believe that if the authors had used hypergraphs instead, the results would have been even better. I would be delighted to see the authors add new experiments to verify my hypothesis.

2. How would the authors define informative and representative samples? They mention these concepts multiple times, but do not provide a detailed explanation.

3. Taking sample as a node is not very reasonable, as it treats the sample as a whole and ignores the local information contained in the sample itself. Perhaps using a Hyper-class representation [2] would yield better results. I would be delighted to see the authors conduct new experiments to verify my hypothesis.

4. The experimental results do not show the variance, so I hope the authors can make up the variance.

5. Just a little confused, why most of the comparison algorithm's experimental results are not as effective as random selection? So what's the meaning of those comparison algorithms? Or are those comparison algorithm chosen by the author appropriate? Are the parameters of the those algorithms not optimally tuned?

6. The effect of blue noise sampling (BNS) is not as good as message passing (MP) when the sampling rate is greater than or equal to 20%, why not directly use MP? Simply because BNS does not require hyperparameter tuning, it sounds far-fetched.

7. As can be seen from Figure 4, (b) has better boundary division, while (c) has several categories of samples grouped together. It seems that (b) is better. How can we understand this phenomenon?

[1] Learning with hypergraphs: Clustering, classification, and embedding.

[2] Hyper-class representation of data.

**Questions:**

Please see the weaknesses.

---

> ### Author Response · Authors · 2024-11-19
> **Response to Review (1/3)**
>
> We sincerely thank you for recognizing the novelty of our paper and providing constructive suggestions. We have carefully revised our paper accordingly. Please see our response below.
>
> ## **Q1**: Hypergraph experiments.
>
> **A1**:  We have conducted an experiment using hypergraphs to capture the relationships between samples. Following the previous work [1], we construct a hypergraph by adding a hyperedge for each node that contains the node itself and its $k$-nearest neighbors. Due to the lack of the structural entropy theory for hypergraphs, we apply clique expansion [2] to convert the hypergraph into a graph and perform our selection method on the resulting graph. The results on CIFAR10 are as follows. This hypergraph-based method slightly degrades performance compared to the graph-based method, but still performs better than the baseline methods in low-sampling-rate settings. We attribute this slight decline in performance to the information loss during clique expansion, for which a straightforward alternative is not currently available. We see significant potential for future work in developing structural entropy theories for hypergraphs and then applying our method to sample selection. We have included this detailed ablation study in **Appendix G** for further reference.
>
>
> | Method                   | 70%                 | 50%                 | 20%                 | 10%                 | 5%                  | 2%                  | 1%                  |
> | ------------------------ | ------------------- | ------------------- | ------------------- | ------------------- | ------------------- | ------------------- | ------------------- |
> | Random                   | 94.29               | 92.33               | 84.17               | 71.95               | 59.88               | 45.51               | 35.44               |
> | Moderate                 | 94.04               | 92.36               | 80.96               | 63.91               | 49.94               | 32.00               | 26.18               |
> | CCS                      | 94.23               | 93.83               | 83.85               | 75.56               | 65.89               | 49.02               | 40.24               |
> | $\mathbb{D}^2$ Pruning   | 93.17               | 93.70               | 86.83               | 76.56               | 65.12               | 45.44               | 39.03               |
> | GraphCut                 | 94.38               | 92.43               | 81.98               | 69.29               | 59.36               | 46.93               | 39.68               |
> | Entropy                  | 94.13               | 92.39               | 75.34               | 60.66               | 46.17               | 35.85               | 29.28               |
> | Forgetting               | 94.76               | 94.34               | 57.72               | 35.21               | 30.87               | 27.22               | 23.62               |
> | EL2N                     | 94.75               | 94.10               | 46.41               | 22.40               | 15.66               | 13.05               | 12.75               |
> | AUM                      | $\underline{94.90}$ | $\underline{94.38}$ | 51.98               | 30.56               | 22.92               | 18.10               | 14.50               |
> | Variance                 | 90.45               | 85.81               | 64.90               | 53.64               | 44.45               | 36.61               | 31.46               |
> | $k$-means                | 93.87               | 93.19               | 83.68               | 72.46               | 62.68               | 48.35               | 38.73               |
> | $k$-DPP                  | 93.84               | 92.72               | 84.09               | 74.62               | 61.95               | 46.58               | 35.16               |
> | ------------------------ | -------             | -------             | -------             | -------             | -------             | -------             | -------             |
> | Hypergraph               | 94.81               | 94.37               | $\underline{88.12}$ | $\underline{80.05}$ | $\underline{69.30}$ | $\underline{54.50}$ | $\underline{44.72}$ |
> | Graph                    | **95.01**           | **94.50**           | **88.31**           | **80.24**           | **69.82**           | **54.78**           | **45.25**           |
>
> ## **Q2**: Definition of informative and representative samples.
>
> **A2**: Thank you for pointing this out. We have now clarified in the **Introduction** that informative samples are those that significantly reduce model uncertainty and are crucial for improving the accuracy and robustness of the training process, while representative samples are those that preserve the diversity and overall distribution of the dataset [3].

---

> ### Author Response · Authors · 2024-11-19
> **Response to Review (2/3)**
>
> ## **Q3**: Hyper-class representation.
>
> **A3**: Thank you for suggesting using a hyper-class representation to better capture information at the sample level. We discuss this method in **Sec. 6**. While we believe it has significant potential and can inspire more follow-up work, we have not included it in our current study for two reasons. First, it is orthogonal to our main contribution in incorporating global information to select high-quality samples. Second, it is not straightforward to combine the hyper-class representation with our method because it is currently designed for tabular data.  We have discussed this with the second author of the referenced paper (the primary implementer of the method). In the discussion, he indicated that adapting the hyper-class representation requires exploring how to handle image and text embeddings rather than tabular data used in the original paper, and fully exploring this idea definitely deserves a separate publication. We believe this is a highly promising research direction to explore in the future.
>
> ## **Q4**: Lack of variance in experimental results.
>
> **A4**: Thank you for the suggestion.  We have added the variance of all experimental results in **Appendix F**. Due to the space limitations, here we only provide the results on CIFAR10.
>
> | Method                 | 70%                         | 50%                         | 20%                         | 10%                         | 5%                           | 2%                           | 1%                           |
> | ---------------------- | --------------------------- | --------------------------- | --------------------------- | --------------------------- | ---------------------------- | ---------------------------- | ---------------------------- |
> | Random                 | $ 94.29\pm 0.08$            | $ 92.33\pm 0.15$            | $ 84.17\pm 0.62$            | $ 71.95\pm 1.50$            | $ 59.88\pm 2.45$             | $ 45.51\pm 1.68$             | $ 35.44\pm 1.49$             |
> | Moderate               | $94.04\pm 0.26$             | $92.36\pm 0.17$             | $80.96\pm 2.56$             | $63.91\pm 1.56$             | $49.94\pm 0.70$              | $32.00\pm 2.11$              | $26.18\pm 1.94$              |
> | CCS                    | $ 94.23\pm 0.35$            | $ 93.83\pm 0.15$            | $ 83.85\pm 1.10$            | $ 75.56\pm 2.39$            | $ \underline{65.89\pm 1.04}$ | $ \underline{49.02\pm 1.35}$ | $ \underline{40.24\pm 0.96}$ |
> | $\mathbb{D}^2$ Pruning | $ 93.17\pm 0.11$            | $ 93.70\pm 0.20$            | $\underline{86.83\pm 0.48}$ | $\underline{76.56\pm 1.14}$ | $ 65.12\pm 0.88$             | $ 45.44\pm 1.75$             | $ 39.03\pm 1.69$             |
> | GraphCut               | $ 94.38\pm 0.20$            | $ 92.43\pm 0.23$            | $ 81.98\pm 0.48$            | $ 69.29\pm 0.57$            | $ 59.36\pm 2.43$             | $ 46.93\pm 2.21$             | $ 39.68\pm 1.15$             |
> | Entropy                | $ 94.13\pm 0.30$            | $ 92.39\pm 0.40$            | $ 75.34\pm 4.32$            | $ 60.66\pm 6.23$            | $ 46.17\pm 5.96$             | $ 35.85\pm 4.33$             | $ 29.28\pm 4.29$             |
> | Forgetting             | $ 94.76\pm 0.38$            | $ 94.34\pm 0.17$            | $ 57.72\pm 1.17$            | $ 35.21\pm 0.87$            | $ 30.87\pm 0.57$             | $ 27.22\pm 0.49$             | $ 23.62\pm 0.74$             |
> | EL2N                   | $ 94.75\pm 0.32$            | $ 94.10\pm 0.45$            | $ 46.41\pm 4.84$            | $ 22.40\pm 0.62$            | $ 15.66\pm 0.18$             | $ 13.05\pm 0.20$             | $ 12.75\pm 0.56$             |
> | AUM                    | $\underline{94.90\pm 0.17}$ | $\underline{94.38\pm 0.20}$ | $ 51.98\pm 1.32$            | $ 30.56\pm 0.42$            | $ 22.92\pm 0.38$             | $ 18.10\pm 0.47$             | $ 14.50\pm 1.35$             |
> | Variance               | $ 90.45\pm 0.09$            | $ 85.81\pm 0.39$            | $ 64.90\pm 0.53$            | $ 53.64\pm 1.02$            | $ 44.45\pm 0.77$             | $ 36.61\pm 0.70$             | $ 31.46\pm 0.92$             |
> | $k$-means              | $93.87\pm 0.33$             | $93.19\pm 0.36$             | $83.68\pm 0.94$             | $72.46\pm 0.51$             | $62.68\pm 1.12$              | $48.35\pm 2.29$              | $38.73\pm 0.87$              |
> | $k$-DPP                | $93.84\pm 0.10$             | $92.72\pm 0.18$             | $84.09\pm 1.21$             | $74.62\pm 0.40$             | $61.95\pm 4.87$              | $46.58\pm 1.76$              | $35.16\pm 2.92$              |
> | SES (Ours)             | **95.01 $\pm$ 0.08**        | **94.50 $\pm$ 0.05**        | **88.31 $\pm$ 0.13**        | **80.24 $\pm$ 0.72**        | **69.82 $\pm$ 0.84**         | **54.78 $\pm$ 1.36**         | **45.25 $\pm$ 0.81**         |

---

> ### Author Response · Authors · 2024-11-19
> **Response to Review (3/3)**
>
> ## **Q5**: Effectiveness and appropriateness of the comparison algorithms.
>
> **A5**: Random selection is a robust method that requires no prior assumptions about what constitutes an ''optimal" sample, and it has consistently served as a strong baseline across various settings in prior evaluations [4, 5]. In contrast, more complex selection methods perform well in specific settings.
> For example, the methods that select the most difficult samples perform better when a large number of samples are selected, as they can enhance the model’s learning by exposing it to challenging samples.  However, when the sample size is limited, these methods often perform worse because only focusing on difficult samples may lead to an unrepresentative subset that lacks coverage, which harms model generalization. To ensure a comprehensive and fair comparison, we mainly followed $\mathbb{D}^2$ pruning [5] to choose the comparison algorithms with diverse characteristics. We set the algorithm hyperparameters according to the configurations reported in the original papers.
>
> ## **Q6**: Comparison between message passing (MP) and blue noise sampling (BNS).
>
> **A6**: The hyperparameter tuning for MP requires a time-consuming grid search based on model training results. In our experiments, the optimal hyperparameter ranged from 0.1 to 100, and the grid search took $15$ times longer than our method (a single model training). We have now clarified the considerable time cost of MP in **Sec. 5.4**. Therefore, we use BNS in our method because it requires no hyperparameter tuning and still achieves performance comparable to that of MP when the sampling rate is greater than or equal to 20% (the differences range from -0.07% to -0.23%).
>
> ## **Q7**: Explanation of Figure 4.
>
> **A7**: Thank you for pointing this out.  We make this clearer in **Sec. 5.5** that the colored contours in Figure 4 indicate the high-density regions for each class. The overlapping regions contain ambiguous samples from different classes, which are important for the model to learn to differentiate. In Figure 4(b), $\mathbb{D}^2$ pruning misses important samples in these regions, limiting the model’s ability to differentiate samples in these regions. Conversely, in Figure 4(c), our method well covers the data distribution, leading to improved model training outcomes.
>
> ## References
>
> [1] Yifan Feng, Haoxuan You, Zizhao Zhang, Rongrong Ji, and Yue Gao. Hypergraph Neural Networks. In *Proceedings of the AAAI conference on artificial intelligence*, pp. 3558-3565, 2019.
>
> [2] Sameer Agarwal, Kristin Branson, and Serge Belongie. Higher Order Learning with Graphs. In *Proceedings of the International Conference on Machine Learning*, pp. 17-24, 2006.
>
> [3] Sheng-Jun Huang, Rong Jin, and Zhi-Hua Zhou. Active learning by querying informative and
> representative examples. *IEEE Transactions on Pattern Analysis and Machine Intelligence*, 36(10):
> 1936–1949, 2014.
>
> [4] Chengcheng Guo, Bo Zhao, and Yanbing Bai. Deepcore: A Comprehensive Library for Coreset Selection in Deep Learning. In *Proceedings of International Conference on Database and Expert Systems Applications*, pp. 181–195, 2022.
>
> [5] Adyasha Maharana, Prateek Yadav, and Mohit Bansal. $\mathbb{D}^2$ Pruning: Message Passing for Balancing Diversity & Difficulty in Data Pruning. In *Proceedings of the International Conference on Learning Representations*, 2024.
>
> We hope that our responses have addressed your concerns satisfactorily. We are grateful for your insightful comments, which have helped us improve the clarity and quality of our manuscript.

---

> ### Author Response · Authors · 2024-11-24
> **Looking forward to further feedback**
>
> We sincerely thank you again for your recognition of our work. As we mentioned in the previous response, we have revised the paper per your suggestions. We hope that these responses have addressed your concerns satisfactorily. We are grateful for your insightful comments, which have helped us improve the clarity and quality of our manuscript.
>
> If you have any additional comments, we will do our best to address them.

---

> > ### Comment · Reviewer_4FM2 · 2024-11-26
> > **Further feedback**
> >
> > I really appreciate the feedback and efforts from the authors. Some of the answers address part of my concerns. I raise my grade from 5 to 6.

---

> > > ### Author Response · Authors · 2024-11-26
> > > **Thank You Very Much for Your Feedback**
> > >
> > > We sincerely thank you for your feedback.
> > > We are glad to hear that our responses address your concerns.
> > > We deeply appreciate your time and effort in reviewing our paper and providing insightful and constructive comments, which are valuable for improving our paper.
> > > If you have any further suggestions or questions, we will make every effort to address them.

---

### Official Review · Reviewer_g3Pt · 2024-10-31

**Soundness:** 2
**Presentation:** 3
**Contribution:** 2
**Rating:** 5
**Confidence:** 3

**Summary:**

The paper proposes a Structural-Entropy-based sampling (SES) method for data selection. This approach integrates global structural information, quantified using structural entropy, and local information (training difficulty) to choose informative and representative samples for training. The authors show that  incorporating global structural information can improve the quality of sample selection. Traditional methods often focus on local data properties, such as training difficulty, while ignoring broader connectivity patterns. SES constructs a sample graph using k-nearest neighbor (kNN) relationships to model sample similarity and applies structural entropy at a node level to evaluate each sample’s significance in preserving the overall data structure.

**Strengths:**

1, The method utilizing the encoding tree effectively measures local structural information across different scales, providing valuable insights.  Various experiments on different learning tasks demonstrate SES’s effectiveness.

2, Structural entropy is decomposed from the overall graph level to individual nodes, resulting in a node-level definition of structural entropy. By leveraging properties of the Shapley value, authors show that the Shapley value of a node can be computed in linear time relative to the number of edges.

**Weaknesses:**

1, The overall idea is relatively simple and incremental, mainly based on the concept from Li and Pan 2016. Also, the proposed method is very heuristic without solid theoretical validation. For example, in line 212-213, "Given our emphasis on ..., we only use....". Can you give more detailed explanation? In my opinion, this is not a serious claim for a research article.  In line 352, you say "10X speedup" because you use only 10\% of dataset. Can you provide the realistic experimental time for this claim? Do you count the construction time of your sample?

2, While the Shapley value of a node can be computed in linear time relative to the number of edges, the edges in a kNN graph is O(k|X|). It is still  time consuming to calculate the Shapley values for all nodes of the encoding tree, especially for large dataset.

3, The author does not clearly explain why preserving the global structure of the dataset is beneficial or provide theoretical guarantees regarding its impact on performance in tasks such as supervised, active, and continual learning. Similarly, the rationale for prioritizing samples with high local information (i.e., training difficulty) is insufficiently justified.

4, The blue noise sampling method effectively promotes diversity, yet the balance between selecting challenging samples and maintaining sample diversity could be further clarified. A comparative analysis with methods that explicitly optimize for diversity, such as clustering-based selections, would help to clarify the advantages of SES.

5, The method relies on the quality of the embedding used to construct the kNN graph. If the embedding representation is poor, structural entropy may not accurately capture the global structure of the data.

6, Some experimental parts are not sufficient. For example, in the continual learning part, they only consider three baselines and two memory sizes (100, 200). Moreover, do you consider both Class-Incremental Learning and Task-Free Continual Learning?

**Questions:**

1, The construction method of the Encoding tree is challenging. The author mentioned the Huffman tree construction; how does this affect the conclusions of this article?

2, Equation (5) in the paper calculates the overall importance score of the node. Is this importance score node-wise or point-wise sampling? A data point may appear in multiple nodes, does this affect the proposed sampling method?

3, Have you considered other ways of combining global structural entropy and local training difficulty indicators, rather than through the multiplication method (S(u) = Se(u) * St(u))?

4, Can you provide more theoretical explanations or proofs to demonstrate why structural entropy, as a global indicator, is helpful for considered tasks (e.g. supervised, active, and continual learning)?

---

> ### Author Response · Authors · 2024-11-19
> **Response to Review (1/4)**
>
> We sincerely appreciate your valuable and helpful suggestions.
> Per your suggestions, we have carefully addressed the identified weaknesses.
> Please see our response below.
>
> ## **Q1**: Incremental contribution, lack of theoretical validation, and missing experimental time.
>
> **A1**: For the **concern on contribution**, we clarified it in the third and fifth paragraphs of the **Introduction**.
> The **first contribution** of our work is a node-level structural entropy metric that quantifies the importance of individual nodes in
> preserving the global structure.
> Existing sample selection methods focus primarily on local information and overlook global information, which can result in suboptimal selection.
> To address this limitation, we explored possible solutions and identified graph structural entropy as a metric that accurately captures the structure of a sample graph.
> However, it only provides a single value for the whole graph.
> It is challenging to decompose this metric to the level of individual nodes, limiting its utility for fine-grained, node-level selection.
> To overcome this, we used the Shapley value to decompose the structural entropy.
> However, this introduced another challenge: the exponential-time complexity of calculating the Shapley value.
> To address this, we reformulated the
> Shapley value and proposed a node-level structural entropy metric that enables linear-time calculation with respect to the edge number.
> Based on this, our **second technical contribution** is developing an efficient structural-entropy-based sample selection method that tightly integrates both global and local metrics.
> The experiment results in **Tables 1, 2, 3** of our paper clearly demonstrate that our method consistently improves state-of-the-art
> methods across all scenarios and tasks.
> We hope that our work can encourage a closer look at the incorporation of global information into sample selection and potentially spark interest in further exploration.
>
> For the **concern on theoretical validation**, we have added a theoretical justification and conducted an ablation study to support quantifying node-level structural entropy by Eq. (4).
> The theoretical justification is provided in **Sec. 4.1** and **Appendix C**.
> We have proved that $S_e(u)$ provides a lower bound for the sample coverage.
> This result indicates that maximizing $S_e(u)$ during selection inherently improves sample coverage.
> Given the strong correlation between sample coverage and the empirical loss of learning algorithms [1], selecting samples with high node-level structural entropy effectively enhances model performance.
> The ablation study is provided in **Appendix G**, and the results on CIFAR 10 are as follows.
> These results demonstrate that removing the second term slightly increases performance across all sampling rates.
> This finding supports our decision to remove the second term in quantifying node-level structural entropy.
>
> |Quantification of node-level structural entropy  | 70%   | 50%   | 20%   | 10%   | 5%    | 2%    | 1%    |
> |---------------------------|-------|-------|-------|-------|-------|-------|-------|
> | $S_e(u)=\frac{1}{vol(V)}(\sum_{\langle u, v \rangle\in E} w_{u,v}\log vol(\alpha_{u\vee v})-d(u)\log d(u))$     | $\underline{94.51}$ | $\underline{93.39}$ | $\underline{88.01}$ | $\underline{79.73}$ | $\underline{67.75}$ | $\underline{53.88}$ | $\underline{43.02}$ |
> | $S_e(u)=\frac{1}{vol(V)}\sum_{\langle u, v \rangle\in E} w_{u,v}\log vol(\alpha_{u\vee v})$  | **95.01** | **94.50** | **88.31** | **80.24** | **69.82** | **54.78** | **45.25** |
>
> For the **concern on missing experimental time**, we have included the realistic time for experiments on the CC SBU Align dataset in **Sec. 5.1.2**.
> Our method reduces the fine-tuning time on a single Nvidia Tesla V100 GPU from approximately 30 minutes to 3 minutes, adding only a negligible selection overhead of 2 seconds.
>
> ## **Q2**: The concern on time complexity.
>
> **A2**:
> In our experiments, we empirically found that an appropriate value for $k$ is $\log_2n$, where $n$ is the number of samples.
> With this choice, the $k$NN graph has $O(n\log_2n)$ edges, resulting in a Shapley value calculation time complexity of $O(n\log_2n)$.
> This complexity is comparable to sorting and is lower than or equal to that of typical sample selection methods, which is at least $O(n\log_2n)$ when sorting is required and $O(n^2)$ when an enumeration of all pairs of samples is required.
> We agree with you that for extremely large datasets (e.g. DataComp [2] with billions of samples), this complexity becomes impractical.
> A potential solution is to split the dataset into bins before selection.
> Determining an optimal splitting method remains an open question and a promising direction for future research.
> We clarified this in **Appendix E.3**.

---

> ### Author Response · Authors · 2024-11-19
> **Response to Review (2/4)**
>
> ## **Q3**: Insufficient justification for global and local information.
>
> **A3**: Thank you for pointing this out.
> For the justification of global information, measured by node-level structural entropy, we have provided a theoretical justification in **Sec. 4.1** and **Appendix C**.
> For prioritizing high-difficulty samples (local information), previous research [3, 4] has demonstrated that training difficulty is an effective local metric for quantifying a sample’s impact on model performance.
> This is because prioritizing high-difficulty samples enables the model to focus on challenging cases, which are typically more informative for improving decision boundaries and generalization [3, 4].
> We make this clearer in **Sec. 4.2**.
>
> ## **Q4**: Comparative analysis with methods that prioritize diversity.
>
> **A4**: Thank you for the constructive suggestion.
> We have added two widely used baselines that prioritize diversity [5, 6]:
> - $k$-means [7], which selects the samples closest to $k$-means clustering centers.
> - $k$-DPP [8], which employs a determinantal point process to encourage diversity.
>
> The experimental results are shown in **Tables 1, 10, 11, 12, 13, 14, 15, 16** of our paper, and the corresponding discussion is added to **Sec. 5.1**.
> These two methods are competitive in low-sampling-rate settings because they ensure the coverage of the dataset.
> However, in high-sampling-rate settings, they face challenges in balancing diversity with sample importance, leading to suboptimal performance.
> Our method consistently performs better than these two methods across all datasets and sampling rates.
> This highlights the importance of balancing the selection of important samples with the preservation of sample diversity.
>
> ## **Q5**: Dependence on embedding quality.
>
> **A5**: We agree with you on the importance of embedding quality for selecting high-quality samples.
> Indeed, most sample selection methods rely on high-quality embeddings to measure sample similarities, which influence the assessment of diversity and coverage.
> Improving embedding quality can benefit many such methods, including ours.
> However, exploring embedding improvement strategies is beyond the scope of our paper.
> Following common practice, we assumed sufficiently high-quality embeddings are available.
> In our experiments, we used CLIP and SentenceBERT due to their demonstrated performance in generating high-quality embeddings that capture semantic similarities across various domains.
> We make this clearer in **Sec. 5.1.1**.
>
> ## **Q6**: Insufficient evaluation for continual learning.
>
> **A6**: Thank you for pointing this out.
> Given the diverse scenarios of continual learning and our limited computational resources, we focused on class-incremental learning in this paper, as it is the most representative scenario for continual learning [9].
> We make this clearer in **Sec. 5.3.1**.
> We have added three widely used baselines on class-incremental learning from prior continual learning studies [10, 11]:
> - $k$-center [12], which iteratively selects samples that are least similar to those already selected.
> - Gradient Matching [13], which selects samples whose average gradient closely approximates the average gradient of all samples.
> - FRCL [14], which optimizes a subset of samples to minimize the posterior uncertainty of the Gaussian process induced from the embedding representations.
>
> We have also added two memory sizes: 50 and 400.
> The experimental results are shown in **Tables 3, 18, 19** of our paper, and the corresponding discussion is added to **Sec. 5.3**.
> Our method consistently performs better than the new baselines across all datasets for both the original memory sizes (100 and 200) and the newly added memory sizes (50 and 400).
> These results validate the effectiveness of our method.
> We believe that a comprehensive evaluation of our method across various continual learning scenarios would be an interesting direction for future research.
>
> In addition, we have added: 1) two widely used baselines that prioritize diversity (**Sec. 5.1**) and 2) additional ablation studies to evaluate the quantification of node-level structural entropy, the methods for combining global and local metrics, and the replacement of graphs with hypergraphs (**Appendix G**).

---

> ### Author Response · Authors · 2024-11-19
> **Response to Review (3/4)**
>
> ## **Q7**: Effects of encoding tree construction.
>
> **A7**: The construction of the encoding tree is indeed an important step in our method.
> To the best of our knowledge, there are two methods for constructing the encoding tree: the one proposed by Li & Pan [15] and the one proposed by Zhu et al. [16].
> Both methods adopt the variations of the Huffman tree to construct the encoding tree.
> The key difference is that Zhu et al.'s method further compresses the tree to a certain height to improve efficiency in subsequent processing.
> We assessed the two methods by calculating the Pearson correlation of the resulting node-level structural entropy.
> The discussion is provided in **Appendix A**, and the results are as follows.
> The correlations on CIFAR10, CIFAR100, and ImageNet-1K exceed $0.99$.
> Therefore, the choice of the construction method has a negligible effect on the sample selection results.
> In this paper, we employed the more recent method proposed by Zhu et al due to its efficiency.
>
> |             | CIFAR10 | CIFAR100 | ImageNet-1K |
> |-------------|---------|----------|-------------|
> | Correlation | 0.996   | 0.999    | 0.992       |
>
>
> ## **Q8**: Importance score calculation.
>
> **A8**: In our method, there is a one-to-one correspondence between the data points and the graph nodes.
> We make this clearer in **Sec. 4**.
> Therefore, the importance score is calculated node-wise, which reflects the importance of the corresponding data point.
> Regarding the second question, since each data point corresponds uniquely to a single node, no data point appears in multiple nodes.
> This ensures that our method is unaffected by redundancy among nodes.
>
> ## **Q9**: Combination methods of global and local metrics.
>
> **A9**: We explored possible alternative methods to combine the global metric $S_e(u)$ and the local metric $S_t(u)$ other than the proposed multiplication.
> Specifically, we experimented with the sum ($S_e(u)+S_t(u)$), the harmonic mean ($\frac{S_e(u) S_t(u)}{S_e(u)+S_t(u)}$), and the maximum ($\max(S_e(u),S_t(u))$).
> The ablation study is provided in **Appendix G**, and the results on CIFAR10 are as follows.
> These results demonstrate that multiplication achieves the best average performance, validating our choice for combining the two metrics.
>
> | Combination method          | 70%   | 50%   | 20%   | 10%   | 5%    | 2%    | 1%    | Avg.   |
> |-----------------------------|-------|-------|-------|-------|-------|-------|-------|--------|
> |$S_e(u) + S_t(u)$              | 94.77 | 94.43 | 87.17 | 79.64 | 69.08 | 53.24 | 44.53 | 71.35  |
> | $\frac{S_e(u)S_t(u)}{S_e(u)+S_t(u)}$ | $\underline{94.84}$ | **94.64** | $\underline{88.11}$ | $\underline{79.97}$ | $\underline{69.75}$ | $\underline{53.95}$ | $\underline{44.88}$ | $\underline{71.88}$  |
> | $\max(S_e(u), S_t(u))$         | 94.22 | 93.22 | 87.24 | 78.33 | 68.49 | 53.04 | 43.30 | 70.60  |
> | $S_e(u) \cdot S_t(u)$          | **95.01** | $\underline{94.50}$ | **88.31** | **80.24** | **69.82** | **54.78** | **45.25** | **72.15**  |
>
>
>
> ## **Q10**: Theoretical explanation.
>
> **A10**: The theoretical explanation is provided in **Sec. 4.1** and **Appendix C**.
> We have proved that $S_e(u)$ provides a lower bound for the sample coverage.
> This result indicates that maximizing $S_e(u)$ during selection inherently improves sample coverage.
> Given the strong correlation between sample coverage and the empirical loss of learning algorithms [1], selecting samples with high node-level structural entropy effectively enhances model performance.

---

> ### Author Response · Authors · 2024-11-19
> **Response to Review (4/4)**
>
> ## References
>
> [1] Haizhong Zheng, Rui Liu, Fan Lai, and Atul Prakash. Coverage-Centric Coreset Selection for High Pruning Rates. In *Proceedings of the International Conference on Learning Representations*, 2022.
>
> [2] Samir Yitzhak Gadre, Gabriel Ilharco, Alex Fang, Jonathan Hayase, Georgios Smyrnis, Thao Nguyen, Ryan Marten, Mitchell Wortsman, Dhruba Ghosh, Jieyu Zhang, et al. DataComp: In Search of the Next Generation of Multimodal Datasets. In *Proceedings of Advances in Neural Information Processing Systems*, 2024.
>
> [3] Mansheej Paul, Surya Ganguli, and Gintare Karolina Dziugaite. Deep Learning on a Data Diet: Finding Important Examples Early in Training. In *Proceedings of Advances in Neural Information Processing
> Systems*, pp. 20596–20607, 2021.
>
> [4] Ben Sorscher, Robert Geirhos, Shashank Shekhar, Surya Ganguli, and Ari Morcos. Beyond Neural Scaling Laws: Beating Power Law Scaling via Data Pruning. In *Proceedings of Advances in Neural
> Information Processing Systems*, pp. 19523–19536, 2022.
>
> [5] Ke Shang, Tianye Shu, Hisao Ishibuchi, Yang Nan, and Lie Meng Pang. Benchmarking Large-Scale Subset Selection in Evolutionary Multi-Objective Optimization. *Information Sciences*, 622:755-770, 2023.
>
> [6] Elisa Celis, Vijay Keswani, Damian Straszak, Amit Deshpande, Tarun Kathuria, and Nisheeth Vishnoi. Fair and diverse DPP-based data summarization. In *Proceedings of International Conference on Machine Learning*, pp. 716-725, 2018.
>
> [7] Zhao Xu, Kai Yu, Volker Tresp, Xiaowei Xu, and Jizhi Wang. Representative Sampling for Text Classification Using Support Vector Machines. In *Proceedings of Advances in Information Retrieval*,
> pp. 393–407, 2003.
>
> [8] Alex Kulesza and Ben Taskar. $k$-DPPs: Fixed-Size Determinantal Point Processes. In *Proceedings of the International Conference on Machine Learning*, pp. 1193–1200, 2011.
>
> [9] Liyuan Wang, Xingxing Zhang, Hang Su, and Jun Zhu. A Comprehensive Survey of Continual Learning: Theory, Method and Application. *IEEE Transactions on Pattern Analysis and Machine Intelligence*, 46(8):5362-5383, 2024.
>
> [10] Zal{\'a}n Borsos, Mojmir Mutny, and Andreas Krause. Coresets via Bilevel Optimization for Continual Learning and Streaming. In *Proceedings of Advances in Neural Information Processing Systems*, pp. 14879–14890, 2020.
>
> [11] Jie Hao, Kaiyi Ji, and Mingrui Liu. Bilevel Coreset Selection in Continual Learning: A New Formulation and Algorithm. In *Proceedings of Advances in Neural Information Processing Systems*, pp. 51026–51049, 2024.
>
> [12] Ozan Sener and Silvio Savarese. Active Learning for Convolutional Neural Networks: A Core-Set Approach. In *Proceedings of the International Conference on Learning Representations*, 2018.
>
> [13] Trevor Campbell and Tamara Broderick. Automated Scalable Bayesian Inference via Hilbert Coresets. *Journal of Machine Learning Research*, 20(15):1–38, 2019.
>
> [14] Michalis K Titsias, Jonathan Schwarz, Alexander G de G Matthews, Razvan Pascanu, and Yee Whye Teh. Functional Regularisation for Continual Learning with Gaussian Processes. In *Proceedings of International Conference on Learning Representations*, 2020.
>
> [15] Angsheng Li and Yicheng Pan. Structural Information and Dynamical Complexity of Networks. *IEEE Transactions on Information Theory*, 62(6):3290–3339, 2016.
>
> [16] He Zhu, Chong Zhang, Junjie Huang, Junran Wu, and Ke Xu. HiTIN: Hierarchy-aware Tree Isomorphism Network for Hierarchical Text Classification. In *Proceedings of the Annual Meeting of the Association for Computational Linguistics*, pp. 7809–7821, 2023.
>
> We hope that our responses have addressed your concerns satisfactorily. We are grateful for your insightful comments, which have helped us improve the clarity and quality of our manuscript.

---

> ### Author Response · Authors · 2024-11-24
> **Looking forward to further feedback**
>
> We sincerely thank you again for your recognition of our work. As we mentioned in the previous response, we have revised the paper per your suggestions. We hope that these responses have addressed your concerns satisfactorily. We are grateful for your insightful comments, which have helped us improve the clarity and quality of our manuscript.
>
> If you have any additional comments, we will do our best to address them.

---

> > ### Comment · Reviewer_g3Pt · 2024-11-24
> > **Further feedback**
> >
> > I really appreciate the feedback and efforts from the authors. Some of the answers address part of my concerns. I raise my grade from 3 to 5, but I think the current version still has  several issues.
> >
> > 1) The limited technical novelty (most of ideas comes from previously proposed concepts). In addition, I think some of the descriptions might not be that rigorous.  For example, in the newly added line 232-233, it said improving S_e inherently improves P(u, r), based on the upper bound (5). I think this statement is not correct, at least in theory. If the upper bound is not tight, say S_e=1 and P(u, r)=10, even you double S_e, it cannot enlarge P(u, r). I can understand the authors try to give some heuristic hint from (5), but this is not quite solid, at least in my opinion.
> >
> > Also, in this revised version, the authors did not explain why the part "-d(u)log(d(u))" in (3) is negligible; they mainly talk about why the first term is important.
> >
> > 2) I do feel the experimental part is not that sufficient. Like the continual learning part.  It cannot be denied that the paper provides an detailed comparison of coreset selection methods. However, the effectiveness of this method compared to commonly used continual learning techniques, such as DER/DER++[1], GSS[2], GEM[3], and GCR[4], still needs to be verified. CL currently is already a very crowd area. So it is important to conduct a more comprehensive validation before claiming an improvement.
> > Also, learning scenarios were only validated for class-incremental learning.  I still suggest that Task-Incremental Learning and Task-Free Continual Learning should both be verified.
> >
> > [1] Dark experience for general continual learning: a strong, simple baseline. NeurIPs 2020.
> >
> > [2] Gradient based sample selection for online continual learning. NeurIPs 2019.
> >
> > [3] Gradient episodic memory for continual learning. NeurIPs 2017.
> >
> > [4] Gcr: Gradient coreset based replay buffer selection for continual learning. CVPR 2022.

---

> > > ### Author Response · Authors · 2024-11-25
> > > **Response to Feedback (2/2)**
> > >
> > > **References**
> > >
> > > [1] Pietro Buzzega, Matteo Boschini, Angelo Porrello, Davide Abati, and Simone Calderara. Dark Experience for General Continual Learning: a Strong, Simple Baseline. In *Proceedings of Advances in Neural Information Processing Systems*, pp. 15920–15930, 2020.
> > >
> > > [2] Rahaf Aljundi, Min Lin, Baptiste Goujaud, and Yoshua Bengio. Gradient based sample selection for online continual learning. In *Proceedings of Advances in Neural Information Processing Systems*, 2019.
> > >
> > > [3] David Lopez-Paz and Marc'Aurelio Ranzato. Gradient Episodic Memory for Continual Learning. In *Proceedings of Advances in Neural Information Processing Systems*, 2017.
> > >
> > > [4] Rishabh Tiwari, Krishnateja Killamsetty, Rishabh Iyer, and Pradeep Shenoy. GCR: Gradient Coreset Based Replay Buffer Selection for Continual Learning. In *Proceedings of the IEEE/CVF Conference on Computer Vision and Pattern Recognition*, pp. 99-108, 2022.

---

> ### Author Response · Authors · 2024-11-25
> **Response to Feedback (1/2)**
>
> Thank you for your timely feedback and constructive suggestions. We have carefully considered your feedback and conducted additional experiments to address your concerns. Please see our response below.
>
> Regarding the **concern on the theoretical justification**, we have conducted an empirical verification to show that $\frac{1}{nk^2R}\exp(\frac{1}{kR}S_e(u))$ provides a relatively tight lower bound for $P(u,r)$. In order to precisely calculate $P(u,r)$, we simulate the data using a Gaussian Mixture Model. For this model, the class centers $\mu_1, \dots, \mu_C$ are sampled from a standard Gaussian distribution $\mathcal{N}(0, I)$. Within each class $c$, the samples are drawn from $\mathcal{N}(\mu_c, I)$. To ensure that the synthetic data aligns with the scale of real-world datasets, we match the statistics of CIFAR10 by generating 10 classes with 5,000 samples per class. Our results show that 1) $P(u,r)/\big(\frac{1}{nk^2R}\exp(\frac{1}{kR}S_e(u))\big)$ ranges between $1.00$ and $1.45$ for $90$% of the samples; 2) $P(u,r)/\big(\frac{1}{nk^2R}\exp(\frac{1}{kR}S_e(u))\big)$ is less than $1.97$ for $99$% of the samples. These findings indicate that maximizing $S_e(u)$ during selection improves sample coverage. In the final version of the paper, we will include a detailed discussion of this empirical verification in **Appendix A**. Nonetheless, we agree that it is important to provide a tighter upper bound with theoretical guarantees. This would be a very promising topic for future research.
>
> Regarding the **insufficient experiments**, we carefully studied the four suggested papers. Three of them (DER++ [1], GSS [2], and GCR [3]) focus on replay memory construction, while the other one (GEM [4]) focuses on improving the downstream use of the replay memory in model training. We have included the three replay memory construction methods as baselines. Regarding GEM, as we focus on replay memory construction, we did not include it in our experiments. Nonetheless, we appreciate its importance and will discuss how GEM can enhance performance when paired with different construction methods in the final version of the paper. Moreover, we have run the experiments under the task-incremental learning scenario and are in the process of implementing the code necessary for the experiments under the task-free learning scenario. We started to implement and launch the experiments immediately after receiving your feedback. All these experiments are run on 40 GPUs, including 16 NVIDIA A800 GPUs, 8 NVIDIA A100 GPUs, and 16 NVIDIA GeForce RTX 3090 GPUs. According to our estimation, the experiments will take three weeks to complete. As the experiments are time-consuming, we report the initial results, the task-incremental learning scenario on CIFAR10 with a memory size of 100, as follows. In the initial results, our method performs better than the new baselines across learning scenarios. In the final version, we will include DER++, GSS, and GCR as baselines, discuss the use of GEM, and present the complete results in **Appendix F.3**.
>
>
> | Method                | Class-incremental learning  | Task-incremental learning   |
> | --------------------- | -------------------------- | -------------------------- |
> | Random                | 62.34$\pm$0.89             | 75.44$\pm$0.14             |
> | Moderate              | 61.51$\pm$0.46             | 74.08$\pm$0.30             |
> | CCS                   | 60.56$\pm$1.27             | 56.93$\pm$0.01             |
> | $\mathbb{D}^2$ Pruning | $\underline{64.54\pm0.63}$ | 74.47$\pm$1.11             |
> | GraphCut              | 61.02$\pm$1.17             | 74.90$\pm$0.73             |
> | Entropy               | 61.53$\pm$0.72             | 73.06$\pm$0.36             |
> | Forgetting            | 59.56$\pm$0.24             | 58.62$\pm$0.90             |
> | EL2N                  | 57.79$\pm$0.75             | 53.86$\pm$0.74             |
> | AUM                   | 58.32$\pm$0.46             | 53.28$\pm$1.04             |
> | Variance              | 58.69$\pm$0.24             | 52.45$\pm$1.13             |
> | $k$-center            | 61.47$\pm$1.71             | 72.52$\pm$0.35             |
> | Gradient Matching      | 61.65$\pm$0.98             | 73.85$\pm$0.11             |
> | FRCL                  | 61.67$\pm$1.02             | 74.80$\pm$0.13             |
> | iCaRL                 | 62.33$\pm$0.89             | 72.95$\pm$1.79             |
> | Greedy Coreset         | 63.18$\pm$0.84             | $\underline{76.30\pm1.56}$ |
> | BCSR                  | 63.23$\pm$2.60             | 75.68$\pm$0.08             |
> | DER++                 | 62.73$\pm$1.38             | 72.33$\pm$1.38             |
> | GSS                   | 63.41$\pm$0.35             | 75.32$\pm$0.14             |
> | GCR                   | 63.12$\pm$2.33             | 72.93$\pm$0.30             |
> | SES (Ours)             | **68.26$\pm$1.24**         | **80.20$\pm$1.17**         |

---

> ### Comment · Reviewer_g3Pt · 2024-11-26
> **Response**
>
> Thanks a lot to the authors for the detailed response to my questions. I will keep my score at this moment, but open to discussion with other reviewers and AC in the discussion phase after rebuttal.

---

> > ### Author Response · Authors · 2024-12-01
> > **Additional Experimental Results (1/2)**
> >
> > Thank you once again for your feedback and openness to further discussion.
> > Following your feedback, we have implemented the necessary code for experiments in both task-incremental and task-free learning scenarios.
> > As previously mentioned, these experiments are time-consuming and are still ongoing.
> > Fortunately, we obtained some initial results on CIFAR10 with memory sizes of $100$ and $200$ as follows.
> > The initial results indicate that our method performs better than the baselines across learning scenarios and memory sizes.
> > We will include the complete results in **Appendix F.3** of the final version.
> > If you have any further suggestions or questions, we will make every effort to address them.
> > | Memory size 100                    |      Class-incremental learning       | Task-incremental learning | Task-free learning |
> > | :--------------------- | :----------------: | :------------------------: | :---------------------------: |
> > | Random                 |   62.34$\pm$0.89   |       75.44$\pm$0.14       |        64.28$\pm$2.09         |
> > | Moderate               |   61.51$\pm$0.46   |       74.08$\pm$0.30       |        62.90$\pm$0.93         |
> > | CCS                    |   60.56$\pm$1.27   |       56.93$\pm$0.01       |        64.14$\pm$1.33         |
> > | $\mathbb{D}^2$ Pruning |   $\underline{64.54\pm0.63}$   |       74.47$\pm$1.11       |  $\underline{66.15\pm0.65}$   |
> > | GraphCut               |   61.02$\pm$1.17   |       74.90$\pm$0.73       |        63.30$\pm$4.07         |
> > | Entropy                |   61.53$\pm$0.72   |       73.86$\pm$0.36       |        64.79$\pm$1.86         |
> > | Forgetting             |   59.56$\pm$0.24   |       58.62$\pm$0.90       |        61.67$\pm$0.70         |
> > | EL2N                   |   57.79$\pm$0.75   |       53.86$\pm$0.74       |        60.51$\pm$3.64         |
> > | AUM                    |   58.32$\pm$0.46   |       53.28$\pm$1.64       |        61.63$\pm$0.32         |
> > | Variance               |   58.69$\pm$0.24   |       52.45$\pm$1.13       |        63.93$\pm$1.35         |
> > | $k$-center             |   61.47$\pm$1.71   |       72.52$\pm$0.35       |        65.19$\pm$3.38         |
> > | Gradient Matching      |   61.65$\pm$6.98   |       73.85$\pm$0.30       |        60.53$\pm$2.44         |
> > | FRCL                   |   61.67$\pm$1.02   |       74.80$\pm$0.13       |        61.26$\pm$0.79         |
> > | iCaRL                  |   62.33$\pm$0.89   |       72.95$\pm$1.79       |        63.03$\pm$2.75         |
> > | Greedy Coreset         |   63.18$\pm$0.84   | $\underline{76.30\pm1.56}$ |        62.63$\pm$4.24         |
> > | BCSR                   |   63.23$\pm$2.60   |       75.68$\pm$0.08       |        59.99$\pm$3.41         |
> > | DER++                  |   62.73$\pm$1.38   |       72.33$\pm$1.38       |        64.00$\pm$1.25         |
> > | GSS                    |   63.41$\pm$0.35   |       75.32$\pm$0.14       |        62.85$\pm$2.30         |
> > | GCR                    |   63.12$\pm$2.33   |       72.93$\pm$0.30       |        64.07$\pm$3.18         |
> > | SES (Ours)             | **68.26$\pm$1.24** |     **80.20$\pm$1.17**     |      **66.56$\pm$1.09**       |

---

> > ### Author Response · Authors · 2024-12-01
> > **Additional Experimental Results (2/2)**
> >
> > | Memory size 200                    |      Class-incremental learning       | Task-incremental learning | Task-free learning |
> > | ---------------------- | :----------------: | :------------------------: | :---------------------------: |
> > | Random                 |   63.69$\pm$1.58   |       75.82$\pm$1.99       |        61.54$\pm$0.09         |
> > | Moderate               |   63.01$\pm$0.58   |       75.67$\pm$0.03       |        62.52$\pm$2.70         |
> > | CCS                    |   61.04$\pm$1.73   |       58.81$\pm$0.34       |        62.23$\pm$0.45         |
> > | $\mathbb{D}^2$ Pruning |   66.08$\pm$1.45   |       77.03$\pm$0.04       |        64.30$\pm$0.40         |
> > | GraphCut               |   61.66$\pm$1.67   |       75.22$\pm$1.41       |        61.26$\pm$0.38         |
> > | Entropy                |   62.72$\pm$0.73   |       75.34$\pm$0.82       |        63.28$\pm$3.41         |
> > | Forgetting             |   61.38$\pm$1.22   |       58.28$\pm$0.85       |        62.72$\pm$1.22         |
> > | EL2N                   |   58.34$\pm$0.68   |       53.92$\pm$2.04       |        59.39$\pm$2.75         |
> > | AUM                    |   58.06$\pm$0.86   |       57.43$\pm$0.68       |        60.70$\pm$4.02         |
> > | Variance               |   57.77$\pm$0.34   |       54.89$\pm$2.54       |  $\underline{66.84\pm1.21}$   |
> > | $k$-center             |   62.74$\pm$1.31   |       74.76$\pm$0.11       |        63.91$\pm$1.33         |
> > | Gradient matching      |   62.65$\pm$1.11   |       76.31$\pm$0.89       |        60.56$\pm$1.86         |
> > | FRCL                   |   62.93$\pm$0.86   |       75.78$\pm$0.95       |        62.18$\pm$2.42         |
> > | iCaRL                  |   64.08$\pm$1.58   |       76.81$\pm$0.71       |        61.37$\pm$1.73         |
> > | Greedy Coreset         |   62.98$\pm$0.91   | $\underline{79.35\pm1.09}$ |        61.75$\pm$0.87         |
> > | BCSR                   |   64.59$\pm$2.86   |       79.26$\pm$6.28       |        59.54$\pm$0.44         |
> > | DER++                  |     62.65$\pm$ 0.02               |       76.86$\pm$0.73       |        60.61$\pm$1.48         |
> > | GSS                    |         62.73$\pm$0.37  |       76.79$\pm$0.80       |        60.51$\pm$3.09         |
> > | GCR                    |      $\underline{66.66\pm 1.06}$              |       75.09$\pm$0.34       |        63.75$\pm$1.71         |
> > | SES (Ours)             | **69.32$\pm$0.99** |     **81.61$\pm$0.57**     |      **67.33$\pm$0.73**       |

---

### Official Review · Reviewer_MHqY · 2024-11-04

**Soundness:** 3
**Presentation:** 3
**Contribution:** 3
**Rating:** 8
**Confidence:** 4

**Summary:**

The paper studies sample selection which aims to extract a small, representative subset from a larger dataset. The authors introduce a novel sample selection scheme, termed Structural-Entropy-based Sample Selection (SES), which uses and extends the concept of "structural entropy" (Li and Pan, 2016), which assesses how nodes and edges within a graph are hierarchically organized to form multi-level communities. The proposed scheme seeks to address the limitations of existing selection methods, which often prioritize local information and neglect the broader, global context of the samples.

The algorithm begins by constructing a k-NN graph G for the dataset based on similarity. It then calculates the structural entropy value for each node in G (referred to as node-level structural entropy). While structural entropy was originally defined for an entire graph rather than individual nodes, the authors extend this concept using the Shapley value method. Roughly speaking, node-level structural entropy calculates the average increase in structural entropy when a node is added to all potential subgraphs of G. After that, each node is assigned an importance value, which is the product of its structural entropy and training difficulty. An importance-biased blue noise sampling method is then used to select samples. instead of sampling solely based on importance scores, this sampling process prevents the selection of overly similar samples, thus maintaining diversity within the selected subset.

The effectiveness of the SES scheme is demonstrated through experimental studies and compared to other selection methods in various tasks such as supervised learning, active learning, and continual learning.

**Strengths:**

Overall, the proposed algorithm is intriguing due to its general empirical performance and some of the novel ideas it introduces. The concept of node-level structural entropy, which quantifies the contribution of an individual node to the global structural entropy of a graph, could be of independent interest. The experiments indicate that the proposed method outperforms existing selection methods in many common learning tasks. Furthermore, ablation studies validate the contribution of each module (node-level structural entropy and importance-biased blue noise sampling) to the overall effectiveness.

**Weaknesses:**

I believe the results could benefit from stronger theoretical justification. While the high-level ideas are reasonable, the mathematical properties are not mentioned in this paper. A deeper discussion on these aspects should be provided. Especially I am interested in what are the mathematical properties of the node-level structural entropy.

**Questions:**

Are there anything you can prove about the node-level structural entropy values?

---

> ### Author Response · Authors · 2024-11-19
> **Response to Review**
>
> We sincerely thank you for your valuable and helpful suggestions.
> We are encouraged by your positive feedback and carefully addressed the concern on the theoretical justification.
> Please see our response below.
>
> ## **Q1**: Theoretical justification and discussion.
>
> **A1**: We have added the theoretical justification and mathematical properties of the node-level structural entropy $S_e(u)$ in **Sec. 4.1** and **Appendix C**.
> Specifically, we have proved that $S_e(u)$ provides a lower bound for sample coverage.
> This result indicates that maximizing $S_e(u)$ during selection inherently improves sample coverage.
> Given the strong correlation between sample coverage and the empirical loss of learning algorithms [1], selecting samples with high node-level structural entropy effectively enhances model performance.
>
> We also empirically demonstrated the effectiveness of our method for quantifying node-level structural entropy through an ablation study.
> The ablation study is provided in **Appendix G**, and the results on CIFAR10 are as follows.
> These results demonstrate that removing the second term slightly increases performance across all sampling rates.
> This finding supports our decision to remove the second term in quantifying node-level structural entropy.
>
>
> |Quantification of node-level structural entropy  | 70%   | 50%   | 20%   | 10%   | 5%    | 2%    | 1%    |
> |---------------------------|-------|-------|-------|-------|-------|-------|-------|
> | $S_e(u)=\frac{1}{vol(V)}(\sum_{\langle u, v \rangle\in E} w_{u,v}\log vol(\alpha_{u\vee v})-d(u)\log d(u))$     | $\underline{94.51}$ | $\underline{93.39}$ | $\underline{88.01}$ | $\underline{79.73}$ | $\underline{67.75}$ | $\underline{53.88}$ | $\underline{43.02}$ |
> | $S_e(u)=\frac{1}{vol(V)}\sum_{\langle u, v \rangle\in E} w_{u,v}\log vol(\alpha_{u\vee v})$  | **95.01** | **94.50** | **88.31** | **80.24** | **69.82** | **54.78** | **45.25** |
>
> ## Reference
>
> [1] Haizhong Zheng, Rui Liu, Fan Lai, and Atul Prakash. Coverage-Centric Coreset Selection for High Pruning Rates. In *Proceedings of the International Conference on Learning Representations*, 2022.
>
> We hope that our responses have addressed your concerns satisfactorily. We are grateful for your insightful comments, which have helped us improve the clarity and quality of our manuscript.

---

> > ### Comment · Reviewer_MHqY · 2024-11-25
> >
> > I thank the authors for providing more detailed theoretical analysis for $S_e(u)$. The relation between $S_e(u)$ and sample coverage is interesting, and seems to justify a bit the use of $S_e(u)$ in the importance-biased process. I am pleased to keep my current score.

---

> > > ### Author Response · Authors · 2024-11-25
> > > **Thank You Very Much for Your Feedback**
> > >
> > > We sincerely thank you for your feedback. We are glad to hear that you find our theoretical analysis interesting.  We deeply appreciate your time and effort in reviewing our paper and providing insightful and constructive comments, which are valuable for improving our paper.

---

> ### Author Response · Authors · 2024-11-24
> **Looking forward to further feedback**
>
> We sincerely thank you again for your recognition of our work. As we mentioned in the previous response, we have revised the paper per your suggestions. We hope that these responses have addressed your concerns satisfactorily. We are grateful for your insightful comments, which have helped us improve the clarity and quality of our manuscript.
>
> If you have any additional comments, we will do our best to address them.

---

### Author Response · Authors · 2024-11-19
**General Response to Reviewers**

Dear reviewers,

We sincerely thank you for your valuable and insightful comments.
We have followed the suggestions and thoroughly revised our paper accordingly.
The changes are summarized below and marked in **blue** in the paper.

- Provided theoretical justifications for quantifying node-level structural entropy defined by Eq. (4) (**Sec. 4.1**).
- Added two baselines that prioritize diversity and three baselines for continual learning (**Secs. 5.1, 5.3**).
- Conducted additional ablation studies to evaluate the quantification of node-level structural entropy, the methods for combining global and local metrics, and the replacement of graphs with hypergraphs (**Appendix G**).
- Addressed other minor issues raised by the reviewers.


Please let us know if you have any additional suggestions.
We will be happy to revise our paper accordingly.
The detailed changes are enclosed in individual responses.

Sincerely,

The authors of #1606

---

### Meta-Review · Area_Chair_6EJk · 2024-12-20

**Metareview:**

The reviewers raised several concerns during the review process regarding largely around lack of novelty and lack of theoretical justification. The authors have addressed most of these concerns. In my opinion the work is somewhat incremental, but the empirical results are interesting and the overall consensus is towards acceptance.

**Additional Comments On Reviewer Discussion:**

The reviewers were active in the discussions with the authors.

---

### Decision · Program_Chairs · 2025-01-22

Accept (Poster)